# A Geometric View of Data Complexity: Efficient Local Intrinsic Dimension Estimation with Diffusion Models

**Hamidreza Kamkari      Brendan Leigh Ross      Rasa Hosseinzadeh**
**Jesse C. Cresswell      Gabriel Loaiza-Ganem**
`{hamid, brendan, rasa, jesse, gabriel}@layer6.ai`
Layer 6 AI, Toronto, Canada

## Abstract

High-dimensional data commonly lies on low-dimensional submanifolds, and estimating the *local intrinsic dimension* (LID) of a datum – i.e. the dimension of the submanifold it belongs to – is a longstanding problem. LID can be understood as the number of local factors of variation: the more factors of variation a datum has, the more complex it tends to be. Estimating this quantity has proven useful in contexts ranging from generalization in neural networks to detection of out-of-distribution data, adversarial examples, and AI-generated text. The recent successes of deep generative models present an opportunity to leverage them for LID estimation, but current methods based on generative models produce inaccurate estimates, require more than a single pre-trained model, are computationally intensive, or do not exploit the best available deep generative models: diffusion models (DMs). In this work, we show that the Fokker-Planck equation associated with a DM can provide an LID estimator which addresses the aforementioned deficiencies. Our estimator, called FLIPD, is easy to implement and compatible with all popular DMs. Applying FLIPD to synthetic LID estimation benchmarks, we find that DMs implemented as fully-connected networks are highly effective LID estimators that outperform existing baselines. We also apply FLIPD to natural images where the true LID is unknown. Despite being sensitive to the choice of network architecture, FLIPD estimates remain a useful measure of relative complexity; compared to competing estimators, FLIPD exhibits a consistently higher correlation with image PNG compression rate and better aligns with qualitative assessments of complexity. Notably, FLIPD is orders of magnitude faster than other LID estimators, and the first to be tractable at the scale of Stable Diffusion.

## 1   Introduction

The manifold hypothesis [7], which has been empirically verified in contexts ranging from natural images [49, 10] to calorimeter showers in physics [15], states that high-dimensional data of interest in $\mathbb{R}^D$ often lies on low-dimensional submanifolds of $\mathbb{R}^D$. For a given datum $x \in \mathbb{R}^D$, this hypothesis motivates using its *local intrinsic dimension* (LID), denoted $\mathrm{LID}(x)$, as a natural measure of its complexity. $\mathrm{LID}(x)$ corresponds to the dimension of the data manifold that $x$ belongs to, and can be intuitively understood as the minimal number of variables needed to describe $x$. More complex data needs more variables to be adequately described, as illustrated in Figure 1. Data manifolds are typically not known explicitly, meaning that LID must be estimated. Here we tackle the following problem: given a dataset along with a query datum $x$, how can we tractably estimate $\mathrm{LID}(x)$?

This is a longstanding problem, with LID estimates being highly useful due to their innate interpretation as a measure of complexity. For example, these estimates can be used to detect outliers [27, 2, 32], AI-generated text [64], and adversarial examples [42]. Connections between the generalization achieved by a neural network and the LID estimates of its internal representations have also

38th Conference on Neural Information Processing Systems (NeurIPS 2024).

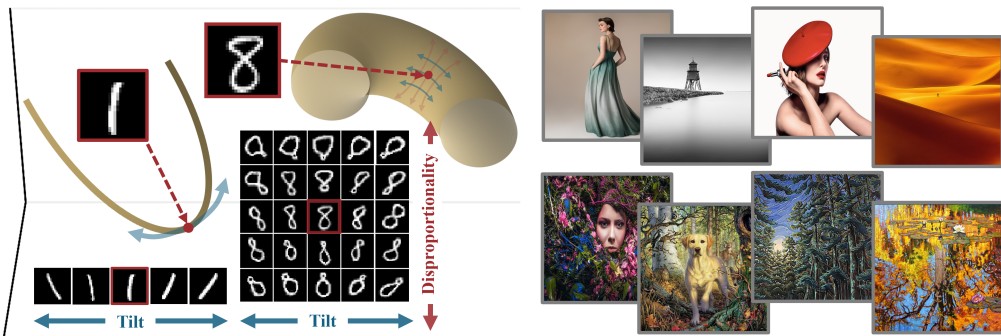

Figure 1: **(Left)** A cartoon illustration showing that LID is a natural measure of relative complexity. We depict two manifolds of MNIST digits, corresponding to 1s and 8s, as 1-dimensional and 2-dimensional submanifolds of $\mathbb{R}^3$, respectively. The relatively simpler manifold of 1s exhibits a single factor of variation ("tilt"), whereas 8s have an additional factor of variation ("disproportionality"). **(Right)** The 4 lowest- and highest-LID datapoints from a subsample of LAION-Aesthetics, as measured by our method, FLIPD, applied to Stable Diffusion v1.5. FLIPD scales efficiently to large models on high-dimensional data, and aligns closely with subjective complexity.

been shown [4, 8, 44, 9]. These insights can be leveraged to identify which representations contain maximal semantic content [66], and help explain why LID estimates can be helpful as regularizers [72] and for pruning large models [70]. LID estimation is thus not only of mathematical and statistical interest, but can also benefit the empirical performance of deep learning models at numerous tasks.

Traditional estimators of intrinsic dimension [22, 38, 43, 12, 30, 20, 1, 5] typically rely on pairwise distances and nearest neighbours, so computing them is prohibitively expensive for large datasets. Recent work has thus sought to move away from these *model-free* estimators and instead take advantage of deep generative models which learn the distribution of observed data. When this distribution is supported on low-dimensional submanifolds of $\mathbb{R}^D$, successful generative models must implicitly learn the dimensions of the data submanifolds, suggesting they can be used to construct LID estimators. However, existing *model-based* estimators suffer from various drawbacks, including being inaccurate and computationally expensive [59], not leveraging the best existing generative models [62, 71] (i.e. diffusion models [56, 25, 58]), and requiring training several models [62] or altering the training procedure rather than relying on a pre-trained model [26]. Importantly, *none* of these methods scale to high-resolution images such as those generated by Stable Diffusion [51].

We address these issues by showing how LID can be efficiently estimated using only a single pre-trained diffusion model (DM) by building on LIDL [62], a model-based estimator. LIDL operates by convolving data with different levels of Gaussian noise, training a normalizing flow [50, 17, 18] for each level, and fitting a linear regression using the log standard deviation of the noise as a covariate and the corresponding log density (of the convolution) evaluated at $x$ as the response; the resulting slope is an estimate of $\text{LID}(x) - D$, thanks to a surprising result linking Gaussian convolutions and LID. We first show how to adapt LIDL to DMs in such a way that only a single DM is required (rather than many normalizing flows). Directly applying this insight leads to LIDL estimates that require one DM but several calls to an ordinary differential equation (ODE) solver; we further show how to circumvent this with an alternative ODE that computes all the required log densities in a single solve. We then argue that the slope of the regression in LIDL aims to capture the rate of change of the marginal log probabilities in the diffusion process, which can be evaluated directly thanks to the Fokker-Planck equation. The resulting estimator, which we call FLIPD,[1] is highly efficient, and circumvents the need for an ODE solver. Notably, FLIPD is *differentiable*; this property opens up exciting avenues for future research as it enables backpropagating through LID estimates.

Our contributions are: $(i)$ showing how DMs can be efficiently combined with LIDL in a way which requires a single call to an ODE solver; $(ii)$ leveraging the Fokker-Planck equation to propose FLIPD, thus improving upon the estimator and circumventing the need for an ODE solver altogether; $(iii)$ motivating FLIPD theoretically; $(iv)$ introducing an expanded suite of LID estimation benchmark

---

[1]Pronounced as "flipped", the acronym is a rearrangement of "FP" from Fokker-Planck and "LID".

tasks that reveals gaps in prior evaluations, and specifically that other estimators do not remain accurate as the complexity of the manifold increases; $(v)$ demonstrating that when using fully-connected architectures for diffusion models, FLIPD outperforms existing baselines – especially as dimension increases – while being much more computationally efficient; $(vi)$ showing that when applied to natural images, despite varying across network architecture (i.e., fully-connected network or UNet [52, 68]), FLIPD estimates consistently align with other measures of complexity such as PNG compression length, and with qualitative assessments of complexity, highlighting that the LID estimates provided by FLIPD remain valid measures of relative image complexity; and $(vii)$ demonstrating that when applied to the latent space of Stable Diffusion, FLIPD can estimate LID for extremely high-resolution images ($\sim 10^6$ pixel dimensions) for the first time.

## 2 Background and Related Work

### 2.1 Diffusion Models

**Forward and backward processes**   Diffusion models admit various formulations [56, 25]; here we follow the score-based one [58]. We denote the true data-generating distribution, which DMs aim to learn, as $p(\cdot, 0)$. DMs define the forward (Itô) stochastic differential equation (SDE),

$$\mathrm{d}X_t = f(X_t, t)\mathrm{d}t + g(t)\mathrm{d}W_t, \quad X_0 \sim p(\cdot, 0), \tag{1}$$

where $f : \mathbb{R}^D \times [0, 1] \to \mathbb{R}^D$ and $g : [0, 1] \to \mathbb{R}$ are hyperparameters, and $W_t$ denotes a $D$-dimensional Brownian motion. We write the distribution of $X_t$ as $p(\cdot, t)$. The SDE in Equation 1 prescribes how to gradually add noise to data, the idea being that $p(\cdot, 1)$ is essentially pure noise. Defining the backward process as $Y_t := X_{1-t}$, this process obeys the backward SDE [3, 24],

$$\mathrm{d}Y_t = \left[ g^2(1-t)s(Y_t, 1-t) - f(Y_t, 1-t) \right] \mathrm{d}t + g(1-t)\mathrm{d}\hat{W}_t, \quad Y_0 \sim p(\cdot, 1), \tag{2}$$

where $s(x, t) := \nabla \log p(x, t)$ is the unknown (Stein) score function,[2] and $\hat{W}_t$ is another $D$-dimensional Brownian motion. DMs leverage this backward SDE for generative modelling by using a neural network $\hat{s} : \mathbb{R}^D \times (0, 1] \to \mathbb{R}^D$ to learn the score function with denoising score matching [67]. Once trained, $\hat{s}(x, t) \approx s(x, t)$. To generate samples $\hat{Y}_1$ from the model, we solve an approximation of Equation 2:

$$\mathrm{d}\hat{Y}_t = \left[ g^2(1-t)\hat{s}(\hat{Y}_t, 1-t) - f(\hat{Y}_t, 1-t) \right] \mathrm{d}t + g(1-t)\mathrm{d}\hat{W}_t, \quad \hat{Y}_0 \sim \hat{p}(\cdot, 1), \tag{3}$$

with $\hat{s}$ replacing the true score and with $\hat{p}(\cdot, 1)$, a Gaussian distribution chosen to approximate $p(\cdot, 1)$ (depending on $f$ and $g$), replacing $p(\cdot, 1)$.

**Density Evaluation**   DMs can be interpreted as continuous normalizing flows [13], and thus admit density evaluation, meaning that if we denote the distribution of $\hat{Y}_{1-t}$ as $\hat{p}(\cdot, t)$, then $\hat{p}(x, t_0)$ can be mathematically evaluated for any given $x \in \mathbb{R}^D$ and $t_0 \in (0, 1]$. More specifically, this is achieved thanks to the (forward) ordinary differential equation (ODE) associated with the DM:

$$\mathrm{d}\hat{x}_t = \left( f(\hat{x}_t, t) - \frac{1}{2}g^2(t)\hat{s}(\hat{x}_t, t) \right) \mathrm{d}t, \quad \hat{x}_{t_0} = x. \tag{4}$$

Solving this ODE from time $t_0$ to time 1 produces the trajectory $(\hat{x}_t)_{t \in [t_0, 1]}$, which can then be used for density evaluation through the continuous change-of-variables formula:

$$\log \hat{p}(x, t_0) = \log \hat{p}(\hat{x}_1, 1) + \int_{t_0}^1 \mathrm{tr}(\nabla v(\hat{x}_t, t))\mathrm{d}t, \tag{5}$$

where $\hat{p}(\cdot, 1)$ can be evaluated since it is a Gaussian, and where $v(x, t) := f(x, t) - g^2(t)\hat{s}(x, t)/2$.

---

[2]Throughout this paper, we use the symbol $\nabla$ to denote the differentiation operator with respect to the vector-valued input, not the scalar time $t$, i.e. $\nabla = \partial/\partial x$.

**Trace estimation** Note that the cost of computing $\nabla v(\hat{x}_t, t)$ for a particular $\hat{x}_t$ amounts to $\Theta(D)$ function evaluations of $\hat{s}$ (since $D$ calls to a Jacobian-vector-product routine are needed [6]). Although this is not prohibitively expensive for a single $\hat{x}_t$, in order to compute the integral in Equation 5 in practice, $(\hat{x}_t)_{t \in [t_0, 1]}$ must be discretized into a trajectory of length $N$. If we denote by $F$ the cost of evaluating $v(\hat{x}_t, t)$ – or equivalently, $\hat{s}(\hat{x}_t, t)$ – deterministic density evaluation is $\Theta(NDF)$, which is computationally prohibitive. The Hutchinson trace estimator [29] – which states that for $M \in \mathbb{R}^{D \times D}$, $\mathrm{tr}(M) = \mathbb{E}_\varepsilon[\varepsilon^\top M \varepsilon]$, where $\varepsilon \in \mathbb{R}^D$ has mean 0 and covariance $I_D$ – is thus commonly used for stochastic density estimation; approximating the expectation with $k$ samples from $\varepsilon$ results in a cost of $\Theta(NkF)$, which is much faster than deterministic density evaluation when $k \ll D$.

## 2.2 Local Intrinsic Dimension and How to Estimate It

**LID** Various definitions of intrinsic dimension exist [28, 21, 37, 11]. Here we follow the standard one from geometry: a $d$-dimensional manifold is a set which is locally homeomorphic to $\mathbb{R}^d$. For a given disjoint union of manifolds and a point $x$ in this union, the *local intrinsic dimension* of $x$ is the dimension of the submanifold it belongs to. Note that LID is not an intrinsic property of the point $x$, but rather a property of $x$ with respect to the manifold that contains it. Intuitively, $\mathrm{LID}(x)$ corresponds to the number of factors of variation present in the manifold containing $x$, and it is thus a natural measure of the relative complexity of $x$, as illustrated in Figure 1.

**Estimating LID** The natural interpretation of LID as a measure of complexity makes estimating it from observed data a relevant problem. Here, the formal setup is that $p(\cdot, 0)$ is supported on a disjoint union of manifolds [10], and we assume access to a dataset sampled from it. Then, for a given $x$ in the support of $p(\cdot, 0)$, we want to use the dataset to provide an estimate of $\mathrm{LID}(x)$. Traditional estimators [22, 38, 43, 12, 30, 20, 1, 5] rely on the nearest neighbours of $x$ in the dataset, or related quantities, and typically have poor scaling in dataset size. Generative models are an intuitive alternative to these methods; because they are trained to learn $p(\cdot, 0)$, when they succeed they must encode information about the support of $p(\cdot, 0)$, including the corresponding manifold dimensions. However, extracting this information from a trained generative model is not trivial. For example, Zheng et al. [71] showed that the number of active dimensions in the approximate posterior of variational autoencoders [33, 50] estimates LID, but their approach does not generalize to better generative models.

**LIDL** Tempczyk et al. [62] proposed LIDL, a method for LID estimation relying on normalizing flows as tractable density estimators [50, 17, 18]. LIDL works thanks to a surprising result linking Gaussian convolutions and LID [39, 62, 71]. We will denote the convolution of $p(\cdot, 0)$ and Gaussian noise with log standard deviation $\delta$ as $\varrho(\cdot, \delta)$, i.e.

$$\varrho(x, \delta) := \int p(x_0, 0) \mathcal{N}(x - x_0; 0, e^{2\delta} I_D) \mathrm{d}x_0. \tag{6}$$

The aforementioned result states that, under mild regularity conditions on $p(\cdot, 0)$, and for a given $x$ in its support, the following holds as $\delta \to -\infty$:

$$\log \varrho(x, \delta) = \delta(\mathrm{LID}(x) - D) + \mathcal{O}(1). \tag{7}$$

This result then suggests that, for negative enough values of $\delta$ (i.e. small enough standard deviations):

$$\log \varrho(x, \delta) \approx \delta(\mathrm{LID}(x) - D) + c, \tag{8}$$

for some constant $c$. If we could evaluate $\log \varrho(x, \delta)$ for various values of $\delta$, this would provide an avenue for estimating $\mathrm{LID}(x)$: set some values $\delta_1, \ldots, \delta_m$, fit a linear regression using $\{(\delta_i, \log \varrho(x, \delta_i))\}_{i=1}^m$ with $\delta$ as the covariate and $\log \varrho(x, \delta)$ as the response, and let $\hat{\beta}_x$ be the corresponding slope. It follows that $\hat{\beta}_x$ estimates $\mathrm{LID}(x) - D$, so that $\mathrm{LID}(x) \approx D + \hat{\beta}_x$ is a sensible estimator of local intrinsic dimension.

Since $\varrho(x, \delta)$ is unknown and cannot be evaluated, LIDL requires training $m$ normalizing flows. More specifically, for each $\delta_i$, a normalizing flow is trained on data to which $\mathcal{N}(0, e^{2\delta_i} I_D)$ noise is added. In LIDL, the log densities of the trained models are then used instead of the unknown true log densities $\log \varrho(x, \delta_i)$ when fitting the regression as described above.

Despite using generative models, LIDL has obvious drawbacks. LIDL requires training several models. It also relies on normalizing flows, which are not only empirically outperformed by DMs by

a wide margin, but are also known to struggle to learn low-dimensional manifolds [14, 39, 40]. On the other hand, DMs do not struggle to learn $p(\cdot, 0)$ even when it is supported on low-dimensional manifolds [48, 16, 40], further suggesting that LIDL can be improved by leveraging DMs.

**Estimating LID with DMs** The only works we are aware of that leverage DMs for LID estimation are those of Stanczuk et al. [59], and Horvat and Pfister [26]. The latter modifies the training procedure of DMs, so we focus on the former since we see compatibility with existing pre-trained models as an important requirement for DM-based LID estimators. Stanczuk et al. [59] consider *variance-exploding* DMs, where $f = 0$. They show that, as $t \searrow 0$, the score function $s(x, t)$ points orthogonally towards the manifold containing $x$, or more formally, it lies in the normal space of this manifold at $x$. They thus propose the following LID estimator, which we refer to as the normal bundle (NB) estimator: first run Equation 1 until time $t_0$ starting from $x$, and evaluate $\hat{s}(\cdot, t_0)$ at the resulting value; then repeat this process $K$ times and stack the $K$ resulting $D$-dimensional vectors into a matrix $S(x) \in \mathbb{R}^{D \times K}$. The idea here is that if $t_0$ is small enough and $K$ is large enough, the columns of $S(x)$ span the normal space of the manifold at $x$, suggesting that the rank of this matrix estimates the dimension of this normal space, namely $D - \text{LID}(x)$. Finally, they estimate $\text{LID}(x)$ as:

$$\text{LID}(x) \approx D - \text{rank } S(x). \tag{9}$$

Numerically, the rank is computed by performing a singular value decomposition (SVD) of $S(x)$, setting a threshold, and counting the number of singular values exceeding the threshold. Computing $S(x)$ requires $K$ function evaluations, and intuitively $K$ should be large enough to ensure the columns of $S(x)$ span the normal space at $x$; the authors thus propose using $K = 4D$, and recommend always at least ensuring that $K > D$. Computing the NB estimator costs $\Theta(KF + D^2K)$, where $F$ again denotes the cost for evaluating $\hat{s}$. Thus, although the NB estimator addresses some of the limitations of LIDL, it remains computationally expensive in high dimensions.

## 3 Method

Although Tempczyk et al. [62] only used normalizing flows in LIDL, they did point out that these models could be swapped for any other generative model admitting density evaluation. Indeed, one could trivially train $m$ DMs and replace the flows with them. Throughout this section we provide a sequence of progressive improvements to this naïve application of LIDL with DMs, culminating with FLIPD. We assume access to a pre-trained DM such that $f(x, t) = b(t)x$ for a function $b : [0, 1] \to \mathbb{R}$. This choice implies that the transition kernel $p_{t|0}$ associated with Equation 1 is Gaussian [53]:

$$p_{t|0}(x_t \mid x_0) = \mathcal{N}(x_t; \psi(t)x_0, \sigma^2(t)I_D), \tag{10}$$

where $\psi, \sigma : [0, 1] \to \mathbb{R}$. We also assume that $b$ and $g$ are such that $\psi$ and $\sigma$ are differentiable and such that $\lambda(t) := \sigma(t)/\psi(t)$ is injective. This setting encompasses all DMs commonly used in practice, including variance-exploding, variance-preserving (of which the widely used DDPMs [25] are a discretized instance), and sub-variance-preserving [58]. In Appendix A we include explicit formulas for $\psi(t)$, $\sigma^2(t)$, and $\lambda(t)$ for these particular DMs.

### 3.1 LIDL with a Single Diffusion Model

As opposed to the several normalizing flows used in LIDL which are individually trained on datasets with different levels of noise added, a single DM already works by convolving data with various noise levels and allows density evaluation of the resulting noisy distributions (Equation 5). Hence, we make the observation that LIDL can be used with a *single* DM. All we need is to relate $\varrho(\cdot, \delta)$ to the density of the DM, $p(\cdot, t)$. In the case of variance-exploding DMs, $\psi(t) = 1$, so we can easily use the defining property of the transition kernel in Equation 10 to get

$$p(x, t) = \int p(x_0, 0)p_{t|0}(x_t \mid x_0)\mathrm{d}x_0 = \int p(x_0, 0)\mathcal{N}(x_t; x_0, \sigma^2(t)I_D)\mathrm{d}x_0, \tag{11}$$

which equals $\varrho(x, \delta)$ from Equation 6 when we choose $t = \sigma^{-1}(e^\delta)$.[3] In turn, we can use LIDL with a single variance-exploding DM by evaluating each $\hat{p}(x, \sigma^{-1}(e^{\delta_i}))$ through Equation 5. This idea

---

[3]Note that we treat positive and negative superindices differently: e.g. $\sigma^{-1}$ denotes the inverse function of $\sigma$, not $1/\sigma$; on the other hand $\sigma^2$ denotes the square of $\sigma$, not $\sigma \circ \sigma$.

extends beyond variance-exploding DMs; in Appendix B.1 we show that for *any* arbitrary DM with transition kernel as in Equation 10, it holds that

$$\log \varrho(x,\delta) = D \log \psi\big(t(\delta)\big) + \log p\Big(\psi\big(t(\delta)\big)x, t(\delta)\Big), \quad (12)$$

where $t(\delta) \coloneqq \lambda^{-1}(e^\delta)$. This equation is relevant because LIDL requires $\log \varrho(\cdot, \delta)$, yet DMs provide $\log p(\cdot, t)$: linking these two quantites as above shows that LIDL can be used with a single DM.

## 3.2 A Better Implementation of LIDL with a Single Diffusion Model

Using Equation 12 with LIDL still involves computing $\log \hat{p}\big(\psi(t(\delta_i))x, t(\delta_i)\big)$ through Equation 5 for each $i = 1, \ldots, m$ before running the regression. Since each of the corresponding ODEs in Equation 4 starts at a different time $t_0 = t(\delta_i)$ and is evaluated at a different point $\psi(t(\delta_i))x$, this means that a different ODE solver call would have to be used for each $i$, resulting in a prohibitively expensive procedure. To address this, we aim to find an explicit formula for $\partial/\partial\delta \log \varrho(x,\delta)$. We do so by leveraging the Fokker-Planck equation associated with Equation 1, which provides an explicit formula for $\partial/\partial t\, p(x,t)$. Using this equation along with the chain rule and Equation 12, we show in Appendix B.2 that, for DMs with transition kernel as in Equation 10,

$$\frac{\partial}{\partial \delta} \log \varrho(x, \delta) = \sigma^2\big(t(\delta)\big) \Bigg( \mathrm{tr}\left(\nabla s\big(\psi(t(\delta))x, t(\delta)\big)\right) + \Big\| s\big(\psi(t(\delta))x, t(\delta)\big) \Big\|_2^2 \Bigg) =: \nu\big(t(\delta); s, x\big). \quad (13)$$

Then, assuming without loss of generality that $\delta_1 < \cdots < \delta_m$, we can use the above equation to define $\log \hat{\varrho}(x, \delta)$ through the ODE

$$\mathrm{d} \log \hat{\varrho}(x, \delta) = \nu\big(t(\delta); \hat{s}, x\big)\mathrm{d}\delta, \quad \log \hat{\varrho}(x, \delta_1) = 0. \quad (14)$$

Solving this ODE from $\delta_1$ to $\delta_m$ produces the trajectory $(\log \hat{\varrho}(x,\delta))_{\delta \in [\delta_1, \delta_m]}$. Since $\nu(t(\delta); \hat{s}, x)$ does not depend on $\hat{\varrho}(x, \delta)$, when $\hat{s} = s$ the solution to the ODE above will be off by a constant, i.e., $\log \hat{\varrho}(x, \delta) = \log \varrho(x, \delta) + c_{\mathrm{init}}$ for some $c_{\mathrm{init}}$ that depends on the initial condition of the ODE (0 in this case) but not on $\delta$. Furthermore, while setting the initial condition to 0 might at first appear odd, recall that LIDL fits a regression using $\{(\delta_i, \log \hat{\varrho}(x, \delta_i))\}_{i=1}^m$, and thus $c_{\mathrm{init}}$ will be absorbed in the intercept without affecting the slope. In other words, the initial condition is irrelevant, and we can use LIDL with DMs by using a single call to an ODE solver on Equation 14.

## 3.3 FLIPD: An Efficient Fokker-Planck-Based LID Estimator

The LIDL estimator with DMs presented in Section 3.2 provides a massive speedup over the naïve approach of training $m$ DMs, and over the method from Section 3.1 requiring $m$ ODE solves. Yet, solving Equation 14 involves computing the trace of the Jacobian of $\hat{s}$ multiple times within an ODE solver, which, as mentioned in Section 2.1, remains expensive. In this section we present our LID estimator, FLIPD, which circumvents the need for an ODE solver altogether. Recall that LIDL is based on Equation 8, which justifies the regression. Differentiating this equation yields that $\partial/\partial\delta \log \varrho(x, \delta_0) \approx \mathrm{LID}(x) - D$ for negative enough $\delta_0$, meaning that Equation 13 directly provides the rate of change that the regression in LIDL aims to estimate, from which we get

$$\mathrm{LID}(x) \approx D + \frac{\partial}{\partial\delta} \log \varrho(x, \delta_0) = D + \nu\big(t(\delta_0); s, x\big) \approx D + \nu\big(t(\delta_0); \hat{s}, x\big) =: \mathrm{FLIPD}(x, t_0), \quad (15)$$

where $t_0 \coloneqq t(\delta_0)$. Computing FLIPD is very cheap since the trace of the Jacobian of $\hat{s}$ has to be evaluated only once when calculating $\nu(t(\delta_0); \hat{s}, x)$. As mentioned in Section 2.1, exact evaluation of this trace has a cost of $\Theta(DF)$, but can be reduced to $\Theta(kF)$ when using the stochastic Hutchinson trace estimator with $k \ll D$ samples. Notably, FLIPD provides a massive speedup over the NB estimator – $\Theta(kF)$ vs. $\Theta(KF + D^2K)$ – especially in high dimensions where $K > D \gg k$. In addition to not requiring an ODE solver, computing FLIPD requires setting only a single hyperparameter, $\delta_0$. Furthermore, since $\nu(t(\delta); \hat{s}, x)$ depends on $\delta$ only through $t(\delta)$, we can directly set $t_0$ as the hyperparameter rather than $\delta_0$, which avoids the potentially cumbersome computation of $t(\delta_0) = \lambda^{-1}(e^{\delta_0})$: instead of setting a suitably negative $\delta_0$, we set $t_0 > 0$ sufficiently close to 0. In Appendix A we include explicit formulas for $\mathrm{FLIPD}(x, t_0)$ for common DMs.

Finally, we present a theoretical result further justifying FLIPD. Note that the $\mathcal{O}(1)$ term in Equation 7 need not be constant in $\delta$ as in Equation 8, even if it is bounded. The more this term deviates from a constant, the more bias we should expect in both LIDL and FLIPD. The following result shows that in an idealized linear setting, FLIPD is unaffected by this problem:

**Theorem 3.1** (FLIPD Soundness: Linear Case). *Let $\mathcal{L}$ be an embedded submanifold of $\mathbb{R}^D$ given by a $d$-dimensional affine subspace. If $p(\cdot, 0)$ is supported on $\mathcal{L}$, continuous, and with finite second moments, then for any $x \in \mathcal{L}$ with $p(x, 0) > 0$, we have:*

$$\lim_{\delta \to -\infty} \frac{\partial}{\partial \delta} \log \varrho(x, \delta) = d - D. \tag{16}$$

*Proof.* See Appendix B.3. □

We conjecture that our theorem can be extended to non-linear submanifolds since it is a local result and every manifold can be locally linearly approximated by its tangent plane. More specifically, $\varrho(x, \delta)$ becomes "increasingly local as $\delta \to -\infty$" in the sense that its dependence on the values $p(x_0, 0)$ becomes negligible as $\delta \to -\infty$ when $x_0$ is not in a close enough neighbourhood of $x$; this is because $\mathcal{N}(0, e^{2\delta} I_D)$ concentrates most of its mass around 0 as $\delta \to -\infty$ (see Equation 6). However, we leave generalizing our result to future work.

## 4 Experiments

Throughout this section, we use variance-preserving DMs, the most popular variant of DMs. We provide a "dictionary" to translate between the score-based formulation of FLIPD and DDPMs in Appendix C. We hope this will enable practitioners who are less focused on the theoretical aspects to effortlessly apply FLIPD to their pre-trained DMs. Our code is available at https://github.com/layer6ai-labs/flipd; see Appendix D for more experimental details.

### 4.1 Experiments on Synthetic Data

**The effect of $t_0$** FLIPD requires setting $t_0$ close to 0 since all the theory holds in the $\delta \to -\infty$ regime. It is important to note that DMs fitted to low-dimensional manifolds are known to exhibit numerically unstable scores $s(\cdot, t_0)$ as $t_0 \searrow 0$ [65, 41, 40]. This fact does not invalidate FLIPD, but it suggests that it might be sensitive to the choice of $t_0$. Our first set of experiments examines the effect of $t_0$ on FLIPD$(x, t_0)$ by varying $t_0$ within the range $(0, 1)$.

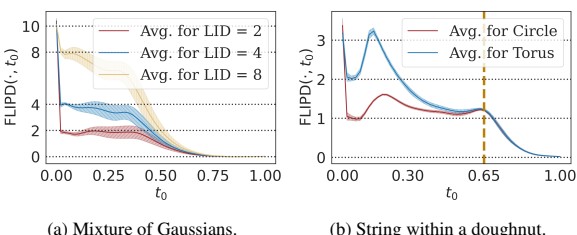

(a) Mixture of Gaussians.    (b) String within a doughnut.

Figure 2: FLIPD curves with knees at the true LID.

In Figure 2, we train DMs on two distributions: $(i)$ a mixture of three isotropic Gaussians with dimensions 2, 4, and 8, embedded in $\mathbb{R}^{10}$ (each embedding is carried out by multiplication against a random matrix with orthonormal columns plus a random translation); and $(ii)$ a "string within a doughnut", which is a mixture of uniform distributions on a 2d torus (with a major radius of 10 and a minor radius of 1) and a 1d circle (aligning with the major circle of the torus) embedded in $\mathbb{R}^3$ (this union of manifolds is shown in the upper half of Figure 3). While FLIPD$(x, t_0)$ is inaccurate at $t_0 = 0$ due to the aforementioned instabilities, it quickly stabilizes around the true LID for all datapoints. We refer to this pattern as a *knee* in the FLIPD curve. In Appendix D.2, we show similar curves for more complex data manifolds.

**FLIPD is a multiscale estimator** Interestingly, in Figure 2b we see that the blue FLIPD curve (corresponding to "doughnut" points with LID of 2) exhibits a second knee at 1, located at the $t_0$ shown with a vertical line. This confirms the multiscale nature of convolution-based estimators, first postulated by Tempczyk et al. [62] in the context of normalizing flows; they claim that when selecting a log standard deviation $\delta$, all directions along which a datum can vary having log standard deviation less than $\delta$ are ignored. The second knee in Figure 2b can be explained by a similar argument: the torus looks like a 1d circle when viewed from far away, and larger values of $t_0$ correspond to viewing the manifolds from farther away. This is visualized in Figure 3 with two views of the "string within a doughnut" and corresponding LID estimates: one zoomed-in view where $t_0$ is small, providing fine-grained LID estimates, and a zoomed-out view where $t_0$ is large, making both the string and

Table 1: MAE (lower is better) │ concordance indices (higher is better with 1.0 being the gold standard). Rows show synthetic manifolds and columns represent LID estimation methods. Columns are grouped based on whether they use a generative model, with the best results for each metric within each group being bolded.

| Synthetic Manifold | Model-based | | | | | | Model-free | | | |
|---|---|---|---|---|---|---|---|---|---|---|
| | FLIPD | | NB | | LIDL | | ESS | | LPCA | |
| String within doughnut $\subseteq \mathbb{R}^3$ | **0.06** | 1.00 | 1.48 | 0.48 | 1.10 | 0.99 | 0.02 | 1.00 | **0.00** | 1.00 |
| $\mathcal{L}_5 \subseteq \mathbb{R}^{10}$ | 0.17 | - | 1.00 | - | **0.10** | - | 0.07 | - | **0.00** | - |
| $\mathcal{N}_{90} \subseteq \mathbb{R}^{100}$ | 0.49 | - | **0.18** | - | 0.33 | - | **1.67** | - | 21.9 | - |
| $\mathcal{U}_{10} + \mathcal{U}_{30} + \mathcal{U}_{90} \subseteq \mathbb{R}^{100}$ | **1.30** | 1.00 | 61.6 | 0.34 | 8.46 | 0.74 | 21.9 | 0.74 | **20.1** | 0.86 |
| $\mathcal{N}_{10} + \mathcal{N}_{25} + \mathcal{N}_{50} \subseteq \mathbb{R}^{100}$ | **1.81** | 1.00 | 74.2 | 0.34 | 8.87 | 0.74 | 7.71 | 0.88 | **5.72** | 0.91 |
| $\mathcal{F}_{10} + \mathcal{F}_{25} + \mathcal{F}_{50} \subseteq \mathbb{R}^{100}$ | **3.93** | 1.00 | 74.2 | 0.34 | 18.6 | 0.70 | 9.20 | 0.90 | **6.77** | 1.00 |
| $\mathcal{U}_{10} + \mathcal{U}_{80} + \mathcal{U}_{200} \subseteq \mathbb{R}^{800}$ | **14.3** | 1.00 | 715 | 0.34 | 120 | 0.70 | 1.39 | 1.00 | **0.01** | 1.00 |
| $\mathcal{U}_{900} \subseteq \mathbb{R}^{1000}$ | 12.8 | - | 100 | - | 24.9 | - | **14.5** | - | 219 | - |

doughnut appear as a 1d circle from this distance. In Appendix D.3 we have an experiment that makes the multiscale argument explicit.

**Finding knees** As mentioned, we persistently see knees in FLIPD curves. This in line with the observations of Tempczyk et al. [62] (see Figure 5 of [62]), and it gives us a fully automated approach to setting $t_0$. We leverage kneedle [54], a knee detection algorithm which aims to find points of maximum curvature. When computationally sensible, rather than fixing $t_0$, we evaluate Equation 15 for 50 values of $t_0$ and pass the results to kneedle to automatically detect the $t_0$ where a knee occurs.

**Experimental setup** We create a benchmark for LID evaluation on complex unions of manifolds where the true LID is known. We sample from simple distributions on low-dimensional spaces, and then embed the samples into $\mathbb{R}^D$. We denote uniform, Gaussian, and Laplace distributions as $\mathcal{U}, \mathcal{N}$, and $\mathcal{L}$, respectively, with sub-indices indicating LID, and a plus sign denoting mixtures. To embed samples into higher dimensions, we apply a random matrix with orthonormal columns and then apply a random translation. For example, $\mathcal{N}_{10} + \mathcal{L}_{20} \subseteq \mathbb{R}^{100}$ indicates a 10-dimensional Gaussian and a 20-dimensional Laplace, each of which undergoes a random affine transformation

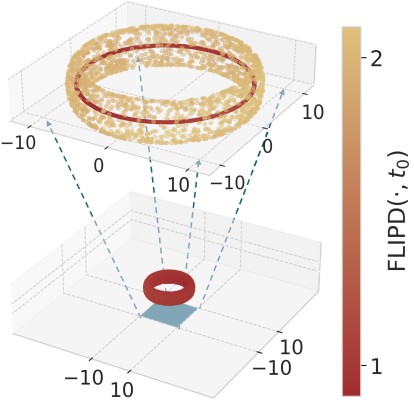

Figure 3: "String within a doughnut" manifolds, and corresponding FLIPD estimates for different values of $t_0$ ($t_0 = 0.05$ on top and $t_0 = 0.65$ on bottom). These results highlight the multiscale nature of FLIPD.

mapping to $\mathbb{R}^{100}$ (one transformation per component). We also generate non-linear manifolds, denoted with $\mathcal{F}$, by applying a randomly initialized $D$-dimensional neural spline flow [18] after the affine transformation (when using flows, the input noise is always uniform); since the flow is a diffeomorphism, it preserves LID. To our knowledge, this synthetic LID benchmark is the most extensive to date, revealing surprising deficiencies in some well-known traditional estimators. For an in-depth analysis, see Appendix D.4.

**Results** Here, we summarize our synthetic experiments in Table 1 using two metrics of performance: the mean absolute error (MAE) between the predicted and true LID for individual datapoints; and the concordance index, which measures similarity in the rankings between the true LIDs and the estimated ones (note that this metric only makes sense when the dataset has variability in its ground truth LIDs, so we only report it for the appropriate entries in Table 1). We compare against the NB and LIDL estimators described in Section 2.2, as well as two of the most performant model-free baselines: LPCA [22, 12] and ESS [30]. For the NB baseline, we use the exact same DM backbone as for FLIPD (since NB was designed for variance-exploding DMs, we use the adaptation to variance-preserving DMs used in [32], which produces extremely similar results), and for LIDL we use 8 neural spline flows. In terms of MAE, we find that FLIPD tends to be the best model-based estimator, particularly as dimension increases. Although model-free baselines perform well in simplistic scenarios, they produce unreliable results as LID increases or more non-linearity is introduced in the data manifold. In terms of concordance index, FLIPD achieves *perfect* scores in all scenarios, meaning that even

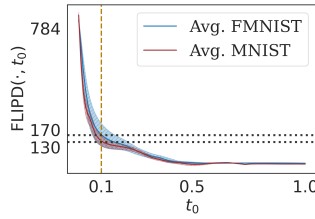
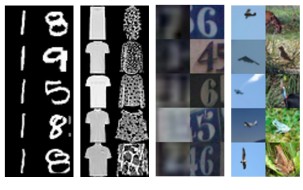
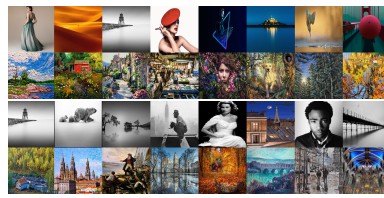

(a) FLIPD of MNIST and FMNIST.

(b) Ordering small images.

(c) Ordering high-resolution images in LAION using FLIPD (top) and PNG compression size (bottom).

Figure 4: Overview of image LID: **(a)** shows the FLIPD curves that are used to estimate average LID for MNIST and FMNIST when using MLP backbones; **(b)** compares images with small and large FLIPD estimates from FMNIST, MNIST, SVHN, and CIFAR10 when using UNet backbones; and **(c)** compares LAION images with small and large FLIPD estimates using Stable Diffusion (top, $t_0 = 0.3$) and PNG compression sizes (bottom).

when its estimates are off, it always provides correct LID rankings. We include additional results in the appendices: in Appendix D.5 we ablate FLIPD, finding that using `kneedle` indeed helps, and that FLIPD also outperforms the efficient implementation of LIDL with DMs described in Section 3.2 that uses an ODE solver. We notice that NB with the setting proposed in [59] consistently produces estimates that are almost equal to the ambient dimension; thus, in Appendix D.6 we also show how NB can be significantly improved upon by using `kneedle`, although it is still outperformed by FLIPD in many scenarios. In addition, in Table 7 and Table 8 we compare against other model-free baselines such as MLE [38, 43] and FIS [1]. We also consider datasets with a single (uni-dimensional) manifold where the average LID estimate can be used to approximate the *global* intrinsic dimension. Our results in Table 9 demonstrate that although model-free baselines indeed accurately estimate global intrinsic dimension, they perform poorly when focusing on (pointwise) LID estimates.

## 4.2 Experiments with Fully-Connected Architectures on Image Data

We first focus on the simple image datasets MNIST [36] and FMNIST [69]. We flatten the images and use the same MLP architecture as in our synthetic experiments. Despite using an MLP, our DMs can generate reasonable samples (Appendix E.1) and the FLIPD curve for both MNIST and FMNIST is shown in Figure 4a. The knee points are identified at $t_0 = 0.1$, resulting in average LID estimates of approximately 130 and 170, respectively. Evaluating LID estimates for image data is challenging due to the lack of ground truth. Although our LID estimates are higher than those in [49] and [10], our experiments (Table 9 of Appendix D.4) and the findings in [62] and [59] show that model-free baselines underestimate LID of high-dimensional data, especially images.

## 4.3 Experiments with UNet Architectures on Image Data

When moving to more complex image datasets, the MLP backbone fails to generate high-quality samples. Therefore, we replace it with state-of-the-art UNets [52, 68] (see Appendix E.2). Surprisingly, we find that using `kneedle` with UNets fails to produce sensible LID estimates with FLIPD (see curves in Figure 10 of Appendix E.1). We discuss why this might be the case in Appendix E.1, and from here on we simply set $t_0$ as a hyperparameter instead of using `kneedle`. Although avoiding `kneedle` when using UNets results in increased sensitivity with respect to $t_0$, we argue that FLIPD remains a valuable measure of complexity as it produces sensible image rankings and it is highly correlated with with PNG compression length. We took random subsets of 4096 images from each of FMNIST, MNIST, SVHN [46], and CIFAR10 [35], and sorted them according to their FLIPD estimates (obtained using UNet backbones). We show the top and bottom 5 images for each dataset in Figure 4b, and include more samples in Appendix E.3. Our visualization shows that higher FLIPD estimates indeed correspond to images with more detail and texture, while lower estimates correspond to less complex ones. Additionally, we show in Appendix E.4 that using only $k = 50$ Hutchinson samples to approximate the trace term in FLIPD is sufficient for small values of $t_0$.

Further, we quantitatively assess our estimates by computing Spearman's rank correlation coefficient between different LID estimators and PNG compression size, used as a proxy for complexity in the absence of ground truth. We highlight that although we expect this coefficient to be high, a perfect LID estimator need not achieve a correlation of 1. As shown in Table 2, FLIPD has a high correlation

with PNG, whereas model-free estimators do not. We find that the NB estimator correlates slightly more with PNG on MNIST and CIFAR10, but significantly less in FMNIST and SVHN. Moreover, in Appendix E.5, we analyze how increasing $t_0$ affects FLIPD by re-computing the correlation with the PNG size at different values of $t_0 \in (0, 1)$. We see that as $t_0$ increases, the correlation with PNG decreases. Despite this decrease, we observe an interesting phenomenon: while the image

Table 2: Spearman's correlation between LID estimates and PNG compression size. FLIPD and NB were computed using the same UNet backbone.

| Method | MNIST | FMNIST | CIFAR10 | SVHN |
|--------|-------|--------|---------|------|
| FLIPD | 0.837 | **0.883** | 0.819 | **0.876** |
| NB | **0.864** | 0.480 | **0.894** | 0.573 |
| ESS | 0.444 | 0.063 | 0.326 | 0.019 |
| LPCA | 0.413 | 0.01 | 0.302 | $-0.008$ |

orderings change, qualitatively, the smallest $\text{FLIPD}(\cdot, t_0)$ estimates still represent less complex data compared to the highest $\text{FLIPD}(\cdot, t_0)$, even for relatively large $t_0$. We hypothesize that for larger $t_0$, similar to the "string within a doughnut" experiment in Figure 3, the orderings correspond to coarse-grained and semantic notions of complexity rather than fine-grained ones such as textures, concepts that a metric such as PNG compression size cannot capture.

We also consider high-resolution images from LAION-Aesthetics [55] and, for the first time, estimate LID for extremely high-dimensional images with $D = 3 \times 512 \times 512 = 786{,}432$. We use Stable Diffusion [51], a latent DM pretrained on LAION-5B [55]. This includes an encoder and a decoder trained to preserve relevant characteristics of the data manifold in latent representations. Since the encoder and decoder are continuous and effectively invert each other, we argue that the Stable Diffusion encoder can, for practical purposes, be considered a topological embedding of the LAION-5B dataset into its latent space of dimension $4 \times 64 \times 64 = 16{,}384$. Therefore, the dimension of the LAION-5B submanifold in latent space should be unchanged. We leave an empirical verification of this hypothesis to future work and thus estimate image LIDs by carrying out FLIPD in the latent space of Stable Diffusion. Here, we set the Hutchinson sample count to $k = 1$, meaning we only require a *single* Jacobian-vector-product. When we order a random subset of 1600 samples according to their FLIPD at $t_0 = 0.3$, the more complex images are clustered at the end, while the least complex are clustered at the beginning: see Figure 1 and Figure 4c for the lowest- and highest-LID images from this ordering, and Figure 25 in Appendix E.6 to view the entire subset and other values of $t_0$. In comparison to orderings according to PNG compression size (Figure 4c), FLIPD prioritizes semantic complexity over low-level details like colouration.

Finally, we compare the runtimes for computing FLIPD and NB for all models using UNet backbones. We show results in Table 3. For $28 \times 28$ greyscale (MNIST/FMNIST) and $3 \times 32 \times 32$ low-resolution RGB (SVHN/CIFAR10) images, we use

Table 3: Time, in seconds, to estimate LID for a single image.

| Method | MNIST/FMNIST | SVHN/CIFAR10 | LAION |
|--------|--------------|--------------|-------|
| FLIPD | **0.10** | **0.13** | **0.20** |
| NB | 1.6 | 10.8 | $> 9 \times 10^3$ |

$k = 50$ Hutchinson samples: at these dimensions, FLIPD achieves $\sim 10\times$ and $\sim 100\times$ respective speedups over NB. For $4 \times 64 \times 64$ LAION-Aesthetics images on the latent space of Stable Diffusion, we use $k = 1$ for FLIPD, which ensures it remains highly tractable. At this resolution NB is completely intractable: constructing $S(x)$ for a single image $x$ takes 2.5 hours, and computing NB would then still require performing a SVD on this $16{,}384 \times 65{,}536$ matrix.

## 5 Conclusions, Limitations, and Future Work

In this work we have shown that the Fokker-Planck equation can be utilized for efficient LID estimation with any pre-trained DM. We have provided strong theoretical foundations and extensive benchmarks showing that FLIPD estimates accurately reflect data complexity. Although FLIPD produces excellent LID estimates on synthetic benchmarks, its instability with respect to the choice of network architecture on images is surprising, and results in LID estimates which strongly depend on this choice. We see this behaviour as a limitation, even if FLIPD nonetheless still provides a meaningful measure of complexity regardless of architecture. Given that FLIPD is tractable, differentiable, and compatible with any DM, we hope that it will find uses in applications where LID estimates have already proven helpful, including OOD detection, AI-generated data analysis, and adversarial example detection.

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

# A Explicit Formulas

## A.1 Variance-Exploding Diffusion Models

Variance-exploding DMs are such that $f(x,t) = 0$ with $g$ being non-zero. In this case [58]:

$$\psi(t) = 1, \quad \text{and} \quad \sigma^2(t) = \int_0^t g^2(u)\mathrm{d}u. \tag{17}$$

Since $g$ is non-zero, $g^2$ is positive, so that $\sigma^2$ is increasing, and thus injective. It follows that $\sigma$ is also injective, so that $\lambda = \sigma/\psi = \sigma$ is injective. Equation 15 then implies that

$$\mathrm{FLIPD}(x,t_0) = D + \sigma^2(t_0)\left[\mathrm{tr}\left(\nabla s(x,t_0)\right) + \|s(x,t_0)\|_2^2\right]. \tag{18}$$

## A.2 Variance-Preserving Diffusion Models (DDPMs)

Variance-preserving DMs are such that

$$f(x,t) = -\frac{1}{2}\beta(t)x, \quad \text{and} \quad g(t) = \sqrt{\beta(t)}, \tag{19}$$

where $\beta$ is a positive scalar function. In this case [58]:

$$\psi(t) = e^{-\frac{1}{2}B(t)}, \quad \text{and} \quad \sigma^2(t) = 1 - e^{-B(t)}, \quad \text{where} \quad B(t) := \int_0^t \beta(u)\mathrm{d}u. \tag{20}$$

We then have that

$$\lambda(t) = \sqrt{\frac{\sigma^2(t)}{\psi^2(t)}} = \sqrt{e^{B(t)} - 1}. \tag{21}$$

Since $\beta$ is positive, $B$ is increasing and thus injective, from which it follows that $\lambda$ is injective as well. Plugging everything into Equation 15, we obtain:

$$\mathrm{FLIPD}(x,t_0) = D + \left(1 - e^{-B(t_0)}\right)\left(\mathrm{tr}\left(\nabla s\left(e^{-\frac{1}{2}B(t_0)}x, t_0\right)\right) + \left\|s\left(e^{-\frac{1}{2}B(t_0)}x, t_0\right)\right\|_2^2\right). \tag{22}$$

## A.3 Sub-Variance-Preserving Diffusion Models

Sub-variance-preserving DMs are such that

$$f(x,t) = -\frac{1}{2}\beta(t)x, \quad \text{and} \quad g(t) = \sqrt{\beta(t)\left(1 - e^{-2B(t)}\right)}, \quad \text{where} \quad B(t) := \int_0^t \beta(u)\mathrm{d}u, \tag{23}$$

and where $\beta$ is a positive scalar function. In this case [58]:

$$\psi(t) = e^{-\frac{1}{2}B(t)}, \quad \text{and} \quad \sigma^2(t) = \left(1 - e^{-B(t)}\right)^2. \tag{24}$$

We then have that

$$\lambda(t) = \frac{\sigma(t)}{\psi(t)} = e^{\frac{1}{2}B(t)} - e^{-\frac{1}{2}B(t)} = 2\sinh\left(\frac{1}{2}B(t)\right). \tag{25}$$

Since $\beta$ is positive, $B$ is increasing and thus injective, from which it follows that $\lambda$ is injective as well due to the injectivity of $\sinh$. Plugging everything into Equation 15, we obtain:

$$\mathrm{FLIPD}(x,t_0) = D + \left(1 - e^{-B(t_0)}\right)^2\left(\mathrm{tr}\left(\nabla s\left(e^{-\frac{1}{2}B(t_0)}x, t_0\right)\right) + \left\|s\left(e^{-\frac{1}{2}B(t_0)}x, t_0\right)\right\|_2^2\right). \tag{26}$$

# B Proofs and Derivations

## B.1 Derivation of Equation 12

We have that:

$$p\big(\psi(t(\delta))x, t(\delta)\big) = \int p_{t(\delta)|0}\Big(\psi\big(t(\delta)\big)x \mid x_0\Big)p(x_0, 0)\mathrm{d}x_0 \tag{27}$$

$$= \int \mathcal{N}\Big(\psi\big(t(\delta)\big)x; \psi\big(t(\delta)\big)x_0, \sigma^2\big(t(\delta)\big)I_D\Big)p(x_0, 0)\mathrm{d}x_0 \tag{28}$$

$$= \frac{1}{\psi\big(t(\delta)\big)^D}\int \mathcal{N}\left(x; x_0, \frac{\sigma^2\big(t(\delta)\big)}{\psi^2\big(t(\delta)\big)}I_D\right)p(x_0, 0)\mathrm{d}x_0 \tag{29}$$

$$= \frac{1}{\psi\big(t(\delta)\big)^D}\int \mathcal{N}\Big(x; x_0, \lambda^2\big(t(\delta)\big)I_D\Big)p(x_0, 0)\mathrm{d}x_0 \tag{30}$$

$$= \frac{1}{\psi\big(t(\delta)\big)^D}\int \mathcal{N}\big(x; x_0, e^{2\delta}I_D\big)p(x_0, 0)\mathrm{d}x_0 = \frac{1}{\psi\big(t(\delta)\big)^D}\varrho(x, \delta), \tag{31}$$

where we used that $\mathcal{N}(ax; ax_0, \sigma^2 I_D) = \frac{1}{a^D}\mathcal{N}(x; x_0, (\sigma^2/a^2)I_D)$, along with the definition $\lambda(t) := \sigma(t)/\psi(t)$. It is thus easy to see that taking logarithms yields Equation 12.

## B.2 Derivation of Equation 13

First, we recall the Fokker-Planck equation associated with the SDE in Equation 1, which states that:

$$\frac{\partial}{\partial t}p(x, t) = -p(x, t)\left[\nabla \cdot f(x, t)\right] - \langle f(x, t), \nabla p(x, t)\rangle + \frac{1}{2}g^2(t)\,\mathrm{tr}\left(\nabla^2 p(x, t)\right). \tag{32}$$

We begin by using this equation to derive $\partial/\partial t \log p(x, t)$. Noting that $\nabla p(x, t) = p(x, t)s(x, t)$, we have that:

$$\mathrm{tr}\left(\nabla^2 p(x, t)\right) = \mathrm{tr}\left(\nabla\left[p(x, t)s(x, t)\right]\right) = p(x, t)\,\mathrm{tr}\left(\nabla s(x, t)\right) + \mathrm{tr}\left(s(x, t)\nabla p(x, t)^\top\right) \tag{33}$$

$$= p(x, t)\left[\mathrm{tr}\left(\nabla s(x, t)\right) + \|s(x, t)\|_2^2\right]. \tag{34}$$

Because

$$\frac{\partial}{\partial t}p(x, t) = p(x, t)\frac{\partial}{\partial t}\log p(x, t), \tag{35}$$

it then follows that

$$\frac{\partial}{\partial t}\log p(x, t) = -\left[\nabla \cdot f(x, t)\right] - \langle f(x, t), s(x, t)\rangle + \frac{1}{2}g^2(t)\left[\mathrm{tr}\left(\nabla s(x, t)\right) + \|s(x, t)\|_2^2\right]. \tag{36}$$

Then, from Equation 12 and the chain rule, we get:

$$\frac{\partial}{\partial \delta}\log \varrho(x, \delta) = \frac{\mathrm{d}}{\mathrm{d}\delta}\left[D\log \psi\big(t(\delta)\big) + \log p\Big(\psi\big(t(\delta)\big)x, t(\delta)\Big)\right] \tag{37}$$

$$= D\left[\frac{\mathrm{d}}{\mathrm{d}\delta}\log \psi\big(t(\delta)\big)\right]$$

$$+ \begin{pmatrix}\nabla \log p\Big(\psi\big(t(\delta)\big)x, t(\delta)\Big) \\ \frac{\partial}{\partial t}\log p\Big(\psi\big(t(\delta)\big)x, t(\delta)\Big)\end{pmatrix}^\top \begin{pmatrix}\frac{\partial}{\partial t}\psi\big(t(\delta)\big)x \\ 1\end{pmatrix}\frac{\partial}{\partial \delta}t(\delta) \tag{38}$$

$$= \left[\frac{\partial}{\partial \delta}t(\delta)\right]\left[D\frac{\frac{\partial}{\partial t}\psi\big(t(\delta)\big)}{\psi\big(t(\delta)\big)} + \begin{pmatrix}s\Big(\psi\big(t(\delta)\big)x, t(\delta)\Big) \\ \frac{\partial}{\partial t}\log p\Big(\psi\big(t(\delta)\big)x, t(\delta)\Big)\end{pmatrix}^\top \begin{pmatrix}\frac{\partial}{\partial t}\psi\big(t(\delta)\big)x \\ 1\end{pmatrix}\right] \tag{39}$$

$$= \left[\frac{\partial}{\partial \delta}t(\delta)\right]\left[\left(\frac{\partial}{\partial t}\psi\big(t(\delta)\big)\right)\left(\frac{D}{\psi\big(t(\delta)\big)} + \Big\langle x, s\big(\psi\big(t(\delta)\big)x, t(\delta)\big)\Big\rangle\right)\right.$$

$$\left.+ \frac{\partial}{\partial t}\log p\Big(\psi\big(t(\delta)\big)x, t(\delta)\Big)\right]. \tag{40}$$

Substituting Equation 36 into Equation 40 yields:

$$\frac{\partial}{\partial\delta}\log\varrho(x,\delta) = \left[\frac{\partial}{\partial\delta}t(\delta)\right]\left[\left(\frac{\partial}{\partial t}\psi\big(t(\delta)\big)\right)\left(\frac{D}{\psi\big(t(\delta)\big)} + \Big\langle x, s\Big(\psi\big(t(\delta)\big)x, t(\delta)\Big)\Big\rangle\right)\right.$$
$$- \left[\nabla\cdot f\Big(\psi\big(t(\delta)\big)x, t(\delta)\Big)\right] - \Big\langle f\Big(\psi\big(t(\delta)\big)x, t(\delta)\Big), s\Big(\psi\big(t(\delta)\big)x, t(\delta)\Big)\Big\rangle$$
$$\left. + \frac{1}{2}g^2\big(t(\delta)\big)\left(\operatorname{tr}\left(\nabla s\Big(\psi\big(t(\delta)\big)x, t(\delta)\Big)\right) + \Big\|s\Big(\psi\big(t(\delta)\big)x, t(\delta)\Big)\Big\|_2^2\right)\right]. \quad (41)$$

From now on, to simplify notation, when dealing with a scalar function $h$, we will denote its derivative as $h'$. Since $t(\delta) = \lambda^{-1}(e^\delta)$, the chain rule gives:

$$\frac{\partial}{\partial\delta}t(\delta) = \frac{e^\delta}{\lambda'\big(\lambda^{-1}(e^\delta)\big)} = \frac{\lambda\big(t(\delta)\big)}{\lambda'\big(t(\delta)\big)}. \quad (42)$$

So far, we have not used that $f(x,t) = b(t)x$, which implies that $\nabla\cdot f(x,t) = Db(t)$ and that $\langle f(x,t), s(x,t)\rangle = b(t)\langle x, s(x,t)\rangle$. Using these observations and Equation 42, Equation 41 becomes:

$$\frac{\partial}{\partial\delta}\log\varrho(x,\delta) = \frac{\lambda\big(t(\delta)\big)}{\lambda'\big(t(\delta)\big)}\left[\left(\frac{\psi'\big(t(\delta)\big)}{\psi\big(t(\delta)\big)} - b\big(t(\delta)\big)\right)D\right.$$
$$+ \Big\langle\Big(\psi'\big(t(\delta)\big) - b\big(t(\delta)\big)\psi\big(t(\delta)\big)\Big)x, s\Big(\psi\big(t(\delta)\big)x, t(\delta)\Big)\Big\rangle$$
$$\left. + \frac{1}{2}g^2\big(t(\delta)\big)\left(\operatorname{tr}\left(\nabla s\Big(\psi\big(t(\delta)\big)x, t(\delta)\Big)\right) + \Big\|s\Big(\psi\big(t(\delta)\big)x, t(\delta)\Big)\Big\|_2^2\right)\right]. \quad (43)$$

If we showed that

$$\psi'(t) - b(t)\psi(t) = 0, \quad \text{and that} \quad \frac{\lambda(t)}{2\lambda'(t)}g^2(t) = \sigma^2(t), \quad (44)$$

for every $t$, then Equation 43 would simplify to Equation 13. From equation 5.50 in [53], we have that

$$\psi'(t) = b(t)\psi(t), \quad (45)$$

which shows that indeed $\psi'(t) - b(t)\psi(t) = 0$. Then, from equation 5.51 in [53], we also have that

$$\big(\sigma^2\big)'(t) = 2b(t)\sigma^2(t) + g^2(t), \quad (46)$$

and from the chain rule this gives that

$$\sigma'(t) = \frac{2b(t)\sigma^2(t) + g^2(t)}{2\sigma(t)}. \quad (47)$$

We now finish verifying Equation 44. Since $\lambda(t) = \sigma(t)/\psi(t)$, the chain rule implies that

$$\frac{\lambda(t)}{2\lambda'(t)}g^2(t) = \frac{\dfrac{\sigma(t)}{\psi(t)}}{\dfrac{\sigma'(t)\psi(t) - \sigma(t)\psi'(t)}{\psi^2(t)}}\frac{g^2(t)}{2} = \frac{\sigma(t)\psi(t)}{\sigma'(t)\psi(t) - \sigma(t)b(t)\psi(t)}\frac{g^2(t)}{2} \quad (48)$$

$$= \frac{\sigma(t)}{\sigma'(t) - b(t)\sigma(t)}\frac{g^2(t)}{2} = \frac{\sigma(t)}{\dfrac{2b(t)\sigma^2(t) + g^2(t)}{2\sigma(t)} - b(t)\sigma(t)}\frac{g^2(t)}{2} \quad (49)$$

$$= \frac{2\sigma^2(t)}{2b(t)\sigma^2(t) + g^2(t) - 2b(t)\sigma^2(t)}\frac{g^2(t)}{2} = \sigma^2(t), \quad (50)$$

which, as previously mentioned, shows that Equation 43 simplifies to Equation 13.

## B.3 Proof of Theorem 3.1

We begin by stating and proving a lemma which we will later use.

**Lemma B.1.** *For any $\epsilon > 0$ and $\xi > 0$, there exists $\Delta < 0$ such that for all $\delta < \Delta$ and $y \in \mathbb{R}^d$ with $\|y\|_2 > \xi$, it holds that:*

$$\mathcal{N}(y; 0, e^{2\delta} I_d) < \epsilon e^{2\delta}. \tag{51}$$

*Proof.* The inequality holds if and only if

$$-\frac{d}{2}\log(2\pi) - d\delta - \frac{e^{-2\delta}\|y\|_2^2}{2} < \log \epsilon + 2\delta, \tag{52}$$

which in turn is equivalent to

$$\|y\|_2^2 > 2\left(-2\delta - \log\epsilon - \left(\delta + \frac{\log(2\pi)}{2}\right)d\right)e^{2\delta}. \tag{53}$$

The limit of the right hand side as $\delta \to -\infty$ is 0 (and it approaches from the positive side), while $\|y\|_2^2$ is lower bounded by $\xi^2$, thus finishing the proof. $\square$

We now restate Theorem 3.1 for convenience:

**Theorem 3.1** (FLIPD Soundness: Linear Case). *Let $\mathcal{L}$ be an embedded submanifold of $\mathbb{R}^D$ given by a $d$-dimensional affine subspace. If $p(\cdot, 0)$ is supported on $\mathcal{L}$, continuous, and with finite second moments, then for any $x \in \mathcal{L}$ with $p(x, 0) > 0$, we have:*

$$\lim_{\delta \to -\infty} \frac{\partial}{\partial \delta} \log \varrho(x, \delta) = d - D. \tag{16}$$

*Proof.* As the result is invariant to rotations and translations, we assume without loss of generality that $\mathcal{L} = \{(x', 0) \in \mathbb{R}^D \mid x' \in \mathbb{R}^d, 0 \in \mathbb{R}^{D-d}\}$. Since $x \in \mathcal{L}$, it has the form $x = (x', 0)$ for some $x' \in \mathbb{R}^d$. Note that formally $p(\cdot, 0)$ is not a density with respect to the $D$-dimensional Lebesgue measure, however, with a slight abuse of notation, we identify it with $p(x')$, where $p(\cdot)$ is now the $d$-dimensional Lebesgue density of $X'$, where $(X', 0) = X \sim p(\cdot, 0)$. We will denote $p(\cdot) \circledast \mathcal{N}(\cdot; 0, e^{2\delta} I_d)$ as $p_\delta(\cdot)$. In this simplified notation, our assumptions are that $p(\cdot)$ is continuous at $x'$ with finite second moments, and that $x'$ is such that $p(x') > 0$.

For ease of notation we use $\mathcal{N}_d^\delta$ to represent the normal distribution with variance $e^{2\delta}$ on a $d$-dimensional space, i.e. $\mathcal{N}_d^\delta(\cdot) = \mathcal{N}(\cdot; 0, e^{2\delta} I_d)$. For any subspace $S$ we use $S^c$ to denote its complement where the ambient space is clear from context. $B_\xi(0)$ denotes a ball of radius $\xi$ around the origin.

We start by noticing that the derivative of the logarithm of a Gaussian with respect to its log variance can be computed as:

$$\frac{\partial}{\partial \delta} \log \mathcal{N}_d^\delta(x') = \frac{\partial}{\partial \delta}\left(-\frac{d}{2}\log(2\pi) - d\delta - \frac{e^{-2\delta}}{2}\|x'\|_2^2\right) = -d + e^{-2\delta}\|x'\|_2^2. \tag{54}$$

We then have that:

$$\varrho(x, \delta) = p_\delta(x') \times \mathcal{N}_{D-d}^\delta(0) \tag{55}$$

$$\implies \log \varrho(x, \delta) = \log p_\delta(x') - \delta(D - d) + c_0 \tag{56}$$

$$\implies \frac{\partial}{\partial \delta} \log \varrho(x, \delta) = \frac{\partial}{\partial \delta} \log p_\delta(x') - (D - d), \tag{57}$$

where $c_0$ is a constant that does not depend on $\delta$. Thus, it suffices to show that $\lim_{\delta \to -\infty} \frac{\partial}{\partial \delta} \log p_\delta(x') = 0$:

$$\lim_{\delta \to -\infty} \frac{\partial}{\partial \delta} \log p_\delta(x') = \lim_{\delta \to -\infty} \frac{\frac{\partial}{\partial \delta} p_\delta(x')}{p_\delta(x')} \tag{58}$$

$$= \lim_{\delta \to -\infty} \frac{\frac{\partial}{\partial \delta} \int_{\mathbb{R}^d} p(x' - y) \mathcal{N}_d^\delta(y) \mathrm{d}y}{\int_{\mathbb{R}^d} p(x' - y) \mathcal{N}_d^\delta(y) \mathrm{d}y} \tag{59}$$

$$= \lim_{\delta \to -\infty} \frac{\int_{\mathbb{R}^d} p(x' - y) \frac{\partial}{\partial \delta} \mathcal{N}_d^\delta(y) \mathrm{d}y}{\int_{\mathbb{R}^d} p(x - y) \mathcal{N}_d^\delta(y) \mathrm{d}y} \tag{60}$$

$$= \lim_{\delta \to -\infty} \frac{\int_{\mathbb{R}^d} p(x' - y)(-d + e^{-2\delta} \|y\|_2^2) \mathcal{N}_d^\delta(y) \mathrm{d}y}{\int_{\mathbb{R}^d} p(x' - y) \mathcal{N}_d^\delta(y) \mathrm{d}y} \tag{61}$$

$$= -d + \lim_{\delta \to -\infty} \frac{e^{-2\delta} \int_{\mathbb{R}^d} p(x' - y) \|y\|_2^2 \mathcal{N}_d^\delta(y) \mathrm{d}y}{\int_{\mathbb{R}^d} p(x' - y) \mathcal{N}_d^\delta(y) \mathrm{d}y}, \tag{62}$$

where we exchanged the order of derivation and integration in Equation 60 using Leibniz integral rule (because the normal distribution, its derivative, and $p$ are continuous; note that $p$ does not depend on $\delta$ so regularity on its derivative is not necessary), and where Equation 61 follows from Equation 54. Thus, proving that

$$\lim_{\delta \to -\infty} \frac{e^{-2\delta} \int_{\mathbb{R}^d} p(x' - y) \|y\|_2^2 \mathcal{N}_d^\delta(y) \mathrm{d}y}{\int_{\mathbb{R}^d} p(x' - y) \mathcal{N}_d^\delta(y) \mathrm{d}y} = d \tag{63}$$

would finish our proof.

Now let $\epsilon > 0$. By continuity of $p$ at $x'$, there exists $\xi > 0$ such that if $\|y\|_2 < \xi$, then $|p(x' - y) - p(x')| < \epsilon$. Let $\Delta$ be the corresponding $\Delta$ from Lemma B.1. Assume $\delta < \Delta$ and define $c_1$ and $c_2$ as follows:

$$c_1 = \int_{B_\xi^c(0)} p(x' - y) \mathcal{N}_d^\delta(y) \mathrm{d}y, \quad \text{and} \quad c_2 = \int_{B_\xi^c(0)} p(x' - y) \|y\|_2^2 (\epsilon e^{2\delta})^{-1} \mathcal{N}_d^\delta(y) \mathrm{d}y. \tag{64}$$

From Lemma B.1 and $p$ having finite second moments it follows that $c_1 \in [0, 1]$ and that $c_2 \in \left[0, \int_{\mathbb{R}^d} \|y\|_2^2 p(x' - y) dy\right]$. We have:

$$\frac{e^{-2\delta} \int_{\mathbb{R}^d} p(x' - y) \|y\|_2^2 \mathcal{N}_d^\delta(y) \mathrm{d}y}{\int_{\mathbb{R}^d} p(x' - y) \mathcal{N}_d^\delta(y) \mathrm{d}y} = \frac{e^{-2\delta} \int_{B_\xi(0)} p(x' - y) \|y\|_2^2 \mathcal{N}_d^\delta(y) \mathrm{d}y + c_2 \epsilon}{\int_{B_\xi(0)} p(x' - y) \mathcal{N}_d^\delta(y) \mathrm{d}y + c_1 \epsilon} \tag{65}$$

$$= \frac{e^{-2\delta} p(x') \int_{B_\xi(0)} \|y\|_2^2 \mathcal{N}_d^\delta(y) \mathrm{d}y + e^{-2\delta} \int_{B_\xi(0)} (p(x' - y) - p(x')) \|y\|_2^2 \mathcal{N}_d^\delta(y) \mathrm{d}y + c_2 \epsilon}{p(x') \int_{B_\xi(0)} \mathcal{N}_d^\delta(y) \mathrm{d}y + \int_{B_\xi(0)} (p(x' - y) - p(x')) \mathcal{N}_d^\delta(y) \mathrm{d}y + c_1 \epsilon}. \tag{66}$$

Analoguously to $c_1$ and $c_2$, there exists $c_3 \in [-1, 1]$ and $c_4 \in [-d, d]$ so that the Equation 66 is equal to:

$$\frac{e^{-2\delta} \int_{B_\xi(0)} \|y\|_2^2 \mathcal{N}_d^\delta(y) \mathrm{d}y + \frac{(c_2 + c_4)\epsilon}{p(x')}}{\int_{B_\xi(0)} \mathcal{N}_d^\delta(y) \mathrm{d}y + \frac{(c_1 + c_3)\epsilon}{p(x')}} = \frac{d - e^{-2\delta} \int_{B_\xi^c(0)} \|y\|_2^2 \mathcal{N}_d^\delta(y) \mathrm{d}y + \frac{(c_2 + c_4)\epsilon}{p(x')}}{1 - \int_{B_\xi^c(0)} \mathcal{N}_d^\delta(y) \mathrm{d}y + \frac{(c_1 + c_3)\epsilon}{p(x')}} =: I. \tag{67}$$

We still need to prove that $\lim_{\delta \to -\infty} I = d$. Taking $\limsup$ and $\liminf$ as $\delta \to -\infty$ yields:

$$\frac{d + \frac{(c_2' + c_4')\epsilon}{p(x')}}{1 + \frac{(c_1'' + c_3'')\epsilon}{p(x')}} \le \liminf_{\delta \to -\infty} I \le \limsup_{\delta \to -\infty} I \le \frac{d + \frac{(c_2'' + c_4'')\epsilon}{p(x')}}{1 + \frac{(c_1' + c_3')\epsilon}{p(x')}}, \tag{68}$$

where $c_i'' = \limsup_{\delta \to -\infty} c_i$ and $c_i' = \liminf_{\delta \to -\infty} c_i$. Note that although the values of $c_i'$ and $c_i''$ depend on $\epsilon$ and $\xi$, the bounds on them do not. We can thus take the limit of this inequality as $\xi$ and $\epsilon$ approach zero:

$$d \le \lim_{\epsilon, \xi \to 0} \liminf_{\delta \to -\infty} I \le \lim_{\epsilon, \xi \to 0} \limsup_{\delta \to -\infty} I \le d. \tag{69}$$

However, note that every step up to here has been an equality, therefore

$$I = \frac{e^{-2\delta} \int_{\mathbb{R}^d} p(x' - y) \|y\|_2^2 \, \mathcal{N}_d^\delta(y) \mathrm{d}y}{\int_{\mathbb{R}^d} p(x' - y) \mathcal{N}_d^\delta(y) \mathrm{d}y}, \tag{70}$$

so that $I$ does not depend on $\epsilon$ nor on $\xi$. In turn, this implies that

$$d \le \liminf_{\delta \to -\infty} I \le \limsup_{\delta \to -\infty} I \le d \implies \lim_{\delta \to -\infty} I = d, \tag{71}$$

which finishes the proof. $\square$

Table 4: Comparing the discretized DDPM notation with score-based DM side-by-side.

| Term | DDPM [25] | Score-based DM [58] |
|---|---|---|
| Timestep | $t \in \{0, 1, \ldots, T\}$ | $t/T = t \in [0, 1]$ |
| (Noised out) datapoint | $x_t$ | $x_{t/T} = x_t$ |
| Diffusion process hyperparameter | $\beta_t$ | $\beta(t/T) = \beta(t)$ |
| Mean of transition kernel | $\sqrt{\bar{\alpha}_t}$ | $\psi(t/T) = \psi(t)$ |
| Std of transition kernel | $\sqrt{1 - \bar{\alpha}_t}$ | $\sigma(t/T) = \sigma(t)$ |
| Network parameterization | $-\epsilon(x, t)/\sqrt{1 - \bar{\alpha}_t}$ | $\hat{s}(x, t/T) = \hat{s}(x, t)$ |

## C  Adapting FLIPD for DDPMs

Here, we adapt FLIPD for state-of-the-art DDPM architectures and follow the discretized notation from Ho et al. [25] where instead of using a continuous time index $t$ from 0 to 1, a timestep $t$ belongs instead to the sequence $\{0, \ldots, T\}$ with $T$ being the largest timescale. We use the colour gold to indicate the notation used by Ho et al. [25]. We highlight that the content of this section is a summary of the equivalence between DDPMs and the score-based formulation established by Song et al. [58].

As a reminder, DDPMs can be viewed as discretizations of the *forward* SDE process of a DM, where the process turns into a Markov *noising* process:

$$p(x_t \mid x_{t-1}) := \mathcal{N}(x_t; \sqrt{1 - \beta_t} \cdot x_{t-1}, \beta_t I_D). \tag{72}$$

We also use sub-indices $t$ instead of functions evaluated at $t$ to keep consistent with Ho et al. [25]'s notation. This in turn implies the following transition kernel:

$$p(x_t \mid x_0) = \mathcal{N}(x_t; \sqrt{\bar{\alpha}_t} x_0, (1 - \bar{\alpha}_t) I_D) \tag{73}$$

where $\alpha_t := 1 - \beta_t$ and $\bar{\alpha}_t := \prod_{s=1}^{t} \alpha_t$.

DDPMs model the *backward* diffusion process (or *denoising* process) by modelling a network $\epsilon : \mathbb{R}^D \times \{1, \ldots, T\} \to \mathbb{R}^D$ that takes in a noised-out point and outputs a residual that can be used to denoise. Song et al. [58] show that one can draw an equivalence between the network $\epsilon(\cdot, t)$ and the score network (see their Appendix B). Here, we rephrase the connections in a more explicit manner where we note that:

$$-\epsilon(x, t)/\sqrt{1 - \bar{\alpha}_t} = s(x, t/T). \tag{74}$$

Consequently, plugging into Equation 22, we get the following formula adapted for DDPMs:

$$\text{FLIPD}(x, t_0) = D - \sqrt{1 - \bar{\alpha}_{t_0}} \, \text{tr} \left( \nabla \epsilon(\sqrt{\bar{\alpha}_{t_0}} x, t_0) \right) + \| \epsilon(\sqrt{\bar{\alpha}_{t_0}} x, t_0) \|_2^2, \tag{75}$$

where $t_0 = t_0 \times T$ (best viewed in colour). We include Table 4, which summarizes all of the equivalent notation when moving from DDPMs to score-based DMs and vice-versa.

Table 5: Essential hyperparameter settings for the Diffusion models with MLP backbone.

| Property | Model Configuration |
|---|---|
| Learning rate | $10^{-4}$ |
| Optimizer | AdamW |
| Scheduler | Cosine scheduling with 500 warmup steps [68] |
| Epochs | 200, 400, 800, or 1000 based on the ambient dimension |
| Score-matching loss | Likelihood weighting [57] |
| SDE drift | $f(x,t) := -\frac{1}{2}\beta(t)x$ |
| SDE diffusion | $g(t) := \sqrt{\beta(t)}$ |
| $\beta(t)$ | Linear interpolation: $\beta(t) := 0.1 + 20t$ |
| Score network | MLP |
| MLP hidden sizes | $\langle 4096, 2048, 2 \times 1024, 3 \times 512, 2 \times 1024, 2048, 4096 \rangle$ |
| Time embedding size | 128 |

# D  Experimental Details

Throughout all our experiments, we used an NVIDIA A100 GPU with 40GB of memory.

## D.1  DM Hyperparameter Setup

To mimic a UNet, we use an MLP architecture with a bottleneck as our score network for our synthetic experiments. This network contains $2 \times L + 1$ fully connected layers with dimensions $\langle h_1, h_2, \ldots, h_L, \ldots h_{2L+1} \rangle$ forming a bottleneck, i.e., $h_L$ has the smallest size. Notably, for $1 \leq i \leq L$, the $i$th transform connects layer $i-1$ (or the input) to layer $i$ with a linear transform of dimensions "$h_{i-1} \times h_i$", and the $(L+i)$th layer (or the output) not only contains input from the $(L+i-1)$th layer but also, contains skip connections from layer $(L-i)$ (or the input), thus forming a linear transform of dimension "$(h_{L+i-1} + h_{L-i}) \times h_{L+i}$". For image experiments, we scale and shift the pixel intensities to be zero-centered with a standard deviation of 1. In addition, we embed times $t \in (0,1)$ using the scheme in [68] and concatenate with the input before passing to the score network. All hyperparameters are summarized in Table 5.

## D.2  FLIPD Estimates and Curves for Synthetic Distributions

Figure 5 shows pointwise LID estimates for a "lollipop" distribution taken from [62]. It is a uniform distribution over three submanifolds: $(i)$ a 2d "candy", $(ii)$ a 1d "stick", and $(iii)$ an isolated point of zero dimensions. Note that the FLIPD estimates at $t_0 = 0.05$ for all three submanifolds are coherent.

Figure 6 shows the FLIPD curve as training progresses on the lollipop example and the Gaussian mixture that was already discussed in Section 4. We see that gradually, knee patterns emerge at the correct LID, indicating that the DM is learning the data manifold. Notably, data with higher LID values get assigned higher estimates *even after a few epochs*, demonstrating that FLIPD effectively ranks data based on LID, even when the DM is underfitted.

Finally, Figure 7 presents a summary of complex manifolds obtained from neural spline flows and high-dimensional mixtures, showing knees around the true LID.

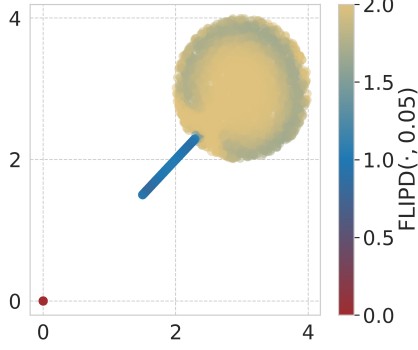

Figure 5: The FLIPD estimates on a Lollipop dataset from [62].

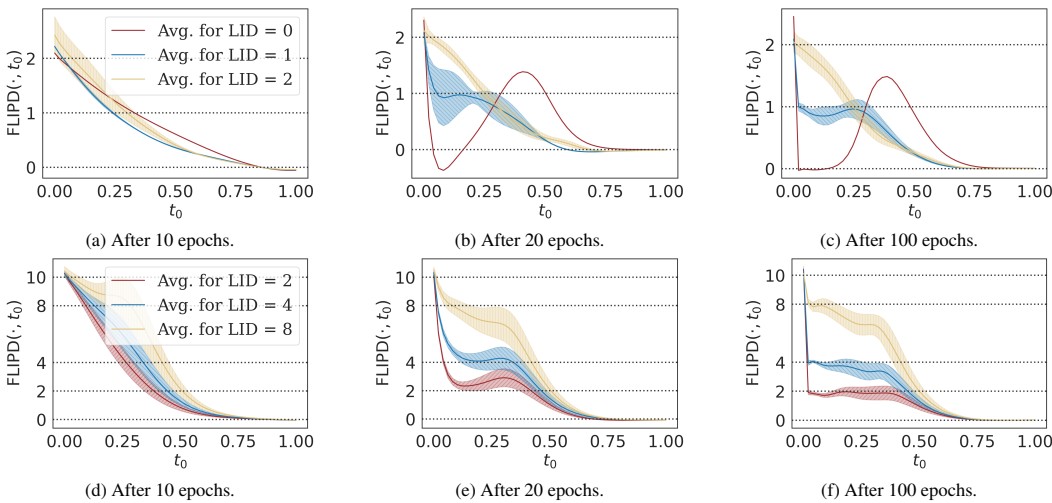

Figure 6: The evolution of the FLIPD curve while training the DM to fit a Lollipop (top) and a manifold mixture $\mathcal{N}_2 + \mathcal{N}_4 + \mathcal{N}_8 \subseteq \mathbb{R}^{10}$ (bottom).

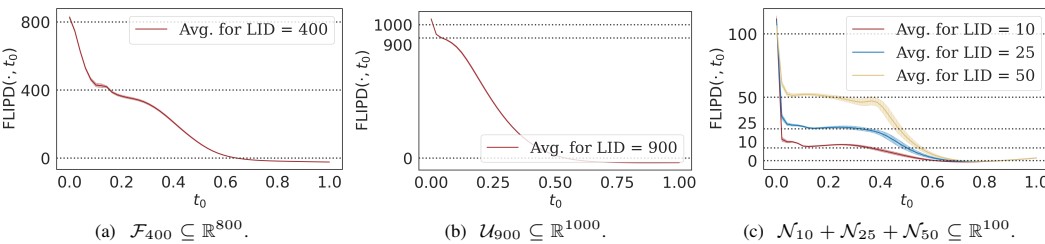

Figure 7: The FLIPD curve for complex and high-dimensional manifolds.

## D.3 A Simple Multiscale Experiment

Tempczyk et al. [62] argue that when setting $\delta$, all the directions of data variation that have a log standard deviation below $\delta$ are ignored. Here, we make this connection more explicit.

We define a multivariate Gaussian distribution with a prespecified eigenspectrum for its covariance matrix: having three eigenvalues of $10^{-4}$, three eigenvalues of $1$, and four eigenvalues of $10^3$. This ensures that the distribution is numerically 7d and that the directions and amount of data variation are controlled using the eigenvectors and eigenvalues of the covariance matrix.

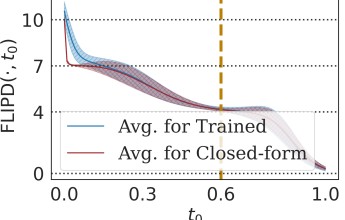

Figure 8: FLIPD curve for a multivariate Gaussian with controlled covariance eigenspectrum.

For this multivariate Gaussian, the score function in Equation 13 can be written in closed form; thus, we evaluate FLIPD both with and without training a DM. We see in Figure 8 the estimates obtained in both scenarios match closely, with some deviations due to imperfect model fit, which we found matches perfectly when training for longer.

Apart from the initial knee at 7, which is expected, we find another at $t_0 = 0.6$ (corresponding to $e^\delta \approx 6.178$ with our hyperparameter setup in Table 5) where the value of FLIPD is 4. This indeed confirms that the estimator focuses solely on the 4d space characterized by the eigenvectors having eigenvalues of $10^3$, and by ignoring the eigenvalues $10^{-4}$ and $1$ which are both smaller than $6.178$.

## D.4 In-depth Analysis of the Synthetic Benchmark

**Generating Manifold Mixtures**  To create synthetic data, we generate each component of our manifold mixture separately and then join them to form a distribution. All mixture components are sampled with equal probability in the final distribution. For a component with the intrinsic dimension

of $d$, we first sample from a base distribution in $d$ dimensions. This base distribution is isotropic Gaussian and Laplace for $\mathcal{N}_d$ and $\mathcal{L}_d$ and uniform for the case of $\mathcal{U}_d$ and $\mathcal{F}_d$. We then zero-pad these samples to match the ambient dimension $D$ and perform a random $D \times D$ rotation on $\mathbb{R}^D$ (each component has one such transformation). For $\mathcal{F}_d$, we have an additional step to make the submanifold complex. We first initialize a neural spline flow (using the `nflows` library [19]) with 5 coupling transforms, 32 hidden layers, 32 hidden blocks, and tail bounds of 10. The data is then passed through the flow, resulting in a complex manifold embedded in $\mathbb{R}^D$ with an LID of $d$. Finally, we standardize each mixture component individually and set their modes such that the pairwise Euclidean distance between them is at least 20. We then translate each component so its barycenter matches the corresponding mode, ensuring that data from different components rarely mixes, thus maintaining distinct LIDs.

**LIDL Baseline** Following the hyperparameter setup in [62], we train 8 models with different noised-out versions of the dataset with standard deviations $e^{\delta_i} \in \{0.01, 0.014, 0.019, 0.027, 0.037, 0.052, 0.072, 0.1\}$. The normalizing flow backbone is taken from [19], using 10 piecewise rational quadratic transforms with 32 hidden dimensions, 32 blocks, and a tail bound of 10. While [62] uses an autoregressive flow architecture, we use coupling transforms for increased training efficiency and to match the training time of a single DM.

**Setup** We use model-free estimators from the `skdim` library [5] and across our experiments, we sample $10^6$ points from the synthetic distributions for either fitting generative models or fitting the model-free estimators. We then evaluate LID estimates on a uniformly subsampled set of $2^{12}$ points from the original set of $10^6$ points. Some methods are relatively slow, and this allows us to have a fair, yet feasible comparison. We do not see a significant difference even when we double the size of the subsampled set. We focus on four different model-free baselines for our evaluation: ESS [30], LPCA [22, 12], MLE [38, 43], and FIS [1]. We use default settings throughout, except for high-dimensional cases ($D > 100$), where we set the number of nearest neighbours $k$ to 1000 instead of the default 100. This adjustment is necessary because we observed a significant performance drop, particularly in LPCA, when $k$ was too small in these scenarios. We note that this adjustment has not been previously implemented in similar benchmarks conducted by Tempczyk et al. [62], despite their claim that their method outperforms model-free alternatives. We also note that FIS does not scale beyond 100 dimensions, and thus leave it blank in our reports. Computing pairwise distances on high dimensions ($D \geq 800$) on all $10^6$ samples takes over 24 hours even with 40 CPU cores. Therefore, for $D \geq 800$, we use the same $2^{12}$ subsamples we use for evaluation.

**Evaluation** We have three tables to summarize our analysis: $(i)$ Table 7 shows the MAE of the LID estimates, comparing each datapoint's estimate to the ground truth at a fine-grained level; $(ii)$ Table 9 shows the average LID estimate for synthetic manifolds with only one component, this average is typically used to estimate *global* intrinsic dimensionality in baselines; finally, $(iii)$ Table 8 looks at the concordance index [23] of estimates for cases with multiple submanifolds of different dimensionalities. Concordance indices for a sequence of estimates $\{\widehat{\text{LID}}\}_{n=1}^N$ are defined as follows:

$$\mathcal{C}\left(\{\text{LID}_n\}_{n=1}^N, \{\widehat{\text{LID}}_n\}_{n=1}^N\right) = \sum_{\substack{1 \leq n_1 \neq n_2 \leq N \\ \text{LID}_{n_1} \leq \text{LID}_{n_2}}} \mathbb{I}(\widehat{\text{LID}}_{n_1} \leq \widehat{\text{LID}}_{n_2}) / \binom{N}{2}, \qquad (76)$$

where $\mathbb{I}$ is the indicator function; a perfect estimator will have a $\mathcal{C}$ of 1. Instead of emphasizing the actual values of the LID estimates, this metric assesses how well the ranks of an estimator align with those of ground truth [60, 45, 61, 31], thus evaluating LID as a "relative" measure of complexity.

**Model-free Analysis** Among model-free methods, LPCA and ESS show good performance in low dimensions, with LPCA being exceptionally good in scenarios where the manifold is affine. As we see in Table 7, while model-free methods produce reliable estimates when $D$ is small, as $D$ increases the estimates become more unreliable. In addition, we include average LID estimates in Table 9 and see that all model-free baselines underestimate intrinsic dimensionality to some degree, with LPCA and MLE being particularly drastic when $D > 100$. We note that ESS performs relatively well, even beating model-based methods in some 800-dimensional scenarios. However, we note that both the $\mathcal{C}$ values in Table 8 and MAE values in Tables 7 and 9 suggest that none of these estimators effectively estimate LID in a pointwise manner and cannot rank data by LID as effectively as FLIPD.

Table 6: MAE (lower is better). Rows show synthetic manifolds and columns represent different variations of our Fokker-Planck-based estimators.

| Synthetic Manifold | FLIPD $t_0 = .05$ | FLIPD `kneedle` | FPRegress `kneedle` | FPRegress $\delta_1 = -1$ |
|---|---|---|---|---|
| Lollipop in $\mathbb{R}^2$ | **0.142** | 0.419 | 0.572 | 0.162 |
| String within doughnut $\mathbb{R}^3$ | **0.052** | 0.055 | 0.398 | 0.377 |
| Swiss Roll in $\mathbb{R}^3$ | **0.053** | 0.055 | 0.087 | 0.161 |
| $\mathcal{L}_5 \subseteq \mathbb{R}^{10}$ | **0.100** | 0.169 | 1.168 | 0.455 |
| $\mathcal{N}_{90} \subseteq \mathbb{R}^{100}$ | 0.501 | **0.492** | 3.142 | 0.998 |
| $\mathcal{U}_{10} + \mathcal{U}_{30} + \mathcal{U}_{90} \subseteq \mathbb{R}^{100}$ | 3.140 | **1.298** | 5.608 | 10.617 |
| $\mathcal{F}_{10} + \mathcal{F}_{25} + \mathcal{F}_{50} \subseteq \mathbb{R}^{100}$ | 14.37 | **3.925** | 16.32 | 21.01 |
| $\mathcal{U}_{10} + \mathcal{U}_{80} + \mathcal{U}_{200} \subseteq \mathbb{R}^{800}$ | 39.54 | **14.30** | 30.06 | 29.99 |

**Model-based Analysis** Focusing on model-based methods, we see in Table 8 that all except FLIPD perform poorly in ranking data based on LID. Remarkably, FLIPD achieves perfect $\mathcal{C}$ values among *all* datasets; further justifying it as a relative measure of complexity. We also note that while LIDL and NB provide better global estimates in Table 9 for high dimensions, they have worse MAE performance in Table 7. This once again suggests that at a local level, our estimator is superior compared to others, beating all baselines in 4 out of 5 groups of synthetic manifolds in Table 7.

### D.5 Ablations

We begin by evaluating the impact of using `kneedle`. Our findings, summarized in Table 6, indicate that while setting a small fixed $t_0$ is effective in low dimensions, the advantage of `kneedle` becomes particularly evident as the number of dimensions increases.

We also tried combining it with `kneedle` by sweeping over the origin $\delta_1$ and arguing that the estimates obtained from this method also exhibit knees. Despite some improvement in high-dimensional settings, Table 6 shows that even coupling it with `kneedle` does not help.

### D.6 Improving the NB Estimators with `kneedle`

We recall that the NB estimator requires computing $\text{rank} \, S(x)$, where $S(x)$ is a $K \times D$ matrix formed by stacking the scores $\hat{s}(\cdot, t_0)$. We set $t_0 = 0.01$ as it provides the most reasonable estimates. To compute $\text{rank} \, S(x)$ numerically, Stanczuk et al. [59] perform a singular value decomposition on $S(x)$ and use a cutoff threshold $\tau$ below which singular values are considered zero. Finding the best $\tau$ is challenging, so Stanczuk et al. [59] propose finding the two consecutive singular values with the maximum gap. Furthermore, we see that sometimes the top few singular values are disproportionately higher than the rest, resulting in severe overestimations of the LID. Thus, we introduce an alternative algorithm to determine the optimal $\tau$. For each $\tau$, we estimate LID by thresholding the singular values. Sweeping 100 different $\tau$ values from 0 to 1000 at a geometric scale (to further emphasize smaller thresholds) produces estimates ranging from $D$ (keeping all singular values) to 0 (ignoring all). As $\tau$ varies, we see that the estimates plateau over a certain range of $\tau$. We use `kneedle` to detect this plateau because the starting point of a plateau is indeed a knee in the curve. This significantly improves the baseline, especially in high dimensions: see the third column of Tables 7, 9, and 8 compared to the second column.

Table 7: MAE (lower is better). Each row represents a synthetic dataset and each column represents an LID estimation method. Rows are split into groups based on the ambient dimension: the first group of rows shows toy examples; the second shows low-dimensional data with $D = 10$; the third shows moderate-dimensional data with $D = 100$; and the last two show high-dimensional data with $D = 800$ and $D = 1000$, respectively.

| Synthetic Manifold | Model-based | | | | Model-free | | | |
|---|---|---|---|---|---|---|---|---|
| | FLIPD kneedle | NB Vanilla | NB kneedle | LIDL | ESS | LPCA | MLE | FIS |
| Lollipop $\subseteq \mathbb{R}^2$ | 0.419 | 0.577 | 0.855 | **0.052** | 0.009 | **0.000** | 0.142 | 0.094 |
| Swiss Roll $\subseteq \mathbb{R}^3$ | 0.055 | 0.998 | **0.016** | 0.532 | 0.017 | **0.000** | 0.165 | 0.018 |
| String within doughnut $\subseteq \mathbb{R}^3$ | **0.055** | 1.475 | 0.414 | 1.104 | 0.017 | **0.000** | 0.128 | 0.041 |
| Summary (Toy Manifolds) | **0.176** | 1.017 | 0.428 | 0.563 | 0.014 | **0.000** | 0.145 | 0.051 |
| $\mathcal{N}_5 \subseteq \mathbb{R}^{10}$ | 0.084 | 5.000 | **0.005** | 0.071 | 0.061 | **0.000** | 0.441 | 0.206 |
| $\mathcal{L}_5 \subseteq \mathbb{R}^{10}$ | 0.169 | 1.000 | 0.146 | **0.101** | 0.068 | **0.000** | 0.462 | 0.203 |
| $\mathcal{U}_5 \subseteq \mathbb{R}^{10}$ | 0.324 | 4.994 | 0.933 | **0.123** | 0.153 | **0.000** | 0.451 | 0.186 |
| $\mathcal{F}_5 \subseteq \mathbb{R}^{10}$ | 0.666 | 1.216 | 0.765 | **0.487** | 0.168 | **0.000** | 0.497 | 0.176 |
| $\mathcal{N}_2 + \mathcal{N}_4 + \mathcal{N}_8 \subseteq \mathbb{R}^{10}$ | **0.287** | 5.706 | 1.078 | 0.308 | 0.156 | **0.000** | 0.406 | 0.206 |
| $\mathcal{L}_2 + \mathcal{L}_4 + \mathcal{L}_8 \subseteq \mathbb{R}^{10}$ | **0.253** | 5.708 | 0.772 | 0.515 | 0.193 | **0.001** | 0.437 | 0.018 |
| $\mathcal{U}_2 + \mathcal{U}_4 + \mathcal{U}_8 \subseteq \mathbb{R}^{10}$ | 0.586 | 5.677 | 3.685 | **0.363** | 0.331 | **0.115** | 0.540 | 0.222 |
| $\mathcal{F}_2 + \mathcal{F}_4 + \mathcal{F}_8 \subseteq \mathbb{R}^{10}$ | **0.622** | 5.709 | 2.187 | 1.013 | 0.428 | **0.115** | 0.642 | 0.269 |
| Summary (10-dimensional) | **0.066** | 0.381 | 0.161 | 0.211 | 0.005 | **0.000** | 0.054 | 0.019 |
| $\mathcal{U}_{10} \subseteq \mathbb{R}^{100}$ | 0.910 | 30.115 | **0.000** | 1.370 | 0.644 | **0.000** | 1.263 | – |
| $\mathcal{U}_{30} \subseteq \mathbb{R}^{100}$ | 0.505 | 50.521 | **0.000** | 0.542 | 1.465 | **0.002** | 7.622 | – |
| $\mathcal{U}_{90} \subseteq \mathbb{R}^{100}$ | 0.640 | 1.157 | 1.327 | **0.332** | **2.034** | 21.90 | 39.65 | – |
| $\mathcal{N}_{30} \subseteq \mathbb{R}^{100}$ | 0.887 | 52.43 | **0.000** | 0.892 | 0.534 | **0.000** | 5.703 | – |
| $\mathcal{N}_{90} \subseteq \mathbb{R}^{100}$ | 0.492 | **0.184** | 2.693 | 0.329 | **1.673** | 21.88 | 39.45 | – |
| $\mathcal{F}_{80} \subseteq \mathbb{R}^{100}$ | **1.869** | 20.00 | 3.441 | 1.871 | **3.660** | 16.78 | 34.41 | – |
| $\mathcal{U}_{10} + \mathcal{U}_{25} + \mathcal{U}_{50} \subseteq \mathbb{R}^{100}$ | **0.868** | 57.96 | 0.890 | 5.869 | **4.988** | 6.749 | 16.12 | – |
| $\mathcal{U}_{10} + \mathcal{U}_{30} + \mathcal{U}_{90} \subseteq \mathbb{R}^{100}$ | **1.298** | 61.58 | 1.482 | 8.460 | 21.89 | **20.06** | 41.05 | – |
| $\mathcal{N}_{10} + \mathcal{N}_{25} + \mathcal{N}_{50} \subseteq \mathbb{R}^{100}$ | 1.813 | 74.20 | **0.555** | 8.873 | 7.712 | **5.716** | 14.37 | – |
| $\mathcal{F}_{10} + \mathcal{F}_{25} + \mathcal{F}_{50} \subseteq \mathbb{R}^{100}$ | **3.925** | 74.20 | 6.205 | 18.61 | 9.200 | **6.769** | 16.78 | – |
| Summary (100-dimensional) | **1.321** | 42.24 | 1.659 | 4.715 | **5.380** | 9.986 | 21.64 | – |
| $\mathcal{U}_{200} \subseteq \mathbb{R}^{800}$ | 11.54 | 600.0 | **7.205** | 55.98 | 3.184 | **0.000** | 159.3 | – |
| $\mathcal{F}_{400} \subseteq \mathbb{R}^{800}$ | 20.46 | 400.0 | **10.15** | 207.2 | 14.48 | **2.919** | 342.6 | – |
| $\mathcal{U}_{10} + \mathcal{U}_{80} + \mathcal{U}_{200} \subseteq \mathbb{R}^{800}$ | **14.30** | 715.3 | 18.82 | 120.7 | 1.385 | **0.004** | 72.25 | – |
| Summary (800-dimensional) | 15.43 | 571.8 | **12.06** | 128.0 | 6.350 | **0.974** | 191.4 | - |
| $\mathcal{N}_{900} \subseteq \mathbb{R}^{1000}$ | **3.913** | 100.0 | 24.38 | 10.45 | **14.16** | 219.8 | 819.2 | – |
| $\mathcal{U}_{100} \subseteq \mathbb{R}^{1000}$ | 12.81 | 900.0 | 62.68 | **12.65** | 1.623 | **0.000** | 72.06 | – |
| $\mathcal{U}_{900} \subseteq \mathbb{R}^{1000}$ | 12.81 | 100.0 | **0.104** | 24.90 | **14.49** | 219.1 | 810.2 | – |
| $\mathcal{F}_{500} \subseteq \mathbb{R}^{1000}$ | **21.77** | 500.0 | 52.19 | 341.3 | **18.83** | 29.99 | 435.7 | – |
| Summary (1000-dimensional) | **12.83** | 400.0 | 34.84 | 97.33 | **12.28** | 117.2 | 534.3 | – |

Table 8: Concordance index (higher is better with 1.000 being the gold standard). Each row represents a mixture of multi-dimensional manifolds, and each column represents an LID estimation method. This table evaluates how accurately different estimators rank datapoints based on their LID.

| Synthetic Manifold | FLIPD kneedle | NB Vanilla | NB kneedle | LIDL | ESS | LPCA | MLE | FIS |
|---|---|---|---|---|---|---|---|---|
| Lollipop $\subseteq \mathbb{R}^2$ | **1.000** | 0.426 | 0.394 | 0.999 | **1.000** | **1.000** | **1.000** | **1.000** |
| String withing doughnut $\subseteq \mathbb{R}^3$ | **1.000** | 0.483 | 0.486 | 0.565 | **1.000** | **1.000** | **1.000** | **1.000** |
| $\mathcal{N}_2 + \mathcal{N}_4 + \mathcal{N}_8 \subseteq \mathbb{R}^{10}$ | **1.000** | 0.341 | 0.725 | 0.943 | **1.000** | **1.000** | **1.000** | **1.000** |
| $\mathcal{L}_2 + \mathcal{L}_4 + \mathcal{L}_8 \subseteq \mathbb{R}^{10}$ | **1.000** | 0.342 | 0.752 | 0.884 | **1.000** | **1.000** | **1.000** | **1.000** |
| $\mathcal{U}_2 + \mathcal{U}_4 + \mathcal{U}_8 \subseteq \mathbb{R}^{10}$ | **1.000** | 0.334 | 0.462 | 0.903 | **1.000** | **1.000** | **1.000** | **1.000** |
| $\mathcal{F}_2 + \mathcal{F}_4 + \mathcal{F}_8 \subseteq \mathbb{R}^{10}$ | **1.000** | 0.342 | 0.578 | 0.867 | **1.000** | **1.000** | 0.999 | **1.000** |
| $\mathcal{U}_{10} + \mathcal{U}_{25} + \mathcal{U}_{50} \subseteq \mathbb{R}^{100}$ | **1.000** | 0.467 | 0.879 | 0.759 | 0.855 | 0.897 | **1.000** | – |
| $\mathcal{U}_{10} + \mathcal{U}_{30} + \mathcal{U}_{90} \subseteq \mathbb{R}^{100}$ | **1.000** | 0.342 | 0.826 | 0.742 | 0.742 | 0.855 | **1.000** | – |
| $\mathcal{N}_{10} + \mathcal{N}_{25} + \mathcal{N}_{50} \subseteq \mathbb{R}^{100}$ | **1.000** | 0.342 | 0.866 | 0.736 | 0.878 | 0.917 | **1.000** | – |
| $\mathcal{F}_{10} + \mathcal{F}_{25} + \mathcal{F}_{50} \subseteq \mathbb{R}^{100}$ | **1.000** | 0.342 | 0.731 | 0.695 | 0.847 | 0.897 | **1.000** | – |
| $\mathcal{U}_{10} + \mathcal{U}_{80} + \mathcal{U}_{200} \subseteq \mathbb{R}^{800}$ | **1.000** | 0.342 | 0.841 | 0.697 | **1.000** | **1.000** | **1.000** | – |

Table 9: Analysis on manifolds with a single global intrinsic dimension. Columns, categorized by whether or not they use a model, display various LID estimators. The first set of rows shows the average LID as a *global* intrinsic dimension estimate and is grouped into lower- ($D \leq 100$) or higher-dimensional ($D > 100$) categories. Methods that most closely match the true intrinsic dimension are bolded. The second set of rows shows the pointwise precision of the estimators, as indicated by MAE (lower is better): while previous baselines generally provide reliable global estimates, their pointwise precision is subpar, especially with higher dimensionality.

| | Model-based | | | | Model-free | | | |
|---|---|---|---|---|---|---|---|---|
| | FLIPD kneedle | NB Vanilla | NB kneedle | LIDL | ESS | LPCA | MLE | FIS |
| Synthetic Manifold | Average LID Estimate | | | | | | | |
| Swiss Roll $\subseteq \mathbb{R}^3$ | **2.012** | 2.998 | 1.984 | 2.527 | 2.008 | **2.000** | 2.013 | 2.003 |
| $\mathcal{N}_5 \subseteq \mathbb{R}^{10}$ | 5.017 | 10.00 | **4.995** | 5.067 | 5.004 | **5.000** | 5.108 | 5.187 |
| $\mathcal{L}_5 \subseteq \mathbb{R}^{10}$ | **4.968** | 6.000 | 4.854 | 5.089 | 4.985 | **5.000** | 5.123 | 5.182 |
| $\mathcal{U}_5 \subseteq \mathbb{R}^{10}$ | 4.796 | 9.994 | 4.067 | **5.107** | 4.880 | **5.000** | 4.833 | 5.152 |
| $\mathcal{F}_5 \subseteq \mathbb{R}^{10}$ | **4.722** | 6.216 | 4.461 | 5.482 | 4.890 | **5.000** | 4.829 | 5.131 |
| $\mathcal{U}_{10} \subseteq \mathbb{R}^{100}$ | 11.68 | 40.12 | **10.00** | 11.29 | 9.361 | **10.00** | 8.905 | – |
| $\mathcal{U}_{30} \subseteq \mathbb{R}^{100}$ | 31.01 | 80.52 | **30.00** | 30.08 | 28.54 | **29.99** | 22.38 | – |
| $\mathcal{U}_{90} \subseteq \mathbb{R}^{100}$ | 89.54 | 91.16 | 91.32 | **90.24** | 88.27 | 68.10 | 50.35 | – |
| $\mathcal{N}_{30} \subseteq \mathbb{R}^{100}$ | 30.79 | 82.43 | **30.00** | 30.85 | 29.67 | **30.00** | 24.38 | – |
| $\mathcal{N}_{90} \subseteq \mathbb{R}^{100}$ | **89.88** | 90.18 | 92.62 | 90.21 | **88.89** | 68.12 | 50.55 | – |
| $\mathcal{F}_{80} \subseteq \mathbb{R}^{100}$ | 77.97 | 100.0 | 83.23 | **81.46** | **76.35** | 63.22 | 45.59 | – |
| $\mathcal{U}_{200} \subseteq \mathbb{R}^{800}$ | 211.5 | 800.0 | **207.2** | 256.0 | 199.3 | **200.0** | 40.73 | – |
| $\mathcal{F}_{400} \subseteq \mathbb{R}^{800}$ | 454.7 | 800.0 | **410.1** | 607.2 | 385.8 | **397.1** | 57.37 | – |
| $\mathcal{N}_{900} \subseteq \mathbb{R}^{1000}$ | **890.3** | 1000. | 924.4 | 924.4 | **897.3** | 680.2 | 80.78 | – |
| $\mathcal{U}_{100} \subseteq \mathbb{R}^{1000}$ | 135.76 | 1000. | 162.7 | **112.6** | 99.54 | **100.0** | 27.94 | – |
| $\mathcal{U}_{900} \subseteq \mathbb{R}^{1000}$ | 864.9 | 1000. | **900.1** | 911.9 | **899.3** | 680.9 | 89.80 | – |
| $\mathcal{F}_{500} \subseteq \mathbb{R}^{1000}$ | 582.7 | 1000. | **552.2** | 841.3 | **481.4** | 470.0 | 64.29 | – |
| Manifolds Summary | Pointwise MAE Averaged Across Tasks | | | | | | | |
| $\subseteq \mathbb{R}^D$ where $D \leq 100$ | 0.633 | 15.20 | 0.924 | **0.561** | 0.954 | **5.506** | 11.83 | – |
| $\subseteq \mathbb{R}^D$ where $D > 100$ | **13.88** | 433.3 | 26.12 | 108.7 | **11.13** | 78.64 | 439.8 | – |

# E Image Experiments

## E.1 FLIPD Curves and DM Samples: A Surprising Result

Figures 9 and 10 show DM-generated samples and the associated LID curves for 4096 samples from the datasets. In Figure 10, the FLIPD curves with MLPs have clearly discernible knees at which the estimated LID averages around a positive number, as expected. When changing the architecture of the DMs from MLPs to UNets, however, the curves either lack knees or obtain them at negative values of LID. This makes it harder to set $t_0$ automatically with `kneedle` when using UNets, and thus makes the estimator non-robust to the choice of architecture. Note that our theory ensures that $\text{FLIPD}(x, t_0)$ converges to $\text{LID}(x)$ as $t_0 \to 0$ when $\hat{s}$ is an accurate estimator of the true score function; indeed, it does not make any claims about the behaviour of FLIPD when $t_0$ is large or when the model fit is poor. Yet, the fact that the FLIPD estimates obtained from UNets are less reliable than the ones from MLPs is somewhat surprising, especially given that DMs with MLP backbones clearly produce worse-looking samples (shown in Figure 9), suggesting that while the model fit is better in the former case, the LID estimates are worse. Throughout the paper, we have argued that, notwithstanding this, FLIPD estimates derived from both architectures offer useful measurements of relative complexity, and can still effectively rank data based on its complexity; refer to all the images in Figures 17, 18, 19, 20, 21, 22, 23, and 24. However, this discrepancy between FLIPD estimates – in absolute terms – when altering the model architecture is counterintuitive and requires further treatment. In what follows, we propose hypotheses for the significant variation in FLIPD estimates when transitioning from MLP to UNet architectures, and we encourage further studies to investigate this issue and to adjust FLIPD to better suit DMs with state-of-the-art backbones.

While MLP architectures do not produce visually pleasing samples, comparing the first and second rows of Figure 9 side-by-side, it also becomes clear that MLP-generated images still match some characteristics of the true dataset, suggesting that they may still be capturing the image manifold in useful ways. To explain the unstable FLIPD estimates of UNets, we hypothesize that the convolutional layers in the UNet provide some inductive biases which, while helpful to produce visually pleasing images, might also encourage the network to over-fixate on high-frequency features. More specifically, score functions can be interpreted as providing a denoiser which removes noise added during the diffusion process; our hypothesis is that UNets are particularly good at removing noise along high-frequency directions but not necessarily along low-frequency ones. Inductive biases or modelling errors which make the learned score function $\hat{s}$ deviate from $s$ might significantly alter the behaviour of FLIPD estimates, even if they result in visually pleasing images; for a more formal treatment of the effect of this deviation, please refer to the work of Tempczyk et al. [63], who link modelling errors to LID estimation errors in LIDL.

The aforementioned inductive biases have been found to produce unintuitive behaviours in other settings as well. For example, Kirichenko et al. [34] showed how normalizing flows over-fixate on these high-frequency features, and how this can cause them to assign large likelihoods to out-of-distribution data, even when the model produces visually convincing samples. We hypothesize that a similar underlying phenomenon might be at play here, and that DMs with UNets might be over-emphasizing high-frequency directions of variation in their LID estimates. On the other hand, MLPs do not inherently distinguish between the high- and low-frequency directions and can therefore more reliably estimate the manifold dimension, even though their generated samples appear less appealing to the human eye. However, an exploration of this hypothesis is outside the scope of our work, and we highlight once again that FLIPD remains a useful measure of *relative* complexity when using either UNets or MLPs.

## E.2 UNet Architecture

We use UNet architectures from `diffusers` [68], mirroring the DM setup in Table 5 except for the score backbone. For greyscale images, we use a convolutional block and two attention-based downsampling blocks with channel sizes of 128, 256, and 256. For colour images, we use two convolutional downsampling blocks (each with 128 channels) followed by two attention downsampling blocks (each with 256 channels). Both setups are mirrored with their upsampling counterparts.

### E.3 Images Sorted by FLIPD

Figures 11, 12, 13, and 14 show 4096 samples of CIFAR10, SVHN, MNIST, and FMNIST sorted according to their FLIPD estimate, showing a gradient transition from the least complex datapoints (e.g., the digit 1 in MNIST) to the most complex ones (e.g., the digit 8 in MNIST). We use MLPs for greyscales and UNets for colour images but see similar trends when switching between backbones.

### E.4 How Many Hutchinson Samples are Needed?

Figure 15 compares the Spearman's rank correlation coefficient between FLIPD while we use $k \in \{1, 50\}$ Hutchinson samples vs. computing the trace deterministically with $D$ Jacobian-vector-products: $(i)$ Hutchinson sampling is particularly well-suited for UNet backbones, having higher correlations compared to their MLP counterparts; $(ii)$ as $t_0$ increases, the correlation becomes smaller, suggesting that the Hutchinson sample complexity increases at larger timescales; $(iii)$ for small $t_0$, even one Hutchinson sample is enough to estimate LID; $(iv)$ for the UNet backbone, 50 Hutchinson samples are enough and have a high correlation (larger than $0.8$) even for $t_0$ as large as $0.5$.

### E.5 More Analysis on Images

While we use 100 nearest neighbours in Table 2 for ESS and LPCA, we tried both with 1000 nearest neighbours and got similar results. Moreover, Figure 16 shows the correlation of FLIPD with PNG for $t_0 \in (0, 1)$, indicating a consistently high correlation at small $t_0$, and a general decrease while increasing $t_0$. Additionally, UNet backbones correlate better with PNG, likely because their convolutional layers capture local pixel differences and high-frequency features, aligning with PNG's internal workings. However, high correlation is still seen when working with an MLP.

Figures 17, 19, 21, and 23 show images with smallest and largest FLIPD estimates at different values of $t_0$ for the UNet backbone and Figures 18, 20, 22, and 24 show the same for the MLP backbone: $(i)$ we see a clear difference in the complexity of top and bottom FLIPD estimates, especially for smaller $t_0$; this difference becomes less distinct as $t_0$ increases; $(ii)$ interestingly, even for larger $t_0$ values with smaller PNG correlations, we qualitatively observe a clustering of the most complex datapoints at the end; however, the characteristic of this clustering changes. For example, see Figures 17 at $t_0 = 0.3$, or 22 and 24 at $t_0 = 0.8$, suggesting that FLIPD focuses on coarse-grained measures of complexity at these scales; and finally $(iii)$ while MLP backbones underperform in sample generation, their orderings are more meaningful, even showing coherent visual clustering up to $t = 0.8$ in all Figures 18, 20, 22, and 24; this is a surprising phenomenon that warrants future exploration.

### E.6 Stable Diffusion

To test FLIPD with Stable Diffusion v1.5 [51], which is finetuned on a subset of LAION-Aesthetics, we sampled 1600 images from LAION-Aesthetics-650k and computed FLIPD scores for each.

We ran FLIPD with $t_0 \in \{0.01, 0.1, 0.3, 0.8\}$ and a single Hutchinson trace sample. In all cases, FLIPD ranking clearly corresponded to complexity, though we decided upon $t_0 = 0.3$ as best capturing the "semantic complexity" of image contents. All 1600 images for $t = 0.3$ are depicted in Figure 25. For all timesteps, we show previews of the 8 lowest- and highest-LID images in Figure 26. Note that LAION is essentially a collection of URLs, and some are outdated. For the comparisons in Figure 26 and Figure 4c, we remove placeholder icons or blank images, which likely correspond to images that, at the time of writing this paper, have been removed from their respective URLs and which are generally given among the lowest LIDs.

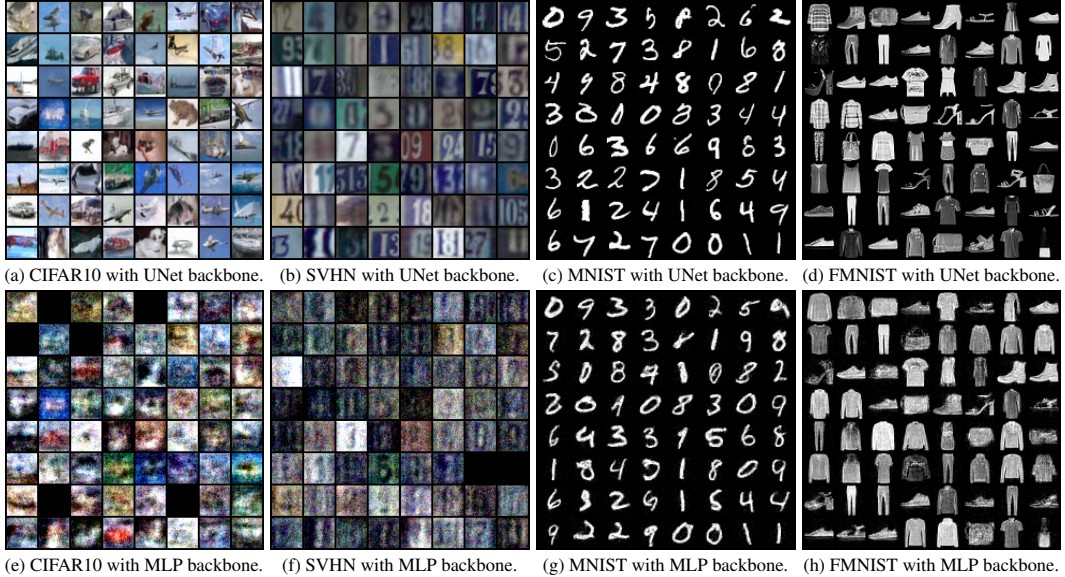

Figure 9: Samples generated from DMs with different score network backbones, using the same seed for control. Despite the variation in backbones, images of the same cell in the grid (comparing top and bottom rows) show rough similarities, especially on CIFAR10 and SVHN.

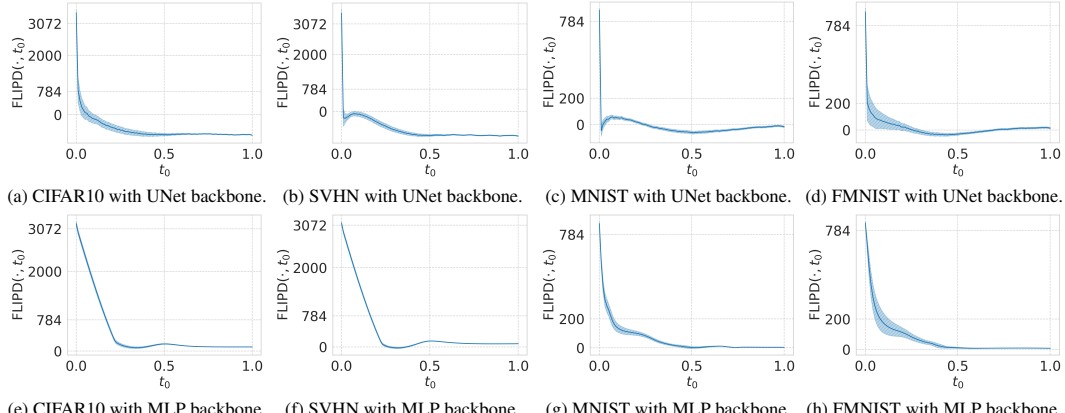

Figure 10: FLIPD curves from all the different DMs with different score network backbones.

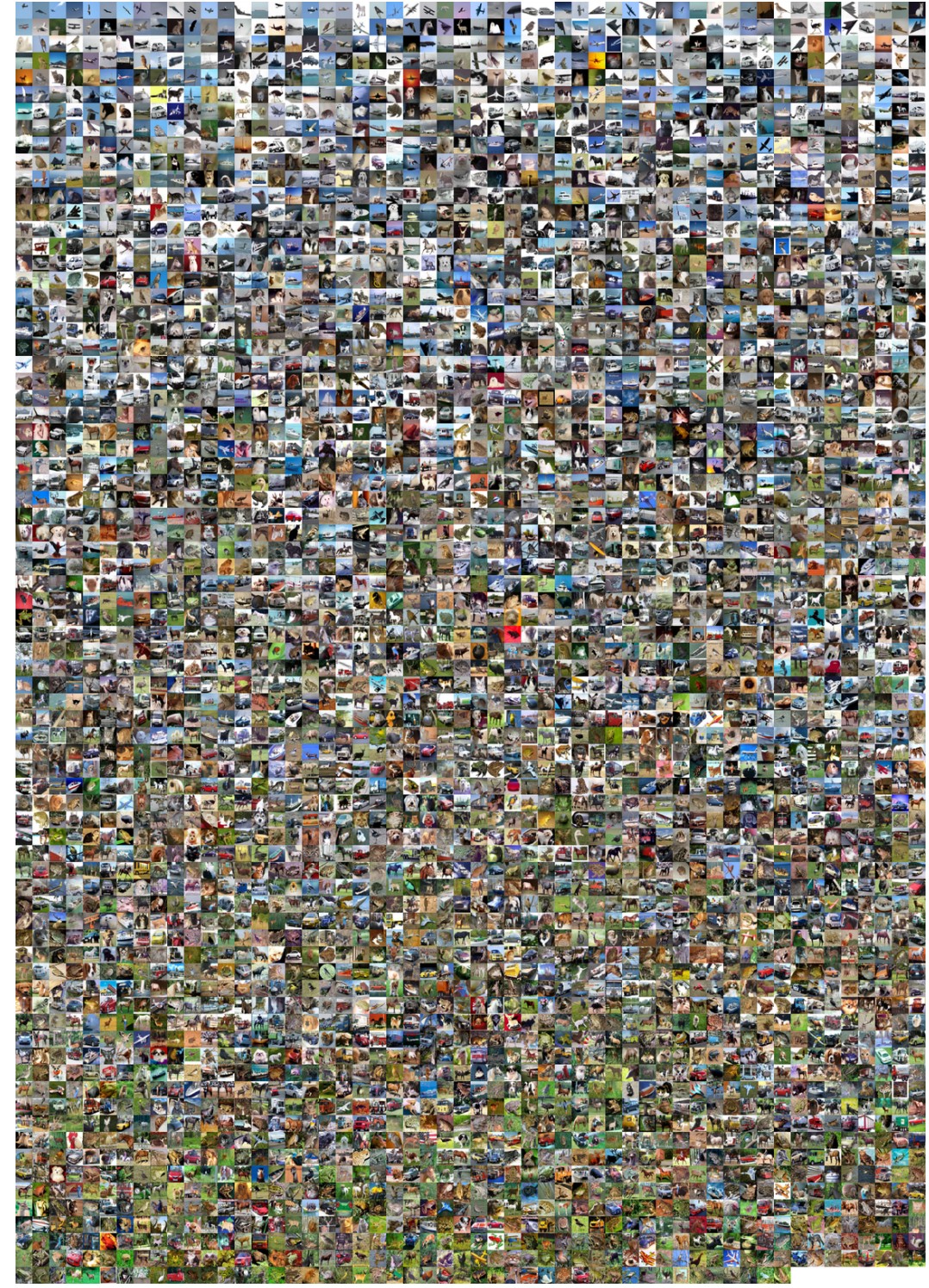

Figure 11: CIFAR10 sorted (left to right and top to bottom) by FLIPD estimate (UNet) at $t_0 = 0.01$.

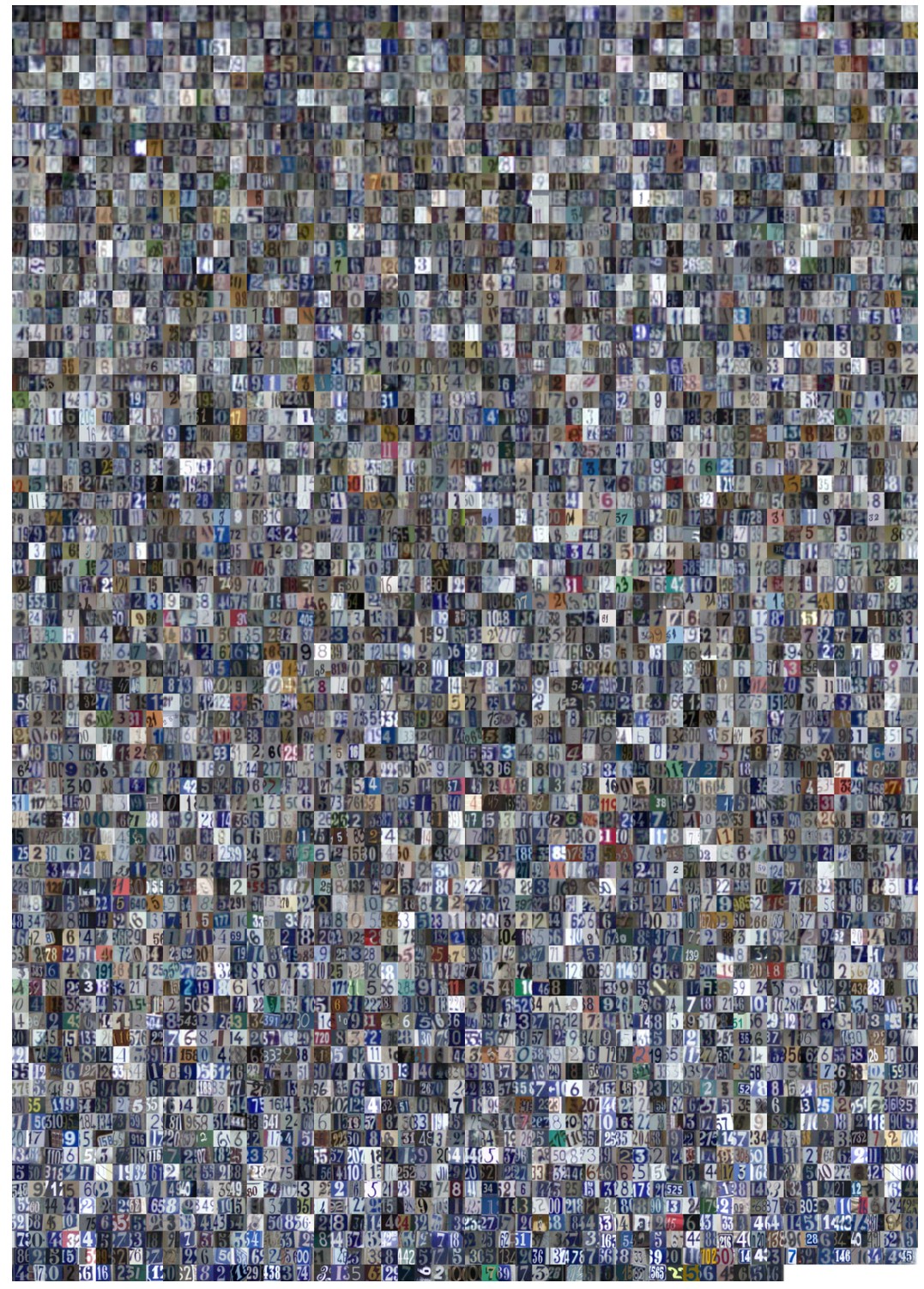

Figure 12: SVHN sorted (left to right and top to bottom) by FLIPD estimate (UNet) at $t_0 = 0.01$.

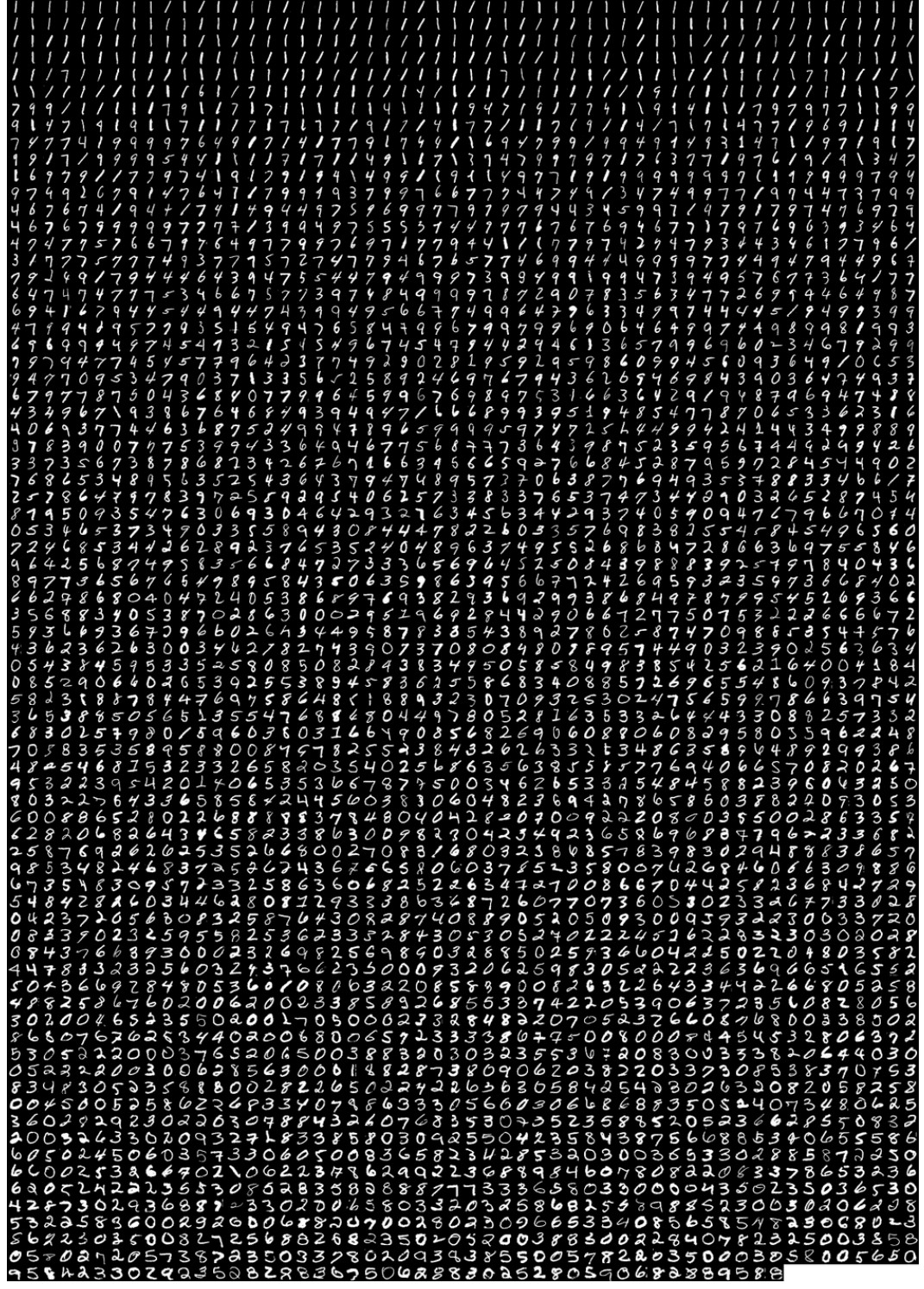

Figure 13: MNIST sorted (left to right and top to bottom) by FLIPD estimate (MLP) at $t_0 = 0.1$.

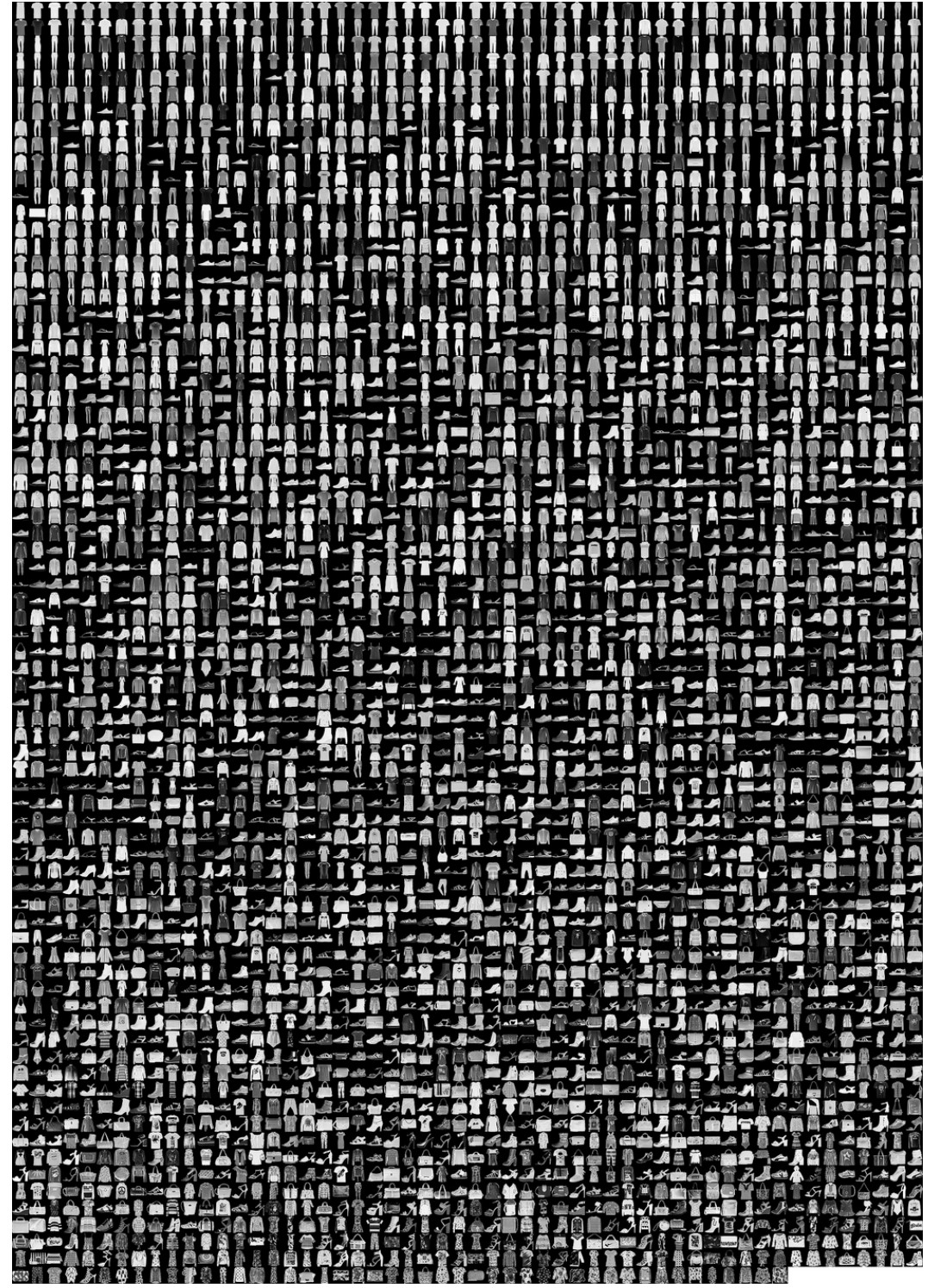

Figure 14: FMNIST sorted (left to right and top to bottom) by FLIPD estimate (MLP) at $t_0 = 0.1$.

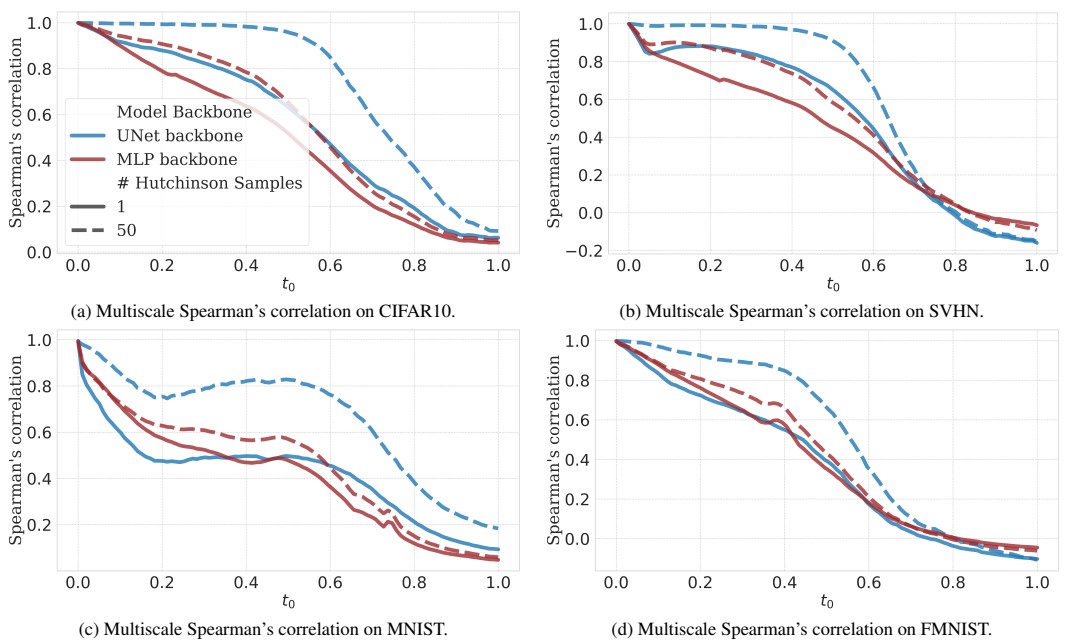

Figure 15: Spearman's correlation of FLIPD estimates while using different numbers of Hutchinson samples compared to computing the trace term of FLIPD deterministically with $D$ Jacobian vector product calls. These estimates are evaluated at different values of $t_0 \in (0, 1)$ on four datasets using the UNet and MLP backbones.

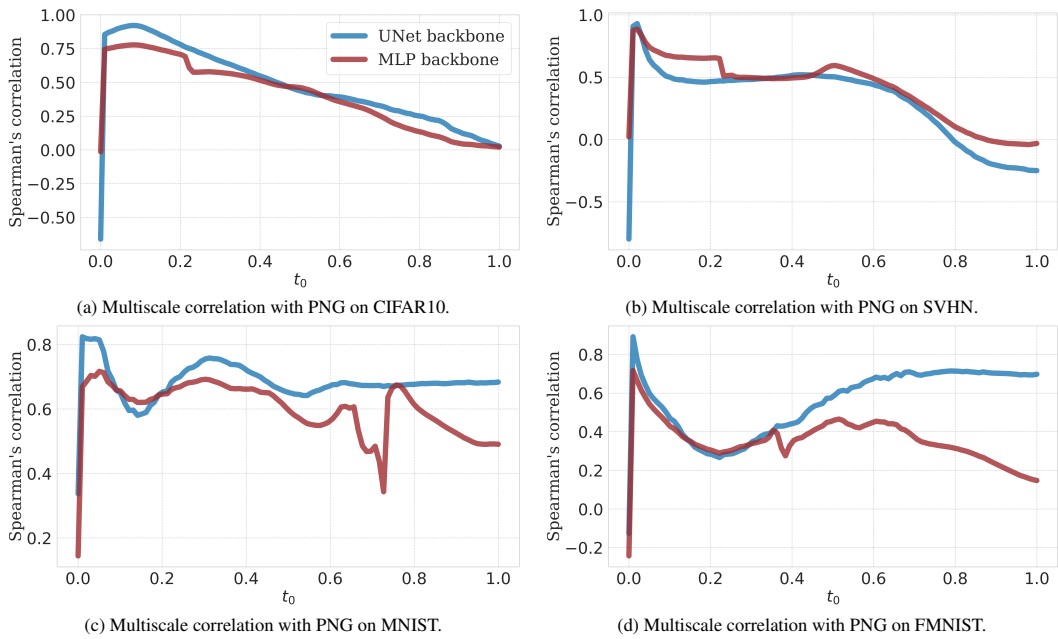

Figure 16: Spearman's correlation of FLIPD estimates with PNG as we sweep $t_0 \in (0, 1)$ on different backbones and different image datasets.

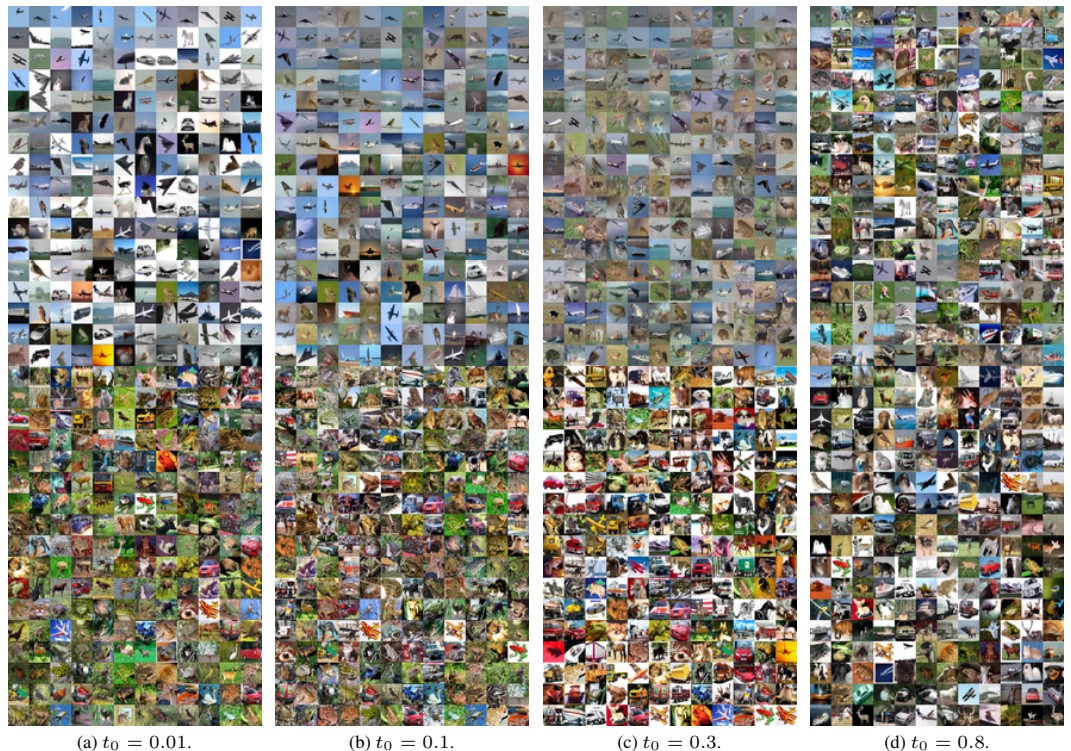

(a) $t_0 = 0.01$.    (b) $t_0 = 0.1$.    (c) $t_0 = 0.3$.    (d) $t_0 = 0.8$.

Figure 17: The 204 smallest (top) and 204 largest (bottom) CIFAR10 FLIPD estimates with UNet evaluated at different $t_0$.

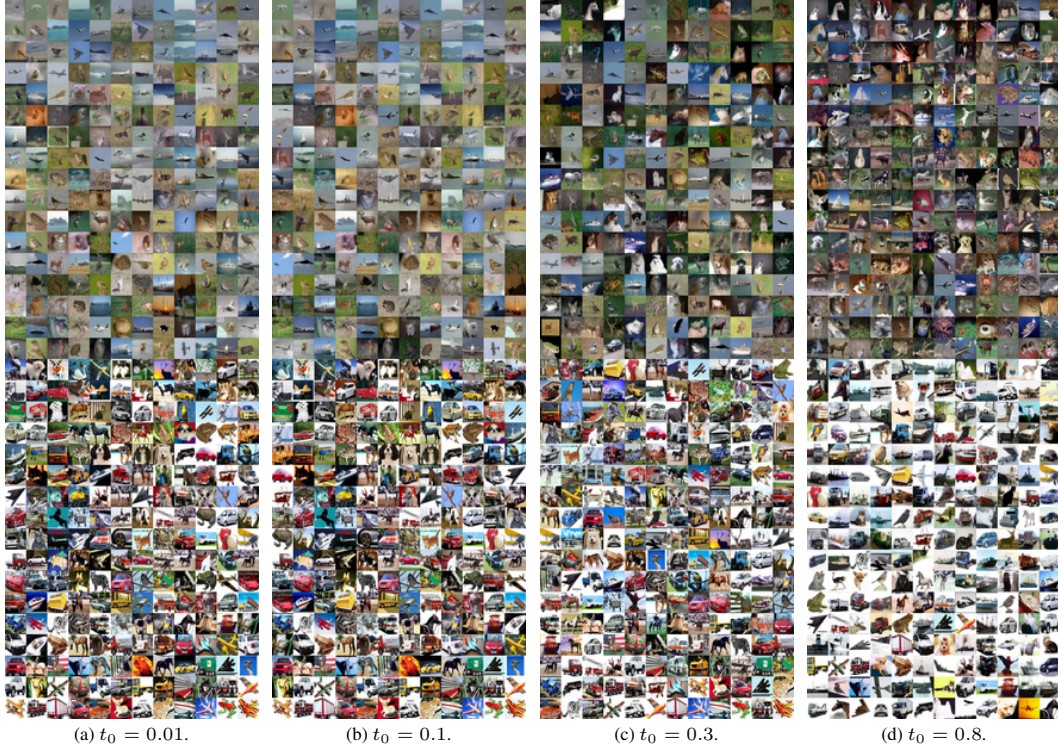

(a) $t_0 = 0.01$.    (b) $t_0 = 0.1$.    (c) $t_0 = 0.3$.    (d) $t_0 = 0.8$.

Figure 18: The 204 smallest (top) and 204 largest (bottom) CIFAR10 FLIPD estimates with MLP evaluated at different $t_0$.

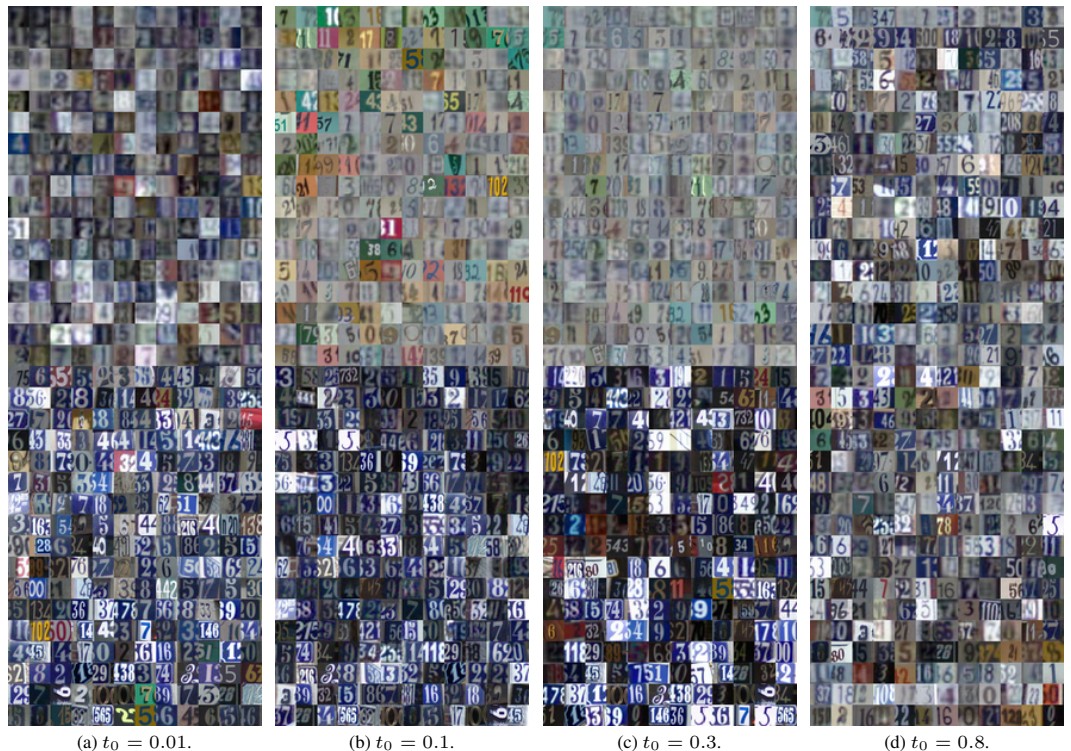

(a) $t_0 = 0.01$.  (b) $t_0 = 0.1$.  (c) $t_0 = 0.3$.  (d) $t_0 = 0.8$.

Figure 19: The 204 smallest (top) and 204 largest (bottom) SVHN FLIPD with UNet estimates evaluated at different $t_0$.

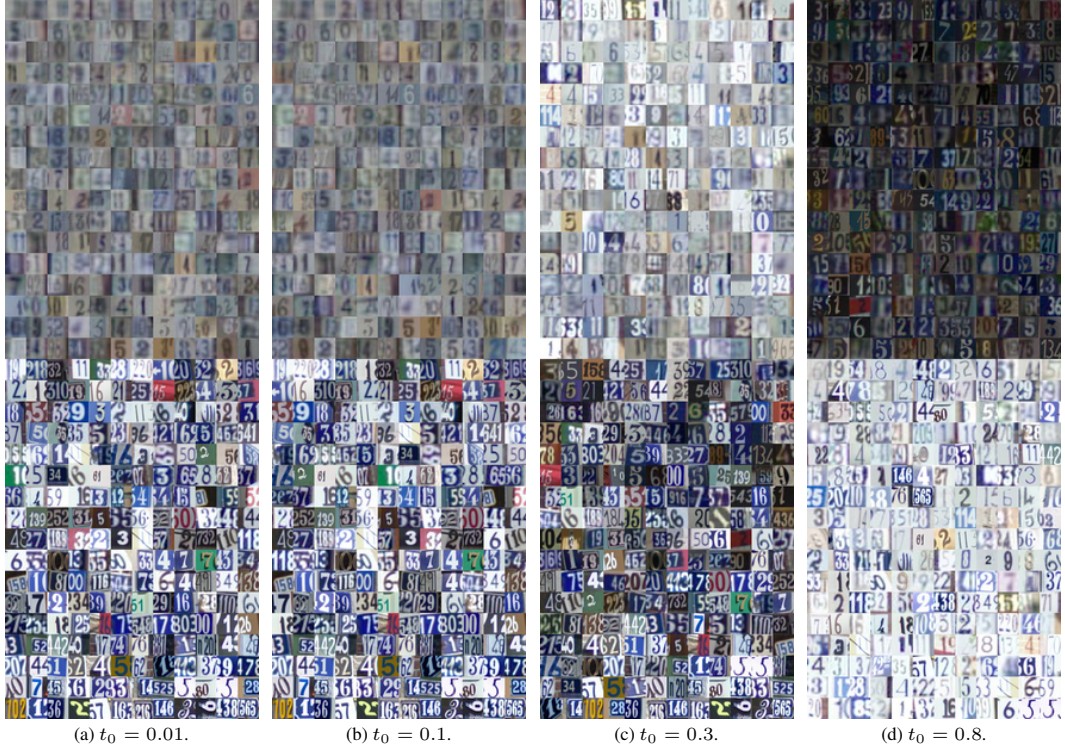

(a) $t_0 = 0.01$.  (b) $t_0 = 0.1$.  (c) $t_0 = 0.3$.  (d) $t_0 = 0.8$.

Figure 20: The 204 smallest (top) and 204 largest (bottom) SVHN FLIPD with MLP estimates evaluated at different $t_0$.

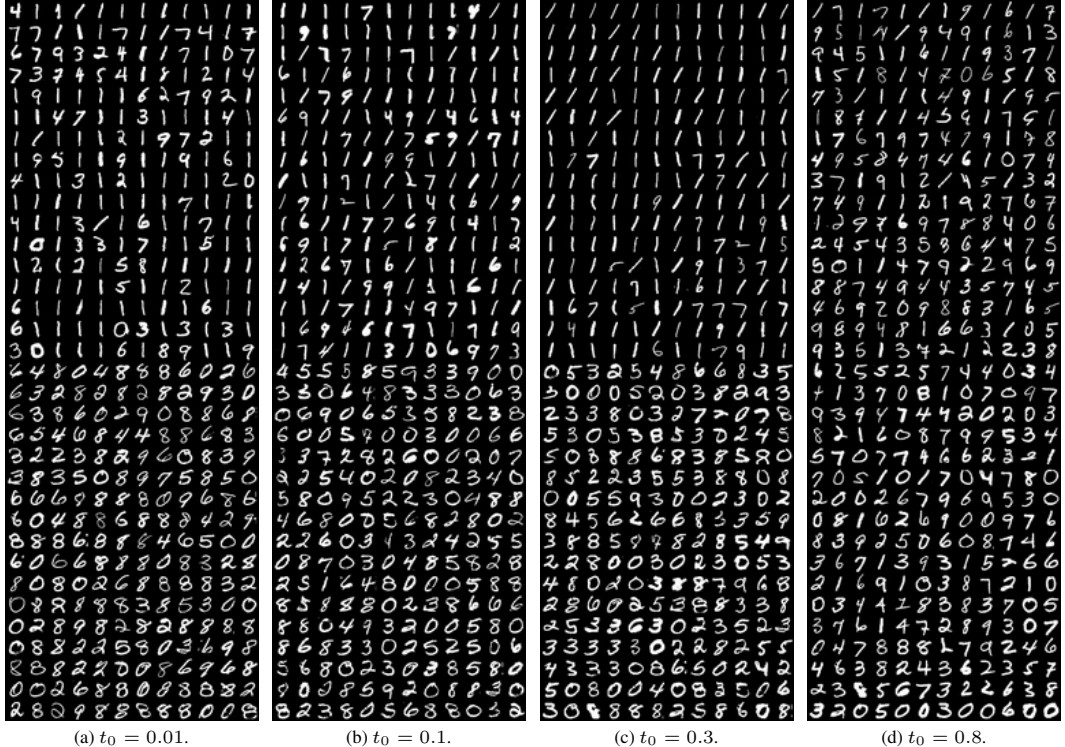

Figure 21: The 204 smallest (top) and 204 largest (bottom) MNIST FLIPD with UNet estimates evaluated at different $t_0$.

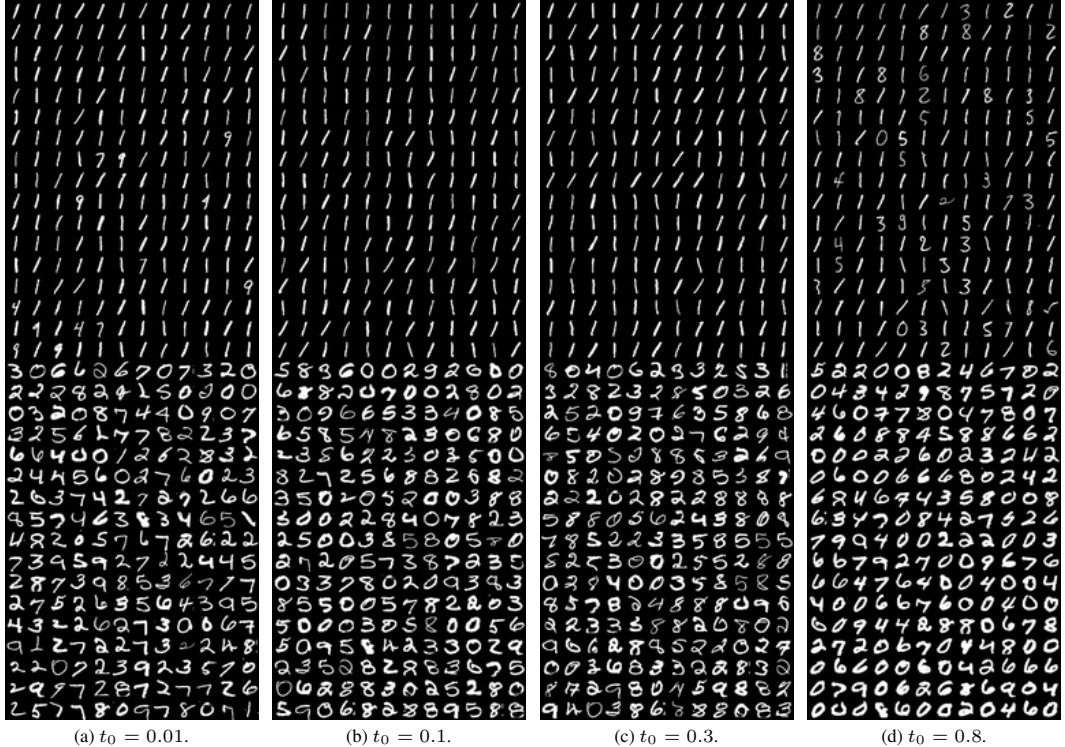

Figure 22: The 204 smallest (top) and 204 largest (bottom) MNIST FLIPD with MLP estimates evaluated at different $t_0$.

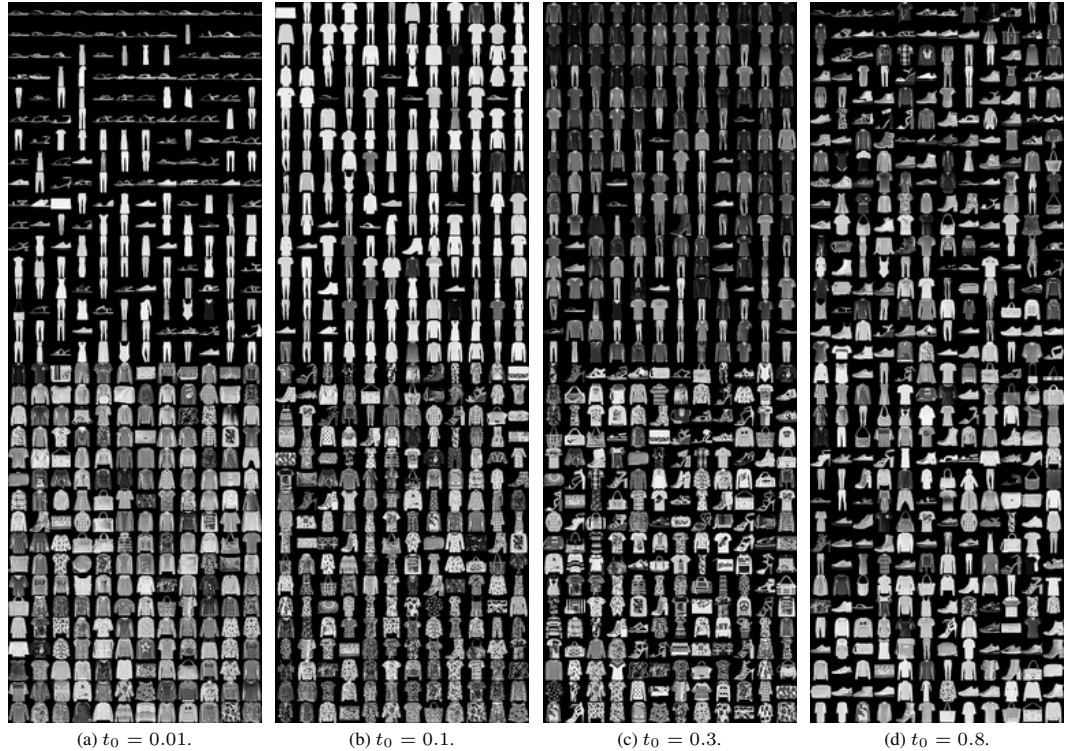

(a) $t_0 = 0.01$.     (b) $t_0 = 0.1$.     (c) $t_0 = 0.3$.     (d) $t_0 = 0.8$.

Figure 23: The 204 smallest (top) and 204 largest (bottom) FMNIST FLIPD estimates with UNet evaluated at different $t_0$.

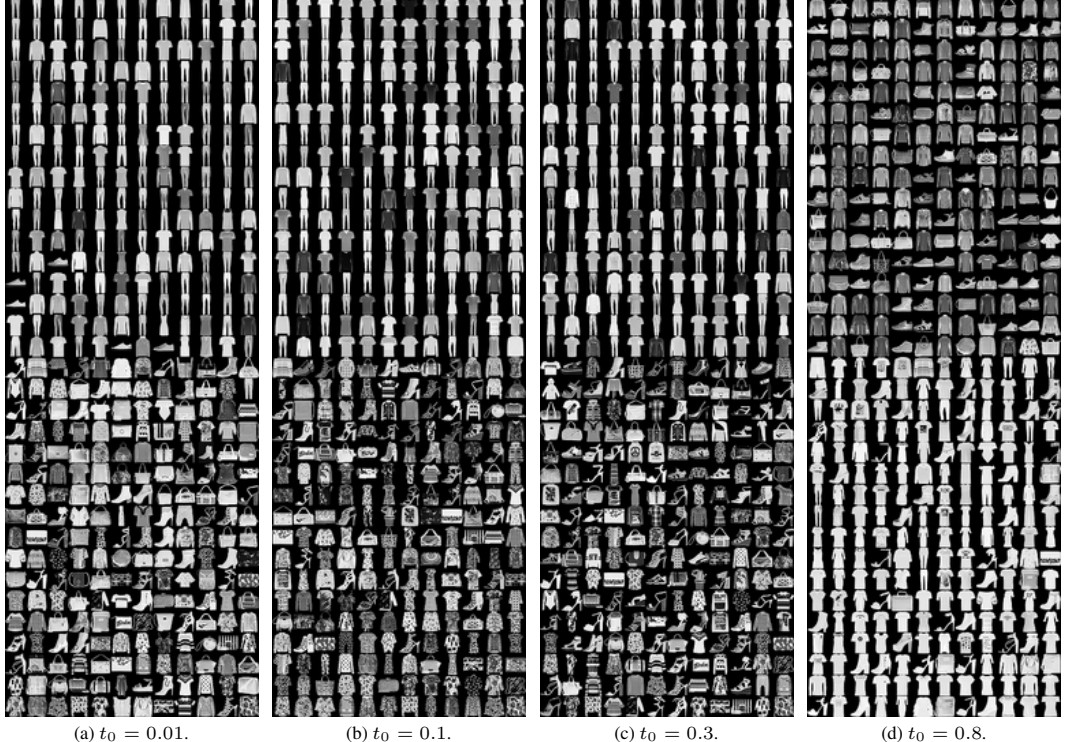

(a) $t_0 = 0.01$.     (b) $t_0 = 0.1$.     (c) $t_0 = 0.3$.     (d) $t_0 = 0.8$.

Figure 24: The 204 smallest (top) and 204 largest (bottom) FMNIST FLIPD estimates with MLP evaluated at different $t_0$.

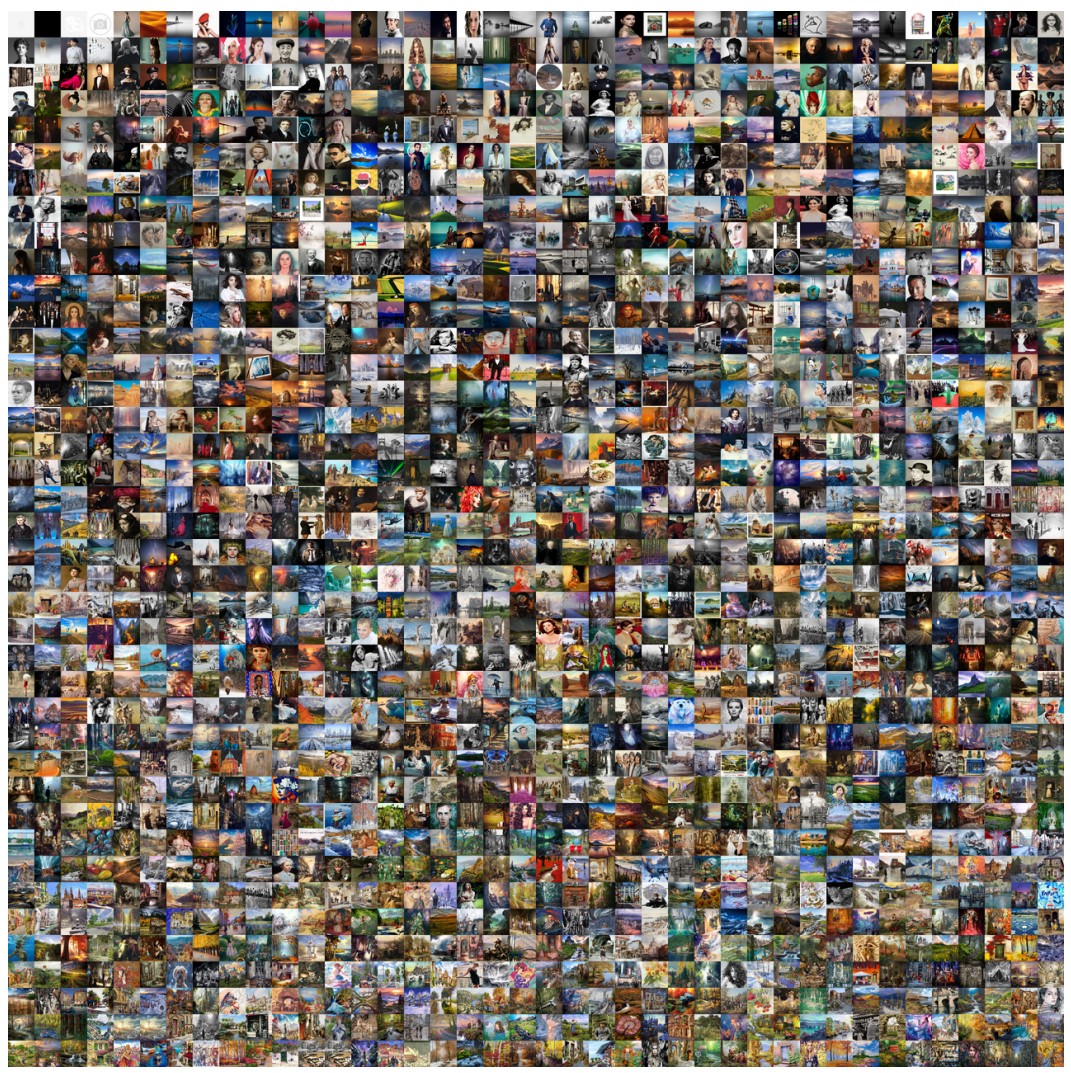

Figure 25: 1600 images from LAION-Aesthetics-625K, sorted by FLIPD estimates with $t_0 = 0.3$.

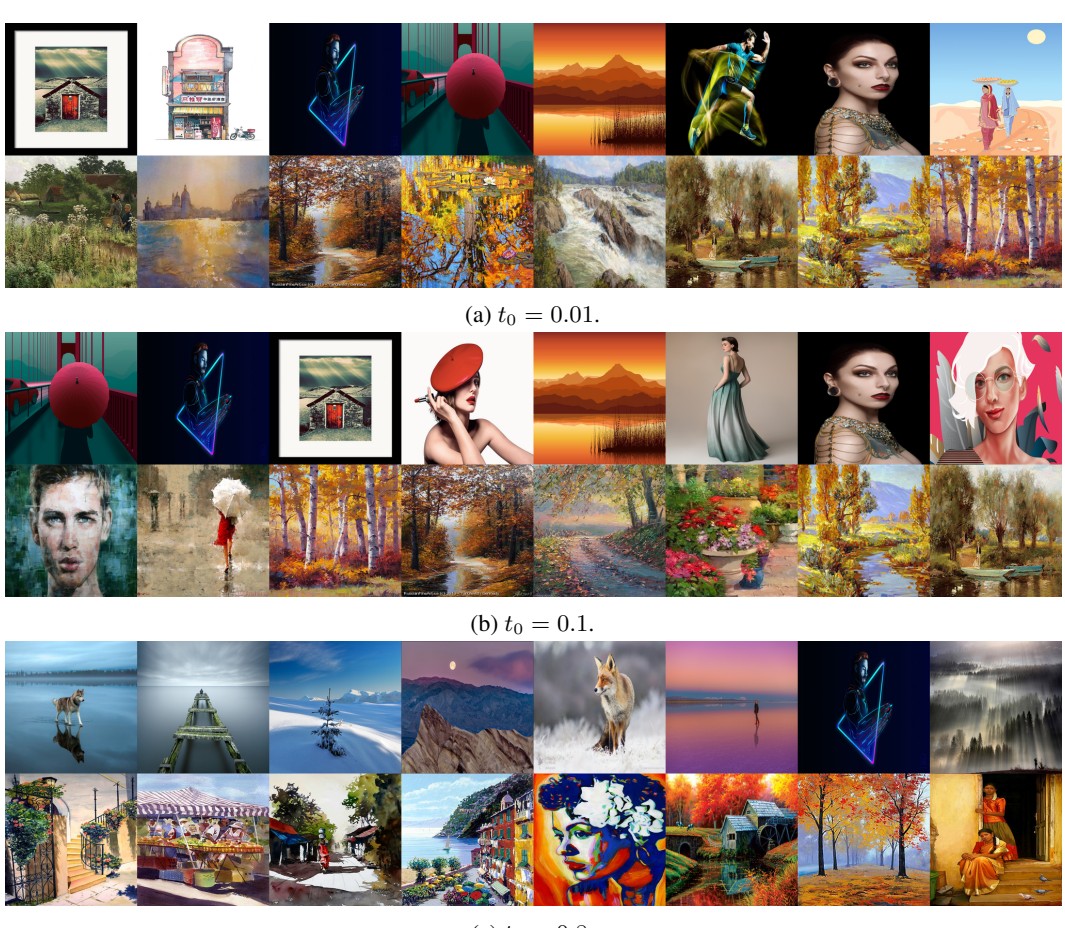

(a) $t_0 = 0.01$.

(b) $t_0 = 0.1$.

(c) $t_0 = 0.8$.

Figure 26: The 8 lowest- and highest-FLIPD values out of 1600 from LAION-Aesthetics-625k evaluated at different values of $t_0$. Placeholder icons or blank images have been excluded from this comparison.

