# OpenReview forum: "A Geometric View of Data Complexity: Efficient Local Intrinsic Dimension Estimation with Diffusion Models"
_NeurIPS.cc/2024/Conference — NeurIPS 2024 spotlight_

### Official Review · Reviewer_MdUJ · 2024-07-12

**Soundness:** 4
**Presentation:** 4
**Contribution:** 3
**Rating:** 7
**Confidence:** 3

**Summary:**

This paper is concerned with estimating the local intrinsic dimension (LID) of a given data manifold. Intuitively, high intrinsic dimension indicates more complex data distribution and an accurate representation of the intrinsic dimension has useful applications in ML such as detecting outliers or adversarial examples.

The approach in the current work suggests incorporating diffusion models as a model-based estimator since these models learn the distribution of the data and implicity the intrinsic dimension. At a high-level the idea is the following: the noise added during diffusion can be seen as a form of Gaussian convolution applied to the data. By analyzing how the data density changes with increasing noise levels, the model is able to implicitly learn the local intrinsic dimension at different points on the manifold.

Furthermore, they uncover and leverage some computational efficiencies using the Fokker-Planck equation which boil down to making a single call to an ODE solver. This computation efficiency also allows for LID estimation for high resolution images, and is the first to be able to do so.

The paper has strong theoretical background and justifications and empirical results that verify their theory. They also provide code with examples to reproduce results. Overall a really excellent and well-executed effort.

**Strengths:**

The ideas behind this paper are very well motivated and compelling. The exposition is very clear and the use of diffusion models to improve LIDL feels like a natural next step.
Also, with respect to the previous work of Stanczuk et al, the improvements to decrease the computational complexity is noteworthy.

Overall the paper is well-written and the theoretical background and proofs are carefully written and appear complete and correct to me. There are extensive experiments including both synthetic and real-world datasets.

**Weaknesses:**

I have a few minor comments and questions below but overall I don't see any substantial weaknesses of the work that are worth addressing further.

**Questions:**

line 107 (suggestion): see also "A Course in Metric Geometry" by Burago, Burago, Ivanov for related discussion about intrinsic dimension and how to compute it for metric spaces.

(nit) Figure 1: For the sake of intuition, this Figure makes sense. However while it is useful to justify that lower/higher FLIPD scores indicate less/more complex images, it leaves open the question of if less/more complex images necessarily have lower/higher FLIPD scores. For example, who's to say that there aren't some images that look even more complex that the 4 you surface which have the max FLIPD values. It's just a personal note and makes figures like this feel misleading without noting that important distinction. That being said, once getting through the Appendix more comprehensive comparison between entire datasets and FLIPD scores are included. Perhaps just point to those additional plots in the appendix when discussing the Figure 1 as well.

---

> ### Author Rebuttal · Authors · 2024-08-07
>
> We greatly appreciate your positive feedback and are delighted that you consider our work to be "an excellent and well-executed effort." In response to your questions and suggestions:
>
> 1. Thank you for the great suggestion! We will make sure to include a reference to the textbook “A Course in Metric Geometry” and include appropriate discussion in the final version of the paper.
> 2. You are absolutely correct to imply that LID is not a perfect measure of complexity. They are related, correlated concepts as per our observations (the correlation between PNG compression and LID, as well as the images presented in the main figure and our appendix), but there is no reason to expect them to be interchangeable. We will ensure that the distinction between the two is made more explicit in the main text.

---

> > ### Comment · Reviewer_MdUJ · 2024-08-12
> >
> > Thank you for your response. These results are interesting. I don't have any additional concerns. I'll maintain my score

---

### Official Review · Reviewer_9X9u · 2024-07-13

**Soundness:** 3
**Presentation:** 3
**Contribution:** 3
**Rating:** 7
**Confidence:** 2

**Summary:**

This paper proposes a novel method for estimating local intrinsic dimension using an existing approximate likelihood estimator with diffusion models. The proposed approach is able to exploit a pre-train diffusion model to evaluate the log density function of the noisy version of the data (i.e., adding Gaussian noise), where solving the ordinary differential equation (ODE) can be avoided by setting $t_0$ as a hyperparameter. Compared to prior works, in which normalizing flow is used for function evaluation and multiple runs of density evaluations are needed for estimation, this framework is much more efficient and shows more promising results.

**Strengths:**

This is a technically sound paper with some theoretical depth and practical usefulness. Although the core idea of the estimation in Eq. (8) is not new, the authors come up with a novel approach to estimate the derivative of the log density function with variance-preserving diffusion models. For real data experiments, using PNG compression as a proxy for evaluation is informative and illustrative. The overall paper is well organized and relatively clearly written.

**Weaknesses:**

If the claim is that the proposed estimation method is simpler and more computationally efficient, it might be useful to include complexity and/or time comparison with existing baselines.

**Questions:**

1.	What are the causes for the spikes in Fig. 2(b) and Fig. 6(c)?
2.	There seems not much difference between different $t_0$’s in Fig. 26. Does it imply that the estimator is not sensitive to the hyperparameter under the case with data of large dimension?
3.	In line 130, what are the ‘mild regularity conditions’ leading to Eq. (7)?

**Limitations:**

Limitations have been discussed.

---

> ### Author Rebuttal · Authors · 2024-08-07
>
> We appreciate your positive review and are pleased that you find our paper well-organized and clearly written. In response to the identified weaknesses and questions, we have addressed them as follows:
>
> ## Weakness: Including time-complexity comparisons
>
> In the general rebuttal section, we provided a time comparison between our method and other baselines, showing that our method delivers 10x faster LID estimates on grayscale 28x28 images and 100x faster on RGB 32x32 images compared to the best baseline, and most importantly, it is the **only** method capable of scaling up to high-resolution images in a tractable manner. Given that computational efficiency is one of the most critical practical aspects of our work, we believe the added content will further solidify our claims and provide greater clarity. We will also include these details in the final version of the paper.
>
> ## Questions:
>
>
> 1. That is a great observation! These curves roughly indicate the rate of change in the Gaussian convolved density $\varrho(x, \delta)$. The observed "spike" in the mentioned curves is likely due to a sudden increase in the Gaussian convolution at that point, potentially caused by mixing distributions from different manifolds. For example, in Figure 6-c where the dataset is a Lollipop (a mixture of 0, 1, and 2-dimensional components), Gaussian noise added to another submanifold may cause datapoints from the 2D or 1D submanifold to mix with the isolated 0D point. A similar situation occurs in the "string within a doughnut" experiment of Figure 2-b. These artifacts are irrelevant for LID estimation with FLIPD, however, as the theory behind our LID estimates holds when we let $\delta \to -\infty$, or equivalently, $t_0 \to 0$. In these regimes, $t_0$ is small and such mixings do not occur.
> 2. You are correct that the orderings are not sensitive to $t_0$, but only in a certain range. Note that for extremely high $t_0$ we see that the ordering does not correspond to the data complexity anymore; this is evident in Figures 17-d, 19-d, 21-d, and 23-d. We also see that in Figure 16, when $t_0 \to 1$ the correlation between FLIPD estimates and PNG compression size significantly reduces. We would also like to refer you to our synthetic experiment "string within a doughnut" illustrated in Figure 3, and the multivariate Gaussian in Figure 8 of the appendix, where the choice of $t_0$ has a significant impact on the overall estimate.
> 3. The regularity conditions mentioned in line 130 are those used by Tempczyk et al. [61]. Here, the support of the data-generating distribution being a disjoint union of manifolds means that this measure can be thought of as a mixture of measures on each individual manifold. The regularity condition is that each of these measures is smooth, that is, that for every chart on the manifold, the pushforward of the corresponding measure through the chart is absolutely continuous with respect to the Lebesgue measure, and that the density of this pushforward is locally bounded away from 0. While these conditions are indeed mild, they involve charts and measure-theoretic terms, which we explicitly wanted to avoid to keep the paper as accessible as possible.

---

> > ### Comment · Reviewer_9X9u · 2024-08-11
> >
> > Thank you for adding the complexity comparison and the response. I think that would make the work more comprehensive and solid.

---

### Official Review · Reviewer_MJca · 2024-07-13

**Soundness:** 3
**Presentation:** 3
**Contribution:** 4
**Rating:** 7
**Confidence:** 4

**Summary:**

The authors propose to employ the best available generative models, i.e., diffusion models, to the estimation of local intrinsic dimension. To this end, they build upon the LIDL estimator of Tempczyk et al. [ICML, 2022], but crucially resolve a number of limitations: (1) direct application of LIDL requires training $m$ diffusion models, (2) even by enabling only a single diffusion model, LIDL still requires $m$ ODE solves, and finally (3) even by enabling a single ODE, it requires repeated estimation of the trace of a Jacobian. The result is a set of explicit formulas for the LID estimate for each of the common diffusion models, leveraging the Fokker-Planck equation associated with the Ito stochastic differential equation. The asymptotic correctness of the estimate is established for an idealized linear setting. This algorithmic contribution is further demonstrated through an enhanced intrinsic dimension benchmark, which highlights the limitations of prior techniques.

**Strengths:**

- Greatly simplifies LID estimation via diffusion models, presenting a significant speed up over prior work.
- The technique is well-principled, and generalizes to different diffusion model families, enabling explicit formulas to be derived as demonstrated by the authors.
- The proposed benchmark (with an in-depth analysis) helps to further test against prior work, better demonstrating their shortcomings.
- Relating LID to PNG compression size is also a valuable idea.

**Weaknesses:**

Unfortunately, there seem to be a number of technical issues; see below. It doesn't help that some of those issues appear out of the blue upon reading into the experiments section, and don't match the impression one gets from the high level summary given in the abstract and introduction.

Disregarding those issues for a moment, it's still unsettling that UNets deviate from MLPs in terms of their FLIPD curves lacking easily detectable knees. This calls into question whether knees are the best or only way to select values of $t$, or whether there are other techniques.

**Questions:**

Technical Issues
=============
Section 3
- L211: In my experience, such constant factors relate to the curvature of the manifold. So, the result makes sense when the manifold is an affine subspace. In this sense, the reasoning given on L219-222 appears erroneous. That is, concentration on a point mass in the limit still leaves the intrinsic dimension intact, rather than totally detaching from the manifold structure to yield an intrinsic dimension of 0.

Section 4
- L309: It seems that this is the first mention of an MLP, which this line suggests is what was used for the synthetic experiments.
  - Unfortunately, this seems like more than a simple lapse in communication.
  - Rather, upon switching from those previously undisclosed MLPs to UNets, those nice findings about knees in the curve no longer hold.
  - This critical transition comes out of the blue, since it wasn't mentioned either in the abstract or the introduction.
  - From a first reading, it was difficult to keep track of which findings reported for MLPs carry over upon switching to UNets, where the authors claim the resulting estimates remain valuable measures of complexity; see L323.
- FLIPD on high resolution images
  - L359: is it correct that running FLIPD in the latent space inherits existing biases, e.g., prioritizing semantics rather than "true" manifold geometry?
  - It's unfortunate that the actual results as conveyed on L345-360 seem to be misrepresented in the high level summaries given in the abstract and introduction.

Other Technical Comments
=====================
Section 3
- L190: I'm not sure I follow how $\nu$ does not depend on $\hat{\varrho}$. Perhaps what's meant is that $\nu$ is computed from Eq. 12, and the resulting values are fixed to define $\hat{\varrho}$ through the ODE in Eq. 13. That way, if $\nu$ was computed with $\hat{s} = s$, then Eq. 13 can be expressed in terms of $\log \varrho(x, \delta)$ itself.

Section 4
- L284: Does the diffeomorphism of the flow hold for practical numerical evaluations? Did the authors observe any fluctuations in the LID due to such numerical errors? If so, can those errors be bounded?

**Limitations:**

I'm guessing the authors wanted to go all the way, even when UNets failed to deliver the same nice story afforded by MLPs. It's unfortunate that the story didn't go that way - or perhaps it's an opportunity? Either way, the current presentation of this story may initially appear to misrepresent the actual findings. In my view, this is a serious issue that must be addressed before publication.

All that aside, the paper does achieve a lot. Only that the story had to be a bit more complex / interesting.

---

> ### Author Rebuttal · Authors · 2024-08-07
>
> We highly appreciate your comprehensive review, and we are grateful that you recognize the significant achievements of our work despite some concerns. Before addressing your points in detail, we would like to clarify a potential misunderstanding regarding the UNet vs. MLP issue. While certain patterns observed with MLPs, like knees, may not carry over to UNets, we firmly maintain that FLIPD estimates remain useful measures of relative complexity, as evidenced by our image rankings with a UNet backbone. Most importantly, the results in **Table 2** use a **UNet** backbone and show a high correlation with PNG compression size. Interestingly, we often observe an *even greater* correlation with PNG compression size when using a UNet backbone compared to MLPs, as shown in Figure 16 of the Appendix.
>
> With that in mind, we propose FLIPD as a strong LID estimator on structured manifolds and a relevant measure of complexity for images. Once again, we thank you for bringing this up, and will ensure that the introduction and abstract are updated to more accurately reflect our contributions in the final version of the paper.
>
> We will now provide detailed responses to the individual concerns:
>
>
> ## Technical Issue (L211)
> Thank you for the astute observation. Indeed, these terms tend to be related to curvature, and studying the effect of curvature in this setting seems an extremely appropriate future direction. However, our conjecture is not that this term does not depend on curvature nor that its derivative wrt $\delta$ is exactly $0$; rather, it’s simply that its derivative goes to $0$ as $\delta \rightarrow -\infty$. We indeed expect the speed of convergence to depend on the curvature. Additionally, we would like to emphasize that L211 is merely a conjecture, and it is based on the strong synthetic results we obtained, as detailed in Appendix D. For this we refer you to our experimental results, particularly those involving non-linear manifolds generated by normalizing flows denoted with $\mathcal{F}$. It is also worth mentioning that LIDL, our baseline, also relies on the curvature-related O(1) term being well-behaved.
>
> Furthermore, the limit $\delta \to -\infty$ is not related to the manifold being detached. It suggests that as $\delta \to -\infty$, $\varrho(x, \delta)$ becomes increasingly local, and the dependence on the global characteristics of the manifold diminishes. Our intuition is that as $\delta \to -\infty$, $\log \varrho(x, \delta)$ becomes as local as possible and, after a certain point, the local approximation to the manifold given by its tangent space can be considered “good enough”.
>
> ## Technical Issue (L309)
> We would like to clarify that using knees is not necessarily the best way to obtain accurate LID estimates and is not a crucial part of our method. In fact, choosing a sufficiently small $t_0$ still produces very reasonable results (see Table 5 of the Appendix). Therefore, the absence of knees does not render the estimator useless. Moreover, we firmly believe that even without the knees, the FLIPD estimators with UNets are useful measures of complexity, as evidenced by the high correlation with PNG compression size shown in Table 2.
> We acknowledge that the differences between UNets and MLPs are counterintuitive and warrant further exploration in future work. Our explanation for why this issue may arise is included in Appendix E.1. We will add more context and move some of that explanation to the main text in the final version of the paper.
>
> ## Technical Issue (FLIPD on high resolution images)
> This is an excellent point and might indeed contribute to the more semantically meaningful orderings we observe. However, we would like to mention that the encoder/decoder of Stable Diffusion is designed not just to encode semantic information; the latent space remains image-shaped and high-dimensional. Additionally, we refer you to our results in Figures 17 through 24, where the model does not use a latent space and still aligns with semantic complexity. That said, we acknowledge that using a latent space for Stable Diffusion might lead to misinterpretation of the claims in the introduction. We will ensure that these claims are made more explicit in the final version of the paper.
>
> ## Technical Comment (L190)
> Your latter interpretation is correct: $\hat{\varrho}$ is simply *defined* using $\nu$ and the ODE in Eq. (13). It is important to note that $\hat{\varrho}$ and $\varrho$ are not the same, and $\hat{\varrho}$ is the solution to the ODE in (12) where the initial value is *artificially* set to zero, and thus $\varrho$ and $\hat{\varrho}$ differ by a constant. In Section 3.2, we demonstrate that the constant difference between $\hat{\varrho}$ and $\varrho$ is unimportant because, when a linear regression is employed (similar to LIDL), this constant only affects the intercept and not the slope; with the slope being the term that is important for computing LID. We will reword this section in the final version of the paper for better clarity.
>
> ## Technical Comment (L284)
> This is yet another excellent point! Indeed, if certain factors are not controlled, normalizing flows can distort the manifold, causing numerical changes in the LID. For example, a square stretched in one direction and squeezed in another can numerically deceive an LID estimator to consider it a line with LID=1 rather than a rectangle with LID=2. We incorporate activation normalizations and use at most 10 rational quadratic transforms in our flow-induced manifolds. This ensures the overall transformation is not ill-conditioned.
>
>
> We thank you once again for your excellent review! Your insightful comments have improved our work. Finally, if you find that our explanation satisfactorily addresses your concerns, we kindly request that you consider raising the score!

---

> ### Comment · Reviewer_MJca · 2024-08-11
> **Follow up**
>
> Thanks for addressing my comments. I revised the language for my confusion about what carries over from MLPs to UNets. Since this was my main reservation, I have increased my score. It would have been nice to see the revised draft, but I trust the authors to reflect discussions with reviewers in the final version.
>
>
> Follow up technical remarks
> =====================
>
> **L211** - Regarding curvature terms, I didn't find any mention of those in the paper. It would be nice to briefly discuss that highlighting how they show up in the formulation of LIDL. And thanks for resolving the simplistic counter-argument I posed, though it was probably triggered by the comment on approaching point-mass. Speaking of the tangent space seems more helpful.
>
> **L309** - I encourage the authors to reflect on how to best smooth out the abandonment of knees upon switching to UNets. (I can see that knee analysis helped demonstrate how the MLP approach captures the true ID for the synthetic dataset, and better aligns with the theory - see also next block.) More concretely, it may help to break up the experiment section, e.g., into two subsections, in order to allow for a more proper anticipation of the added complications of image experiments. That is, those complications appear to be more than just nuance with the specific experiments and their setup, but rather stem from remaining gaps in the development.
>
> Beyond revising the experiments section, I would also recommend tuning down the claimed contributions, as early as the abstract/introduction/summary, to be based more heavily on MLPs and the synthetic evaluations against true IDs, while suitably positioning contributions related to image datasets and the adaptations needed for them as more of an empirical contribution or an extension of the core contributions. I trust the authors to make the right call there, and to further identify and discuss the remaining gaps in the limitations section.
>
> **L787-788**: FLIPD curves with MLPs, which have clearly-discernible knees as predicted by the theory
> - Please clarify explicitly which parts of the theory predict those clearly-discernible knees.
>
> **L792-793**: surprisingly poor FLIPD estimates of UNets
> - It would help to reflect on this language in light of the authors' assertion*. It would help to tune down the more subjective language and refer to specific qualitative/quantitative empirical results.
> (*) From the rebuttal: *we firmly believe that even without the knees, the FLIPD estimators with UNets are useful measures of complexity, as evidenced by the high correlation with PNG compression size shown in Table 2*
>
> **Overall recommendation:**
> Given the remaining gaps in analysis - theoretical (i.e. non-linear manifolds) and empirical (i.e. MLP to UNets), I strongly recommend to tune down further hypotheses and conjectures, and instead utilize the space to better present the known facts and available evidence.

---

> > ### Author Response · Authors · 2024-08-12
> > **Follow up and thanks**
> >
> > Thank you for your reply and for raising your score! Of course it is not possible to make manuscript modifications in the rebuttal phase for NeurIPS this year, but we commit to making the requested updates for the final version of the paper: We will clearly distinguish between the experiments involving MLPs and those involving UNets, highlight in the experiments section that some empirical findings observed in MLPs (such as knees) do not fully extend to UNets. Additionally, we will clarify that identifying knee points is not essential in the theoretical analysis. Despite these modifications, we will highlight the results obtained from UNet experiments still correspond to meaningful measures of image complexity. Furthermore, we will make our conjecture in Section 3.3 more explicit by adding extra context from the rebuttal.
> >
> >
> > As for further modifications:
> >
> >
> > ## Please clarify explicitly which parts of the theory predict those clearly-discernible knees. (L787-788)
> >
> > We will clarify phrasing here: the theory does not formally suggest that a knee should occur, it simply predicts the limit as $t \to 0$ being the LID.  Therefore, it is reasonable to expect the FLIPD curve to stabilize around a certain value (i.e., the LID) as $t$ approaches zero. The appearance of knees in the MLP experiments is an empirical observation on our part. In other words, while the theory suggests that observing knees is plausible, their absence in the UNet experiments is also consistent with the theoretical framework.
> >
> > ## Surprisingly poor FLIPD estimates of UNets (L792-793)
> >
> > We will also clarify phrasing here. Our intention here was to point out that even though the lack of knees observed when using UNets makes it harder to select $t_0$ to recover the true LID, the relative rankings obtained by FLIPD remain meaningful. This is evidenced qualitatively when looking at images sorted by their FLIPD values in Figures 17, 19, 21, 23, and 25, and quantitatively by measuring the correlation against PNG compression length in Table 2. This is why we claim that FLIPD remains a valid measure of relative complexity when using UNets, despite the lack of knees.

---

> > > ### Comment · Reviewer_MJca · 2024-08-12
> > > **Remaining gaps and concerns over presentation & claims**
> > >
> > > TL;DR please provide (1) the updated abstract, and (2) the contribution summary as typically included near the end of the introduction section.
> > >
> > > Will follow up with more discussion.

---

> > > > ### Author Response · Authors · 2024-08-13
> > > > **Updated phrasing**
> > > >
> > > > Thank you again for the continued engagement. Below we include revised versions of the abstract and of the last paragraph of the introduction. If you still have lingering concerns about phrasing please let us know and we will make further updates.
> > > >
> > > > ## Abstract
> > > >
> > > > High-dimensional data commonly lies on low-dimensional submanifolds, and estimating the local intrinsic dimension (LID) of a datum -- i.e. the dimension of the submanifold it belongs to -- is a longstanding problem. LID can be understood as the number of local factors of variation: the more factors of variation a datum has, the more complex it tends to be. Estimating this quantity has proven useful in contexts ranging from generalization in neural networks to detection of out-of-distribution data, adversarial examples, and AI-generated text. The recent successes of deep generative models present an opportunity to leverage them for LID estimation, but current methods based on generative models produce inaccurate estimates, require more than a single pre-trained model, are computationally intensive, or do not exploit the best available deep generative models, i.e. diffusion models (DMs). In this work, we show that the Fokker-Planck equation associated with a DM can provide a LID estimator which addresses all the aforementioned deficiencies. Our estimator, called FLIPD, is compatible with all popular DMs, and relies on a single hyperparameter. FLIPD outperforms existing baselines on synthetic LID estimation benchmarks while staying robust to the choice of hyperparameter. We also apply FLIPD on natural images where the true LID is unknown. Despite the resulting LID estimates being less stable over the choice of hyperparameter, FLIPD remains a valid measure of relative image complexity; compared to competing estimators, FLIPD exhibits a consistently higher correlation with non-LID measures of image complexity, and better aligns with qualitative assessments of complexity. Moreover, FLIPD is orders of magnitude faster than other LID estimators, and the first one to be tractable at the scale of Stable Diffusion.
> > > >
> > > > ## Introduction
> > > >
> > > > Our contributions are: $(i)$ showing how DMs can be efficiently combined with LIDL in a way which requires a single call to an ODE solver; $(ii)$ leveraging the Fokker-Planck equation to propose FLIPD, thus improving upon the estimator and circumventing the need for an ODE solver altogether; $(iii)$ motivating FLIPD theoretically; $(iv)$ introducing an expanded suite of LID estimation benchmark tasks that reveals gaps in prior evaluations, especially that other estimators do not remain accurate as the complexity of the manifold increases; $(v)$ demonstrating that when using fully-connected architectures for diffusion models, FLIPD produces estimates which are robust with respect to the choice of its hyperparameter and outperforms existing baselines -- especially as dimension increases -- while being much more computationally efficient; $(vi)$ showing that when applied to natural images along with UNet architectures, despite not exhibiting the same hyperparameter robustness, FLIPD estimates consistently align with other measures of complexity such as PNG compression length, and with qualitative assessments of complexity, highlighting that the LID estimates provided by FLIPD remain valid measures of relative image complexity;  and $(vii)$ demonstrating that when applied to the latent space of Stable Diffusion, FLIPD can estimate LID for extremely high-resolution images ($\sim 10^6$ pixel dimensions) for the first time.

---

> ### Comment · Reviewer_MJca · 2024-08-13
> **Making progress**
>
> Thanks for the updates. I appreciate the authors reflecting on the core claims in light of the discussion. To be clear, it is not my intention to diminish the contributions, but to present exactly what has been achieved and what's left.
>
> TL;DR: please (1) tune down the **Momentum** adequately positioning the theoretical contributions separately from their practical implementations, (2) Surface and discuss **Issue#1** and **Issue#2** early on in the introduction, all through the experiments section. (3) Consider reducing focus on knees a little bit, and instead paying more attention to how to best use the estimates across multiple settings of the hyperparameters as commonly done in robust estimation, (4) Revisit the wording in "addresses all the aforementioned deficiencies" in the abstract, noting remaining gaps applying the estimators in the absence of knees.
>
> Taking a step back after yet another reading, I think I can identify the main underlying gap here. Please feel free to clarify any of those points:
> - The paper builds momentum based off the theoretical derivations [**Momentum**] culminating in the proposed estimator as encoded in Theorem 3.1 (Equation 15). This is a great theoretical result with lots of merit towards publication.
> - However, applying this neat result in practice immediately hits a couple issues as recounted across a number of pages:
>   - [**Issue#1**] L230-232: The effect of $t_0$ FLIPD requires setting $t_0$ close to 0 (since all the theory holds in the $\delta  \to -\infty$ regime). It is important to note that DMs fitted to low-dimensional manifolds are known to exhibit numerically unstable scores $s(·, t_0)$ as $t_0 \to 0$.
>   - [**Issue#2**] L319-320: As long as the DM accurately fits the data manifold, our theory should hold, regardless of the choice of backbone. Yet, we see that UNets do not produce a clear knee in the FLIPD curves
>   - [**Issue#2**] L792-795:  To explain the surprisingly poor FLIPD estimates of UNets, we hypothesize that the convolutional layers in the UNet provide some inductive biases which, while helpful to produce visually pleasing images, might also encourage the network to over-fixate on high-frequency features which are not visually perceptible.
>   - (Note that those last two remarks seem to contradict each other)
> - To cope with the challenges of applying the theory in practice, the authors adopted a number of accommodations
>   - First, the notion of knees, which seem to be partly motivated by related work on normalizing flows
>     - L269-270: As mentioned, we persistently see knees in FLIPD curves. Not only is this in line with the observations of LIDL on normalizing flows
>   - From there, attention shifts heavily to knees overshadowing **Issue#1**].
>     - I really did not appreciate conflating the observation of knees with possible implications of Theorem 3.1 (L787-788), which follows the general sense of overshooting the **Momentum**.
>     - The multiscale interpretation was appreciated though (L249)
>   - Upon transitioning to UNets, the focus is still on knees and their sudden disappearance, which further dilutes **Issue#1** which is now mixed in non-trivial ways with **Issue#2**.
>   - There's clearly more work to be done on how to best utilize the resulting estimates for various hyperparameters, whereas the experiments section mainly focuses on validating the estimates to demonstrate the theory despite the accommodations needed to make it work.
>
> Looking at the revised summaries, I note the following elements of the abstract:
> - [..] current methods based on generative models *produce inaccurate estimates*, require more than a single pre-trained model, are computationally intensive, or do not exploit the best available deep generative models, i.e. diffusion models (DMs)
>   -  In this work, we show that the Fokker-Planck equation associated with a DM can provide a LID estimator which *addresses all the aforementioned deficiencies*
>   - I don't think the work addresses all the aforementioned deficiencies. There are still major questions regarding the accuracy of the resulting estimates in practice.
> - Other changes were more favorable
>   - (before) outperforms existing baselines on LID estimation benchmarks
>     - (after) outperforms existing baselines on synthetic LID estimation benchmarks while staying robust to the choice of hyperparameter <<<
>   - (after) Despite the resulting LID estimates being less stable over the choice of hyperparameter, FLIPD remains a valid measure of relative image complexity <<<
>   - (before) FLIPD exhibits higher correlation with non-LID measures of complexity
>     - (after) FLIPD exhibits a consistently higher correlation [..]
>   - (before) remain tractable with high-resolution images at the scale of Stable Diffusion
>     - (after) first one to be tractable at the scale of Stable Diffusion

---

> > ### Author Response · Authors · 2024-08-13
> > **Discussion**
> >
> > Thank you once again for the additional feedback.
> >
> > We will incorporate points (1) and (3) from your TLDR into our final version of the paper. As for point (2), we are not sure how you want us to surface these issues: the discussion about $t_0$ is very technical and we believe that it only makes sense to discuss this once the method has been fully presented. We are nonetheless happy to surface this much more clearly throughout the experiments, and also note that in the updated version of the introduction from our previous post, we do discuss a lack of stability of the FLIPD estimates when using UNets, which we believe gets to the point you are concerned about. Also, as for point (4), we will rephrase, but we do highlight that calling FLIPD more accurate than other LID estimators is perfectly valid, see e.g. Table 1. We will also clarify the relationship between our theory, knees, and architectures when updating the manuscript.
> >
> > `(Note that those last two remarks seem to contradict each other)`
> >
> > We do not believe these remarks contradict each other: the theory holds when the true score function $s$ is used, but FLIPD uses a learned score function $\hat{s}$ given by a neural network. If two different architectures managed to recover the true score, they would produce the exact same FLIPD estimates. In practice different architectures do not recover the exact same function, and each has different inductive biases. Thus, some deviation from the theory (which assumes access to the exact score function) is not contradictory, and neither is it to observe different behaviours from different architectures. We will also elaborate on this point for added clarity in the camera-ready version.
> >
> > `whereas the experiments section mainly focuses on validating the estimates to demonstrate the theory despite the accommodations needed to make it work.`
> >
> > We respectfully disagree with this characterization of the experiments section, which we believe achieves more than just validating theory. The section has two main goals. First, to validate our estimator on synthetic data where we have access to ground truth LID values; this is indeed about validating theory and about establishing that FLIPD outperforms competing estimators. Second, on images, we aim to show that FLIPD values, regardless of how accurate they are as LID estimates (because of the lack of knees), remain meaningful measures of relative image complexity and produce useful rankings of images based on complexity; this part is not about validating theory, but about showing the practical usefulness of FLIPD as a measure of complexity.

---

> > > ### Comment · Reviewer_MJca · 2024-08-13
> > > **Thank you**
> > >
> > > Thanks for your continued engagement and patience with my detailed inquisitions.
> > >
> > > Now that we're on the same page, we can actually think about the remaining questions regarding (2) and (4). I'm noting the following, which I'm only sharing as suggestions:
> > > - (re: 4) L49 and the abstract: the word "address" may give the impression that the noted shortcomings are resolved, which is not the same as pushing SOTA with remaining shortcomings still.
> > > - (re: 2) L60-61: this is a good place to discuss how the theory holds in the limit. I gather this is a common situation for algorithms based on DEs, and can be seen in how diffusion models need to sample time with carefully tuned schedules as well as e.g. Varadhan’s formula for the heat equation. A brief survey of techniques to cope with such difficulties would be appreciated, especially in light of the known numerical instabilities, and the aim to require a single ODE solve.
> > >
> > > For the authors' reference:
> > > - Yes, the two remarks aren't necessarily contradictory but the first one can be taken to over promise while only the second acknowledges the intricacies of practical implementations.
> > > - Yes, that characterization was a bit blunt. Doing my best to share feedback in time before the rebuttal window closes.

---

### Official Review · Reviewer_g4nR · 2024-07-26

**Soundness:** 3
**Presentation:** 2
**Contribution:** 3
**Rating:** 6
**Confidence:** 4

**Summary:**

The paper addresses the challenge of estimating the local intrinsic dimension (LID) of high-dimensional data, a measure reflecting the number of local factors of variation and data complexity. Traditional methods for LID estimation have limitations such as inaccuracy, high computational demand, and dependency on multiple pre-trained models. This work introduces FLIPD, an LID estimator based on the Fokker-Planck equation associated with a single pre-trained diffusion model. FLIPD outperforms existing baselines in LID estimation benchmarks and remains tractable even for high-resolution images. The authors adapt the LIDL method to DMs, leveraging the Fokker-Planck equation to enhance efficiency and accuracy. FLIPD's performance is validated through theoretical motivations and empirical results, demonstrating its superior correlation with non-LID complexity measures and qualitative complexity assessments compared to other estimators.

**Strengths:**

**1.** This paper introduces a new approach to estimating local intrinsic dimension (LID) by leveraging the Fokker-Planck equation associated with diffusion models (DMs). This new method effectively addresses several limitations of existing LID estimation techniques, such as inaccuracy, high computational demand, and the need for multiple pre-trained models.

**2.** The authors thoroughly validate the performance of the FLIPD estimator through a series of benchmark tasks, demonstrating its superiority over existing baselines, particularly in high-dimensional settings.

**Weaknesses:**

**1.** The most critical section of this paper is Section 3.2, which introduces the core innovation of the FLIPD estimator. However, the insights from Equations (12) and (13) are not immediately clear. It is difficult to understand the insights of these derivations. Moreover, many details are deferred to appendix. The authors should provide more detailed explanations of the Fokker-Planck equation and its relevance to LID estimation.

**2.** One weakness of the paper is the unclear demonstration of the computational advantages of the proposed FLIPD estimator compared to the method in [58]. The authors argue that while the method in [58] addresses some limitations of LIDL, it remains computationally expensive. However, the paper lacks a thorough theoretical analysis or experimental evidence to substantiate the claim that FLIPD is more computationally efficient.

**Questions:**

**Q1.** It seems that the proposed estimator over-estimate the LID compared to [10,48]. Can you provide some insights?

**Q2.** How does the FLIPD estimator maintain accuracy and efficiency in LID estimation across varying levels of data complexity and different types of high-dimensional datasets, and what specific mechanisms within the Fokker-Planck framework contribute to these properties?

---

> ### Author Rebuttal · Authors · 2024-08-07
>
> We thank you for the positive feedback and the insightful comments. If the following responses satisfactorily address your concerns, we kindly request you consider raising your score!
>
> ## Weakness 1
> Thank you for bringing this concern to our attention. We have provided a clarification for Section 3.2 and how it relates to the Fokker-Planck equation, along with some additional intuition. This will be included in the final version of the paper.
>
> The Fokker-Planck equation is a partial differential equation (PDE) that describes how the marginal probabilities of an SDE evolve over time $t$. Formally, it states that for the given SDE in Eq. (1), we have the following:
> \begin{equation}
> \frac{\partial}{\partial t} p(x, t) = - \frac{\partial}{\partial x} \left[ f(x, t) p(x, t)\right] + \frac{\partial^2}{\partial x^2} \left[ g^2(t) p(x, t) /2\right]
> \end{equation}
> Given this, it is straightforward to derive the PDE for $\log p(x, t)$ too. While Section 3.1 explores the connections between $\varrho(x, \delta)$ and $\log p(x, t)$, Section 3.2 utilizes the PDE for the evolution of $\log p(x, t)$ to develop an ODE that describes how $\varrho(x, \delta)$ evolves as a function of $\delta$. Consequently, the entire trajectory of $\varrho(x, \delta_1)$ through $\varrho(x, \delta_m)$ can be determined using a *single* ODE solver.
>
> Let us begin by explaining how Eq. (12) relates to the Fokker-Planck equation. This derivation involves two main steps: (i) rewriting the $\log \varrho(x, \delta)$ term on the LHS using the log marginal probabilities from Eq. (11), and (ii) transforming the differentiation with respect to the log-standard deviation $\delta$ into the *time domain* using the mapping $t(\delta)$. The following derivation formally connects the LHS of Eq. (12) to the Fokker-Planck equation:
> \begin{equation}
> \frac{\partial}{\partial \delta} \varrho(x, \delta) = \frac{\partial}{\partial \delta} D \log \gamma(\delta) + \frac{\partial}{\partial \delta} \log p(\gamma(\delta) x, t(\delta)) \quad \text{Using (11)}
> \end{equation}
> \begin{equation}
> = \frac{\partial}{\partial \delta} D \log \gamma(\delta) +  \frac{\partial t(\delta)}{\partial \delta}  \underset{\text{Rewrite using Fokker-Planck}}{\underbrace{\frac{\partial}{\partial t(\delta)} \log p(\psi(t(\delta)) x, t(\delta))}}
> \end{equation}
> $\frac{\partial}{\partial t(\delta)} \log p(\cdot, t)$ can be rewritten in terms of $s(x, t)$ (Eq. (41) of Appendix C.2) and replacing the Fokker-Planck term above will yield $\nu$ in Eq. (12). By solving the entailed ODE, we can evaluate $\varrho(x, \delta)$ at multiple $\delta$s and use the linear regression in LIDL to estimate LID.
>
> Furthermore, while a general solution for Eq. (12) can be obtained, it will always differ by a constant from the true $\varrho(x, \delta)$ due to the unspecified and difficult-to-determine initial value $\varrho(x, \delta_1)$. Eq. (13) introduces an initial value problem where this constant is *artificially* set to zero, defining its solution as $\hat{\varrho}(x, \delta)$. Notably, although $\hat{\varrho}(x, \delta)$ and $\varrho(x, \delta)$ differ by a constant, we perform a linear regression on $m$ sampled points from these functions, and this constant only affects the intercept in the regression, not the slope. LIDL focuses solely on the slope and not the intercept; thus, regression on $\hat{\varrho}$ instead of $\varrho$ yields the same result. Finally, we derive an LID estimator that directly extends LIDL, requiring only a single pre-trained diffusion model and a single ODE solve.
>
> ## Weakness 2
> The computational benefits are one of the main advantages of our method, and we will substantiate this further in the final version of the paper. We have included a dedicated section with time comparisons to the NB method from [58] in the general rebuttal to address this concern.
>
> ## Question 1
> Both [10] and [48] use the model-free, KNN-based MLE estimator. As outlined in Table 7 of Appendix D, MLE severely underestimates the true LID on synthetic data where the ground truth is known. We note that all the experiments in [10] and [48] are conducted on images, where the ground truth LID is unknown. Thus, in the image context, it is impossible to concretely say whether FLIPD overestimates LID, or MLE underestimates it. However, the fact that MLE underestimates LID in synthetic scenarios suggests that the estimated values on high-dimensional images are likely too low as well. Additionally, we point out that our estimate of average LID for MNIST closely aligns with that of [58], which is a more recent paper.
>
> ## Question 2
> We are not sure we entirely understand the question - please respond back with clarification so that we can better address your concerns and continue the discussion. Regardless, we would like to emphasize that the Fokker-Planck equation itself does not inherently provide better LID estimates; rather, it is a tool for efficiently extracting LID from a diffusion model trained to fit the data manifold. Our estimates are more accurate than prior methods because we use diffusion models, whereas many baseline methods are either model-free or use normalizing flows. Normalizing flows are known to struggle with data coming from low-dimensional manifolds embedded in high-dimensional space [38, A], whereas diffusion models do not exhibit these limitations [B]. In fact, we further study the NB estimator, which also uses a pretrained diffusion model, in Appendix D. We find that enhancements to the NB estimator could yield a highly accurate estimator, despite remaining intractable for high-dimensional data. This supports the notion that the high quality of our LID estimator is mainly due to the superior performance of diffusion models, rather than the Fokker-Planck equation.
>
> # References
>
> [A] Behrmann et al. "Understanding and mitigating exploding inverses in invertible neural networks." ICML 2021.
>
> [B] Pidstrigach. "Score-based generative models detect manifolds." NeurIPS 2022.

---

> > ### Comment · Reviewer_g4nR · 2024-08-11
> >
> > Thanks for the reviewers' rebuttal. My main concerns have been addressed. I have raised my score.

---

> > > ### Author Response · Authors · 2024-08-12
> > > **Thank you**
> > >
> > > Thank you! We are happy we were able to address your concerns.

---

### Author Rebuttal · Authors · 2024-08-07

We greatly appreciate the time reviewers have spent on our paper and are delighted to see that all four reviewers recommended acceptance. Reviewers found our method “well-principled” and “clearly written” (**MJca**, **9X9u**, **MdUJ**) and described it as “a really excellent and well-executed effort” (**MdUJ**). They also noted that our method is “thoroughly motivated” by our theory (**g4nR**) and synonymously praised our experimental results as “thorough” (**g4nR**) and posed them as enabling further testing against prior baselines (**MJca**). Additionally, two reviewers highlighted the value, informativeness, and illustrative nature of the links established between LID estimates and PNG compression size (**MJca**, **MdUJ**).

Here, we provide an in-depth discussion of the computational benefits of our method compared to other baselines. Additionally, we present time comparisons from our experiments using our GPU hardware, demonstrating that FLIPD is the **only** estimator capable of scaling up to Stable Diffusion-scale data, further highlighting the scalable nature of our approach.

Before we continue, we find NB to be the only relevant baseline for this comparison because:
1. LIDL, the only other model-based LID estimator, requires training multiple models, which is extremely time-consuming. For example, obtaining LID estimates for LAION would necessitate training multiple instances of Stable Diffusion, making it impractical. Additionally, while LIDL suggests using normalizing flows for their straightforward likelihood computation, it is well-known that normalizing flows struggle to produce high-quality images even on relatively low resolution data such as CIFAR10, let alone on high-resolution, internet-scale datasets such as LAION.
2. Traditional model-free methods, such as ESS, LPCA, or MLE, not only underperform on high-dimensional data but also require performing kNN on extremely high-dimensional data, which is impractical. In fact, as detailed in Appendix D, for baseline comparisons on these methods, we had to use subsamples on synthetic datasets with 1000 dimensions because computing LID estimates for the entire dataset would have taken more than a day.

To further substantiate our claims, we included time comparisons for our image experiments in the table below, corresponding to the results reported in Table 2 of the main text, as well as all of our Stable Diffusion results. We grouped the data in Table 2 into two categories: RGB datasets with 32x32 dimensions and 3 channels (SVHN and CIFAR10), and grayscale datasets with 28x28 dimensions (MNIST and FMNIST). Our results show how long it takes to estimate LID for a single datapoint. We see that FLIPD is more than **10 times faster** than NB on grayscale data and more than **100 times faster** on RGB data, demonstrating that as dimensionality increases, FLIPD remains the only scalable estimator. Finally, as depicted in the table, FLIPD is the only estimator capable of estimating LID for LAION.

To dive deeper into why the NB baseline cannot estimate LID for LAION, we note that the method requires $4 \times D$ forward passes of the score network to construct a matrix $\mathbf{S}(x)$ of dimension $D$ by $4D$. This is then followed by a singular value decomposition, meaning that this portion of NB scales *cubically* with $D$, the ambient dimension of the data. In our experiments with LAION images, we found that just constructing this matrix $\mathbf{S}(x)$ for a single data point takes over *2.5 hours*, and performing the singular value decomposition on such a large matrix is **intractable** using the built-in PyTorch linear algebra functionality.

Finally, for the first two columns of the table below and the results in Table 2, we used 50 JVPs for our Hutchinson estimator and a standard **UNet architecture** from the diffusers library. It is worth noting that we can adjust the number of JVPs to trade-off between accuracy and performance. For our Stable Diffusion results, we opted for a *single* JVP to estimate the LID for a given datapoint which still gave a reasonable measure of complexity while only taking *195 milliseconds*. Notably, all experiments were conducted on a single NVIDIA A100-40GB GPU.

| Dataset               | 28x28 (Greyscale)        |    32x32 (RGB)     | LAION (High-resolution) |
|-----------------------|:--------------:|:--------------:|:--------------:|
| FLIPD | **0.101** seconds | **0.133** seconds | **0.195** seconds |
| NB                    |  1.648 seconds | 10.766 seconds | - |

---

### Decision · Program_Chairs · 2024-09-25

**Decision:**

Accept (spotlight)

**Comment:**

This paper proposes a Fokker-Planck-based estimator for local intrinsic dimension estimation, which uses diffusion models in implementation and scales to high-resolution images at the level of Stable Diffusion. Both the theoretical and empirical contributions are appraised by the reviewers.

The proposed method is indeed novel and demonstrates promising performance compared with existing baselines. The paper is clearly written in general and I invite the authors to revise according to the discussion with the reviewers. (Special thanks to Reviewer MJca and the authors for the very dedicated efforts in revision and I trust the authors to make necessary polishing in the final version.) An outstanding concern about the computation efficiency in the initial review is further addressed by the authors' general rebuttal. Therefore, I am recommending acceptance of the paper.